# Runoff from Greenland's firn area – why do MODIS, RCMs and a firn model disagree?

Horst Machguth[1], Andrew Tedstone[1, 2], Peter Kuipers Munneke[3], Max Brils[3, 4], Brice Noël[5], Nicole Clerx[1, 6], Nicolas Jullien[1], Xavier Fettweis[5], and Michiel van den Broeke[3]

[1]Department of Geosciences, University of Fribourg, Fribourg, Switzerland
[2]Institute of Earth Surface Dynamics, University of Lausanne, Lausanne, Switzerland
[3]Institute for Marine and Atmospheric research Utrecht, Utrecht University, Utrecht, Netherlands
[4]Geography and Environmental Sciences Department, Northumbria University, Newcastle, United Kingdom
[5]Laboratory of Climatology, Department of Geography, SPHERES research unit, University of Liège, Liège, Belgium
[6]Environmental Remote Sensing Laboratory, EPFL, Lausanne, Switzerland

**Correspondence:** Horst Machguth (horst.machguth@unifr.ch)

**Abstract.** Due to increasing air temperatures, surface melt and meltwater runoff expand to ever higher elevations on the Greenland ice sheet and reach far into its firn area. Here, we evaluate how two regional climate models (RCMs) simulate the expansion of the ice sheet runoff area: MAR, and RACMO with its offline firn model IMAU-FDM. For the purpose of this comparison we first improve an existing algorithm to detect daily visible runoff limits from MODIS satellite imagery. We then apply the improved algorithm to most of the Greenland ice sheet and compare MODIS to RCM runoff limits for the years 2000 to 2021. We find that RACMO/IMAU-FDM runoff limits are on average somewhat lower than MODIS and show little fluctuation from year to year. MAR runoff limits are substantially higher than MODIS, but their inter-annual fluctuations are more similar to MODIS. Both models apply a bucket scheme to route meltwater vertically. Using the K-transect as an example, we demonstrate that differences in the implementation of the bucket scheme are responsible for the disparity in RCM simulated runoff limits. The formulation of the runoff condition is of large influence: in RACMO/IMAU-FDM meltwater is only considered runoff when it reaches the bottom of the simulated firn pack; in MAR runoff can also occur from within the firn pack, which contributes to its high runoff limits. We show that total runoff along the K-transect, simulated by the two RCMs, diverges by up to 29 % in extraordinary melt years. This difference is mostly caused by the diverging simulated runoff limits, which shows the importance of the melting firn areas in Greenland's mass changes.

## 1 Introduction

Polar regional climate models (RCMs) are widely used to assess past, present and future surface mass balance of the Greenland and Antarctic ice sheets (Box et al., 2004; Fettweis et al., 2008; Noël et al., 2016; IMBIE Team, 2018, 2020). The accuracy of RCM output relies, among other factors, on data available for model calibration and evaluation. Essential for RCM evaluation are meteorological observations (e.g. Steffen and Box, 2001; Fausto et al., 2021), surface mass balance measurements (e.g. Benson, 1962; Greuell et al., 2001; van de Berg et al., 2006; Machguth et al., 2016b; Karlsson et al., 2016; Fausto et al., 2021) and remote sensing products (e.g. Fettweis et al., 2006; Mohajerani et al., 2019; Slater et al., 2021). RCMs have been

extensively evaluated for Greenland's ablation area (e.g. Gallee and Duynkerke, 1997; Lefebre et al., 2005; Noël et al., 2016) and its higher accumulation area (e.g. Rae et al., 2012; Noël et al., 2016) and have been found to perform well when compared to meteorological observations, surface mass balance measured at stake locations as well as in ice cores and gravimetric ice sheet mass balance (e.g. Fettweis et al., 2017, 2020).

Comprehensive model evaluation requires also testing the RCMs in the transition zone in-between the ablation and the higher accumulation area. In this area a delicate balance exists between accumulation and ablation processes. In summer, when melt, runoff and accumulation can occur simultaneously, working conditions are challenging (e.g. Holmes, 1955; Clerx et al., 2022). Consequently, the availability of field data is limited and few studies (e.g. Covi et al., 2022; Zhang et al., 2023) have evaluated RCMs in this transition zone.

Within the elevation range of the transition zone lie the equilibrium line altitude (ELA) and the runoff limit. The former is the elevation which separates accumulation and ablation areas, the latter is defined as the uppermost elevation from where meltwater can reach the ocean and contribute to mass loss (Cogley et al., 2011; Tedstone and Machguth, 2022; Clerx et al., 2022). The elevation of the equilibrium line and the runoff limit varies from year to year in response to weather conditions. Thereby, the runoff limit lies within the accumulation area (Shumskii, 1955, 1964) and is thus located above the ELA.

Tedstone and Machguth (2022) compared seasonal maxima of visible runoff limits mapped from Landsat satellite imagery to runoff extent simulated by the two RCMs RACMO 2.3p2 (Noël et al., 2018) and MAR v3.11 (Fettweis et al., 2017) forced by ERA-40/ERA-I/ERA5. The comparison revealed substantial differences between RCMs and remotely sensed visible runoff limits, but also between the two RCMs involved. While remotely sensed visible runoff limits are subject to uncertainties, it remains unclear what causes the remarkable differences between the RCMs. If RCMs differ in simulating the runoff extent of the Greenland ice sheet, this results in inaccuracies in future scenarios of mass loss and sea-level contribution. Indeed, Glaude et al. (2024) found large differences in RCM simulated runoff area for the year 2100 under a high-end warming scenario (SSP5-8.5). Glaude et al. (2024) point out that the three RCMs studied, among them RACMO and MAR, differ by a factor of two in their predicted surface mass balance for the year 2100.

Here we aim at explaining why simulated runoff limits differ between models. For this purpose we compare remotely sensed visible runoff limits and simulated runoff limits by MAR, RACMO and the firn model IMAU-FDM. We use daily Moderate Resolution Imaging Spectroradiometer (MODIS) visible runoff limits for the years 2000 to 2021, derived by an improved version of the algorithm used in Machguth et al. (2023). We use MODIS visible runoff limits instead of the aforementioned Landsat visible runoff limits because the former offer higher temporal resolution. We analyze the differences between observed and modelled runoff limits in the context of modelled parameters that potentially influence simulated runoff. Among the selected parameters are surface albedo, firn density and temperature, as well as refreezing. We identify which of the parameterizations in the models likely cause the deviations. Finally, we quantify their impact on simulated mass balance along a transect in south-west Greenland.

## 2 Data

### 2.1 Data for MODIS visible runoff limit detection

The detection of MODIS visible runoff limits $\Upsilon_{\mathrm{obs}}$ is based on an optimized version of the algorithm by Machguth et al. (2023). The improved algorithm (Sec. 3.1) relies on the following input: (i) daily MOD10A1 data (MODIS/Terra Daily Snow Cover at 500 m resolution, version 6.0; Hall and Riggs, 2016); (ii) daily MOD09GA data (MODIS/Terra Surface Reflectance Daily at 500 m, version 6.0; Vermote and Wolfe, 2015); (iii) the Arctic DEM (100 m resolution mosaic, v.3.0; Porter et al., 2018, here downsampled to the 500 m MODIS grid); (iv) outlines of the Greenland ice sheet according to Rastner et al. (2012) and (v) Greenland-wide arrays of surface ice flow velocity in x and y direction (Joughin et al., 2016, 2017).

### 2.2 Model data

To quantify modelled runoff limits $\Upsilon_{\mathrm{rcm}}$ we use (i) simulated runoff from the polar regional climate model MAR (version 3.14, 10 km resolution, forced by ERA5), (ii) the polar version of the Regional Atmospheric Climate Model RACMO (Noël et al., 2019, version 2.3p2 on the grid FGRN055, forced by ERA5) at a resolution of 5.5 km as well as (iii) the offline firn model IMAU-FDM v1.2G (Ligtenberg et al., 2011, 2018; Brils et al., 2022). Descriptions of these three models, with special focus on their firn simulation, are provided in Sec. 3.2. RACMO data are frequently used in a version that is further downscaled to 1 km resolution and bias corrected (Noël et al., 2016; Noël et al., 2019). The downscaled data have a temporal resolution of 1 day, which is insufficient to force IMAU-FDM v1.2G, so for firn applications these data cannot be used. Nevertheless, as the 1 km data are frequently applied to assess Greenland mass balance we here provide basic comparison to the other models and MODIS.

We use a set of RCM parameters (Table 1) to explore the reasons behind potential differences in MODIS and RCM runoff limits. Various parameters are unavailable from RACMO2.3p2 and are instead obtained from the offline firn model IMAU-FDM v1.2G henceforth IMAU-FDM. The model is forced in offline mode by RACMO2.3p2 and is run on an identical spatial grid. In the following we refer to 'MAR' for MARv3.14, to 'RACMO' for RACMO2.3p2 at native resolution of 5.5 km and we use 'RACMO 1 km' when we refer to downscaled and bias corrected RACMO2.3p2 data.

MAR output and RACMO 1 km data are obtained at daily temporal resolution. Output from RACMO and IMAU-FDM are at 10-day intervals. Where needed, MAR data are averaged or summed to the lower temporal resolution.

In the following our usage of the term RCM also refers to the offline firn model IMAU-FDM. As explained in Sec. 3.2.2, the latter is forced by RACMO and very similar to RACMO's firn module whose output is not available at a sufficient level of detail for the present study.

**Table 1.** List of RCM simulated parameters used to calculate and investigate runoff limits. "RACMO 1 km" stands for RACMO2.3p2 downscaled to 1 km and bias corrected, "RACMO" stands for RACMO2.3p2 at native 5.5 km resolution, "IMAU-FDM" stands for IMAU-FDM v1.2G and "MAR" stands for MARv3.14.

| Parameter | Source | | | | Unit | Description |
|---|---|---|---|---|---|---|
| | RACMO 1 km | RACMO | IMAU-FDM | MAR | | |
| $\alpha$ | | x | | x | | Surface albedo |
| $C$ | | x | | x | m w.e. | Accumulation |
| $M$ | | x | | x | m w.e. | Melt |
| $R$ | | | x | x | m w.e. | Refreezing |
| $Q$ | x | | x | x | m w.e. | Runoff |
| $\mathrm{fac}_{10m}$ | | | x | x | m | Firn air content top 10 m |
| $\mathrm{lwc}_{1m}$ | | | x | x | kg | Liquid water content top 1 m |
| $\mathrm{lwc}_{tot}$ | | | x | x | kg | Liquid water content 0 to 20 m depth |
| $T$ | | | x | x | °C | Firn/ice temperature profile 0 to 20 m depth |
| $T_{10m}$ | | | x | x | °C | Firn/ice temperature at 10 m depth |
| $\rho$ | | | x | x | $\mathrm{kg\,m^{-3}}$ | Density profile 0 to 20 m depth |

## 3 Methods

### 3.1 Detecting MODIS $\Upsilon_{\mathrm{obs}}$ along flowlines

The algorithms by Greuell and Knap (2000) and Machguth et al. (2023) detect $\Upsilon_{\mathrm{obs}}$ on relatively coarse resolution AVHRR
(1.1 km; Greuell and Knap, 2000) or MODIS (500 m; Machguth et al., 2023) satellite imagery. Given the low spatial resolution as compared to e.g. Landsat, $\Upsilon_{\mathrm{obs}}$ is identified indirectly, that is where spatial variability of surface albedo $\alpha$ transitions from low to high. Low spatial variability of $\alpha$ indicates a monotonous snow covered surface. Variability of $\alpha$ is high where dark meltwater streams, lakes and slush fields intersect the bright snow cover. Despite this indirect approach, MODIS $\Upsilon_{\mathrm{obs}}$ highly agree with visible runoff limits detected on finer resolution (30 m) Landsat imagery (Machguth et al., 2023).

Machguth et al. (2023) scanned rectangular polygons of width $p_w$ and length $p_l \gg p_w$ for the location where the standard deviation of surface albedo $\sigma_\alpha$ falls below a certain threshold. If a set of additional conditions and tests are fulfilled (see Machguth et al., 2023), the location is considered to represent $\Upsilon_{\mathrm{obs}}$. The long axes of the polygons needed to be oriented along the strongest gradient in $\alpha$, which is in the direction of the surface slope. Polygons in Machguth et al. (2023) were strictly oriented west-east. Consequently, the application of the method was restricted to areas of the western flank of the ice sheet.

Here we apply the method by Machguth et al. (2023) with two major modifications that allow application to all of the Greenland Ice Sheet: (1) We create so called flowline-polygons of $p_w = 20$ km, henceforth simply called flowlines, and (2) implement an improved calculation of $\sigma_\alpha$. The former allows detection of $\Upsilon_{\mathrm{obs}}$ in complex topography sloping in any direction, the latter improves detection of $\Upsilon_{\mathrm{obs}}$ by calculating and subtracting the influence of temporally persistent albedo features. These

modifications, as well as smaller optimizations, are detailed in Appendix A and Fig. A1. For further details on the algorithm
we refer to Machguth et al. (2023).

## 3.2 RCM simulations of the firn cover and runoff

### 3.2.1 MAR

We use daily outputs at 10 km resolution from version 3.14 of MAR, forced every 6 hours by the ERA5 reanalysis. The data
are composed of two transient simulations: the first one starts in September 1974 but only the period 1980-1999 is used. The
second one begins in September 1994 and the period 2000-2023 is used. Together, the two simulations cover the years 1980 to
2023. In the set-up used here, MAR resolves the uppermost 21 m of snow and firn using a time-varying number of layers up
to a maximum of 21 layers. For densities lower than $450 \, \mathrm{kg \, m^{-3}}$, the CROCUS snow model albedo (Brun et al., 1992) is used
with a minimum value of 0.7. Where snowpack is present but has a surface density higher than $450 \, \mathrm{kg \, m^{-3}}$ (the maximum
density of pure snow), then the minimum value of albedo declines between the minimum pure snow albedo (0.7) and clean ice
albedo (0.55) as a linear function of increasing density. On bare ice (surface density higher than $900 \, \mathrm{kg \, m^{-3}}$), CROCUS snow
model albedo is not used and the albedo varies exponentially between 0.55 (clean ice) and 0.5 (wet ice) as a function of the
accumulated surface water height and the slope (Lefebre et al., 2003). The dependency on water depth and slope is of limited
impact but is maintained to address the effect of supraglacial lakes in future model versions.

The main changes of MARv3.14 with respect to MARv3.12 (Vandecrux et al., 2024) are as follows: Some bugs in the clouds
scheme have been corrected and a continuous snowfall-rainfall limit has been introduced for near-surface temperature between
-1 °C (100 % of precipitation falls as snow) and +1 °C (100 % rain). MARv3.14 now uses the radiative scheme from ERA5
(Hogan and Bozzo, 2018; Grailet et al., 2025) instead of the one from ERA40 (Morcrette, 2002) in former MAR versions.

MAR parameterizes meltwater percolation through an instantaneous bucket scheme. Slush is not allowed in MARv3.14
simulations and the maximum liquid water saturation in snow and firn (i.e. irreducible water saturation, expressed in % of the
pore volume) is 7 % at the surface and linearly reduces to 2 % at 1 m depth. Below that depth, irreducible water saturation
is set to 2 %. Meltwater that percolates into a snow or firn layer can refreeze if the layer temperature is below 0 °C or it can
be retained as irreducible water if the layer is temperate. If neither of the two processes are possible, that is if the layer has
become temperate and irreducible water saturation is at its maxima, the remaining meltwater will either percolate to the next
layer below or run off immediately. The following conditions decide between percolation and immediate runoff. If the density
of a layer is $< 830 \, \mathrm{kg \, m^{-3}}$, percolation to the next deeper layer takes place. For layers of density $\geq 830 \, \mathrm{kg \, m^{-3}}$, a density
runoff threshold determines how much of any meltwater gets removed immediately as runoff: 0 % for $830 \, \mathrm{kg \, m^{-3}}$ to 100 % for
densities above $900 \, \mathrm{kg \, m^{-3}}$. The remainder percolates to the next layer below. Where ice lenses are simulated by MAR, $2/3$
of the percolating meltwater progress to underlying layers and the remaining $1/3$ are considered run off. Thereby an ice lens is
defined as a layer with a density of $> 900 \, \mathrm{kg \, m^{-3}}$ that lies on top of a layer where density is $\leq 900 \, \mathrm{kg \, m^{-3}}$. Furthermore, any
meltwater that reaches the bottom of the MAR firn column is also considered runoff.

For further details on MAR we refer to Fettweis et al. (2013, 2017, 2020). Previous MAR versions have been successfully validated over the Greenland ice sheet by comparison with surface mass balance measurements (Fettweis et al., 2020), satellite derived melt extent (Fettweis et al., 2011) and *in situ* atmospheric measurements (Delhasse et al., 2020).

### 3.2.2 IMAU-FDM and RACMO

We use data from the RCM RACMO and the offline firn model IMAU-FDM, which is very similar to RACMO's firn module. While it would be preferable to consistently use RACMO data for comparison to MAR, we here use IMAU-FDM firn simulations because RACMO outputs only depth integrated firn data. In the following we first explain IMAU-FDM, then explain differences to RACMO's firn module, and finally provide information on RACMO and its forcing of IMAU-FDM.

IMAU-FDM v1.2G (Brils et al., 2022) is a semi-empirical firn densification model that simulates the time evolution of 140 firn density, temperature, liquid water saturation and changes in surface elevation owing to variability of firn depth. Vertical water transport in IMAU-FDM is instantaneous and calculated via the bucket method. When liquid water is added to the firn column by melt or rain, it is transported vertically downwards. Starting at the uppermost model layer, the scheme checks if there is cold content and pore space available for refreezing. If so, refreezing takes place, raising the layer's temperature and density, until either all water has been refrozen, the layer has turned into ice (i.e. has a density of $917\,\mathrm{kg\,m^{-3}}$), or reaches $0\,^{\circ}\mathrm{C}$. 145 Irreducible water will be retained in liquid form within the pores of a temperate firn layer. The maximum amount that can be retained depends on the layer's porosity, following Coléou and Lesaffre (1998) (irreducible water saturation is $\sim 5.8\,\%$ at a snow density of $300\,\mathrm{kg\,m^{-3}}$ and $\sim 15\,\%$ at $800\,\mathrm{kg\,m^{-3}}$. Any water that cannot refreeze or be retained as irreducible water will percolate to the next layer below. These steps are then repeated in the next firn layer, and so on until no more liquid water is present aside the irreducible water saturation within temperate layers. This is all done within a single time step, which means 150 that vertical percolation is instantaneous. The bucket method also implies that liquid water percolates through any ice layer, because they contain no pore space to accommodate refreezing. Water is not allowed to pond or run off on top of ice layers.

When the water reaches the interface between firn and glacial ice, it is assumed to run off instantaneously. The depth of the horizontal modelling domain of IMAU-FDM varies in space and time and is defined by the condition that the deepest 200 grid cells must all exceed a density of $910\,\mathrm{kg\,m^{-3}}$. Consequently the thickness of the firn layer, that is from the surface to the depth 155 below which all grid cells exceed a density of $830\,\mathrm{kg\,m^{-3}}$, varies and reaches maxima of $100\,\mathrm{m}$ in high-accumulation regions of the south-east of the ice sheet. A more typical maximum firn thickness is $\sim 70\,\mathrm{m}$.

RACMO's firn module also simulates the firn column from the surface down to glacial ice and uses similar physical parametrisations as IMAU-FDM, albeit at a lower vertical resolution (max. 100 but typically 40 layers in RACMO; up to 3000 layers in IMAU-FDM) and less comprehensive initialisation to save computing costs.

IMAU-FDM is forced at the upper boundary by 3-hourly RACMO surface temperature and mass fluxes, interpolated to 15 minutes. In RACMO, the snow albedo scheme is based on prognostic snow grain size, cloud optical thickness, solar zenith angle and impurity concentration in snow (Kuipers Munneke et al., 2011). Impurity concentration is assumed constant in time and space. Bare ice albedo is prescribed from the $500\,\mathrm{m}$ MODIS 16-day albedo version 5 product (MCD43A3v5) as the lowest $5\,\%$ surface albedo records for the period 2000–2015. Thresholds are applied to these values: minimum ice albedo is set to 0.3

for dark ice in the low-lying ablation zone, and a maximum value of 0.55 is used for bright ice under perennial snow cover in the accumulation zone, i.e. only used when all firn melts away which does not happen in this run. RACMO snow albedo typically ranges between ∼0.7 for highly metamorphosed, coarse grained snow under clear-sky conditions and ∼0.95 for fine grained snow under cloudy conditions. RACMO2.3p2 surface energy balance, surface mass balance and melt output over the GrIS have been extensively evaluated, notably along the K-transect, and were found to be generally robust (Noël et al., 2019).

The RACMO 1 km data are a statistically downscaled and bias corrected version of the RACMO2.3p2 data (Noël et al., 2019). Here we use only RACMO 1 km runoff which differs from the original RACMO data due to (i) an albedo bias correction, being applied only in the bare ice zone, and (ii) an elevation gradient correction (Noël et al., 2016; Noël et al., 2019). For details on downscaling and bias correction we refer to the aforementioned sources.

## 3.3   Calculating $\Upsilon_{\mathrm{rcm}}$ from RCM output

We distinguish between daily runoff limits $\Upsilon_{\mathrm{rcm}}$ and annual maximum runoff limits $\max\Upsilon_{\mathrm{rcm}}$, which mark the highest elevation where runoff occurs for each year. Both $\Upsilon_{\mathrm{rcm}}$ and $\max\Upsilon_{\mathrm{rcm}}$ are calculated on the same 20 km wide flowlines as used for the detection of $\Upsilon_{\mathrm{obs}}$. For each flowline, we consider RCM grid cells whose center falls within the flowline. Given the elevation of each grid cell and simulated runoff, we then calculate runoff against elevation.

  There is no generally accepted definition of $\max\Upsilon_{\mathrm{rcm}}$ in terms of runoff per year. Tedstone and Machguth (2022) quantified

the sensitivity of $\max\Upsilon_{\mathrm{rcm}}$ to runoff thresholds of >1, >5, >10, and >20 mm w.e. a$^{-1}$. They found that MAR and RACMO $\max\Upsilon_{\mathrm{rcm}}$ are rather insensitive to the choice of threshold. Furthermore, they stated that the uncertainties associated with the choice of thresholds are small compared to the substantial differences in $\max\Upsilon_{\mathrm{rcm}}$ between the two RCMs. We here adopted their chosen threshold of >10 mm w.e. a$^{-1}$ to calculate $\max\Upsilon_{\mathrm{rcm}}$. To estimate daily $\Upsilon_{\mathrm{rcm}}$ we use a threshold of >1 mm w.e. day$^{-1}$.

## 185  3.4   Analyzing RCM process simulations near the runoff limit

  Our goal is to understand why deviations occur (i) between $\Upsilon_{\mathrm{obs}}$ and $\Upsilon_{\mathrm{rcm}}$ and (ii) between the two $\Upsilon_{\mathrm{rcm}}$ (labeled $\Upsilon_{\mathrm{rcm}}^{\mathrm{IMAU-FDM}}$ and $\Upsilon_{\mathrm{rcm}}^{\mathrm{MAR}}$). We focus this part of the analysis on the K-transect which has been studied intensively with respect to ice sheet boundary layer meteorology (van den Broeke et al., 1994), surface mass balance (Van de Wal et al., 2005, 2012), firn processes (Machguth et al., 2016a; Mikkelsen et al., 2016; Rennermalm et al., 2021) and firn hydrology (Clerx et al., 2022). Here we

defined the K-transect as the line that follows the 67 °N parallel, starts at the ice margin at ∼250 m a.s.l. / 50 °W, and reaches to the ice divide at ∼2520 m a.s.l. / 42.7 °W (Fig. 1). For both RCMs and IMAU-FDM, we extract the grid cells which are closest to the ∼320 km long transect. This results in lines of RCM grid cells which are one cell wide and 33 (MAR) or 57 cells (RACMO, IMAU-FDM) in length.

  Along the K-transect we analyse the RCM simulated parameters listed in Table 1. We quantify temporal and spatial changes

and search for parameters that show peculiar or unexpected values in the broader elevation range around the runoff limit. If found, we investigate the underlying RCM parameterizations in order to understand their potential influence on $\Upsilon_{\mathrm{rcm}}$.

## 4  Results

### 4.1  MODIS $\Upsilon_{\mathrm{obs}}$ detections

Figure 1 summarizes the MODIS-derived $\Upsilon_{\mathrm{obs}}$ for all of the Greenland Ice Sheet. The approach creates few and meaningless
$\Upsilon_{\mathrm{obs}}$ in areas dominated by meltwater discharge through aquifers. This is to be expected as surface meltwater features are
largely absent in such areas. Consequently Fig. 1 does not show retrievals from 60 to 68.4 °N along Greenland's east coast.
However, we show detected $\Upsilon_{\mathrm{obs}}$ located in smaller aquifer regions elsewhere on the ice sheet. Excluding retrievals from 60 to
68.4 °N along the east cost, 63,400 $\Upsilon_{\mathrm{obs}}$ in 417 flowlines remain, which corresponds on average to ∼7 retrievals per flowline
and year. The actual number of annual retrievals varies geographically and is highest in the southwest, exceeding on average
18 retrievals per flowline and melt season.

Compared to Machguth et al. (2023) and their study area, we find that the updated algorithm yields ∼80 % more $\Upsilon_{\mathrm{obs}}$
detections. This difference is mainly due to the new algorithm being able to place more flowlines that are optimized for
complex topographies. The average number of $\Upsilon_{\mathrm{obs}}$ detections per flowline is 5.5 % higher than per stripe, which were the
strictly east-west oriented bands in Machguth et al. (2023). Outside of the area investigated by Machguth et al. (2023), the new
approach provides numerous detections of $\Upsilon_{\mathrm{obs}}$ in the north-west of Greenland, from near Pituffik Space Base to Humboldt and
Petermann glaciers, as well as in the region of the north-east Greenland ice stream. Few detections occur along the central part
of the east coast where the terrain is complex and steep, with numerous outlet glaciers. The approach appears not well suited
to such terrain because most outlet glaciers are narrow, compared to the 20 km width of the flowline polygons. Consequently,
along the outlet glaciers few glacier pixels are available for retrieval of the $\Upsilon_{\mathrm{obs}}$. Apart from Petermann Glacier, there are few
detections beyond 80 °N, the reasons for which are unclear. Tedstone and Machguth (2022), who used Landsat to detect surface
hydrology, also noted few detections in the region.

Figure A2 compares $\Upsilon_{\mathrm{obs}}$ to the Landsat-derived visible runoff limits from Tedstone and Machguth (2022). The comparison
yields a good agreement between the two data sets and is discussed in Appendix A.

Figures 2, C1 and C2 exemplify the temporal detail of the $\Upsilon_{\mathrm{obs}}$ data. The figures demonstrate frequent behavior where
$\Upsilon_{\mathrm{obs}}$ rises relatively early in the melt season and reaches a plateau before melting ends (see also Machguth et al., 2023). By
design of the detection and filtering algorithms, there is typically no decrease in $\Upsilon_{\mathrm{obs}}$ towards the end of the melt season: Most
decreasing $\Upsilon_{\mathrm{obs}}$ are filtered out because optical remote sensing is poorly suited to detect continued hydrological activity under
freshly fallen autumn snow (Machguth et al., 2023).

### 4.2  Comparing $\Upsilon_{\mathrm{obs}}$ and $\Upsilon_{\mathrm{rcm}}$

#### 4.2.1  Comparing annual maxima

Figure 3 shows how $\max\Upsilon_{\mathrm{obs}}$ and $\max\Upsilon_{\mathrm{rcm}}$ vary along Greenland's western flank. The RCMs and MODIS show a general
decrease of the runoff limit towards higher latitudes (Fig. 3b). Certain deviations from this trend are common to all data:
$\max\Upsilon_{\mathrm{obs}}$ and $\max\Upsilon_{\mathrm{rcm}}$ are depressed south of ∼63 °N and elevated in-between ∼71 °N and ∼72.5 °N. Where firn aquifers

are present, $\max\Upsilon_{\mathrm{obs}}$ are biased low and standard deviation is increased. Otherwise, the differences between $\max\Upsilon_{\mathrm{obs}}$ and $\max\Upsilon_{\mathrm{rcm}}$ depend strongly on the RCM. IMAU-FDM simulated runoff limits are mostly lower than $\max\Upsilon_{\mathrm{obs}}$ and they have low standard deviation in comparison to MODIS. MAR $\max\Upsilon_{\mathrm{rcm}}$ and its standard deviation are substantially higher than MODIS. Figure 4 illustrates for two selected regions how $\max\Upsilon_{\mathrm{obs}}$ and $\max\Upsilon_{\mathrm{rcm}}$ fluctuate over time. IMAU-FDM simulated runoff limits vary little between the years. The intense melt seasons of 2012 and 2019 leave virtually no trace in its runoff limits. MAR $\max\Upsilon_{\mathrm{rcm}}$ vary with the intensity of the melt season. Temporal variability of $\max\Upsilon_{\mathrm{rcm}}^{\mathrm{MAR}}$ exceeds MODIS in the south (Fig. 4b), but is rather similar further north (Fig. 4a).

Because RACMO 1 km data are frequently used in research, they are shown in Figs. 3 and 4 for the interested reader to assess differences to IMAU-FDM and MAR. RACMO 1 km data show $\max\Upsilon_{\mathrm{rcm}}$ that are on average similar to MODIS but with the same small standard deviation as IMAU-FDM. Similar to IMAU-FDM, RACMO 1 km runoff limits vary little between the years.

### 4.2.2 Comparing seasonal evolution of $\Upsilon_{\mathrm{obs}}$ and $\Upsilon_{\mathrm{rcm}}$

Comparing the seasonal evolution of $\Upsilon_{\mathrm{rcm}}$ and $\Upsilon_{\mathrm{obs}}$ shows that MODIS and RCM runoff limits often reach their seasonal maxima at similar points in time (Figs. 2, C1 and C2). The dates of the first appearance of the runoff limit are often similar between RCMs and MODIS. However, $\Upsilon_{\mathrm{rcm}}$ fluctuate strongly, often dropping and increasing, within a few days, over hundreds of meters in elevation (e.g. Fig. 2). The effect is more pronounced for MAR which is due to the higher temporal resolution of the MAR data. MODIS $\Upsilon_{\mathrm{obs}}$ indicate a more continuous process where the visible runoff limit remains at high elevations, also during cold spells.

Agreement of $\Upsilon_{\mathrm{rcm}}^{\mathrm{IMAU-FDM}}$ to the seasonal evolution of $\Upsilon_{\mathrm{obs}}$ is generally good (Figs. 2, C1 and C2). However, $\Upsilon_{\mathrm{rcm}}^{\mathrm{IMAU-FDM}}$ always tends to reach its maxima at very similar elevations, regardless of the intensity of the melt season. This is the same behavior shown for $\max\Upsilon_{\mathrm{rcm}}^{\mathrm{IMAU-FDM}}$ in Fig. 4. MAR $\Upsilon_{\mathrm{rcm}}$ typically overshoot $\Upsilon_{\mathrm{obs}}$ (Figs. C1 and C2).

### 4.3 RCM process simulations at the runoff limit

Potential causes for the large differences between $\Upsilon_{\mathrm{rcm}}$ are (i) differences in the amount of simulated melt or snowfall in MAR or RACMO, or (ii) differences in the firn parameterizations that impact simulated runoff. In Appendix B we demonstrate that differences in melt or accumulation at the $\max\Upsilon_{\mathrm{rcm}}$ are small and cannot explain the differences between $\max\Upsilon_{\mathrm{rcm}}^{\mathrm{IMAU-FDM}}$ and $\max\Upsilon_{\mathrm{rcm}}^{\mathrm{MAR}}$. Here we therefore investigate whether reasons for the differences in $\max\Upsilon_{\mathrm{rcm}}$ can be found in the models' firn parameterizations. For the sake of clarity, we focus the analysis on the K-Transect, whose representativeness for the entire ice sheet will be assessed in the Discussion. Furthermore, we focus on the two contrasting melt seasons of 2012 and 2017. The former was dominated by early, persistent and intense melting, the latter by intermittent and moderate melt. They represent the end members of the last 25 mass balance years that were dominated by mass loss (see Fig. B1).

Figures 5 and 6 visualize and compare RCM simulated parameters for the 2012 and 2017 melt seasons. Figure 5 shows average or summed values over the time period 1 May to 31 October and Fig. 6 illustrates the spatio-temporal evolution of parameters over the same time frame. In 2012, IMAU-FDM shows discontinuities at the location of $\max\Upsilon_{\mathrm{rcm}}^{\mathrm{IMAU-FDM}}$: Mean

albedo increases by $\sim$0.05 (Fig. 5c) while melt drops by $\sim$400 mm w.e. or 31 % (Fig. 5e). The contrast in albedo is even higher (an increase from 0.65 to 0.78) when averaging only from mid-July to mid-August 2012. At $\max\Upsilon_{\mathrm{rcm}}^{\mathrm{IMAU-FDM}}$, runoff drops from slightly higher than 1000 mm w.e. to zero (Fig. 5e). Across $\max\Upsilon_{\mathrm{rcm}}^{\mathrm{IMAU-FDM}}$, the percentage of melt running off drops from $\sim$80 % to zero (Fig. 5g). This sudden shut-down of runoff is compensated by an abrupt increase in refreezing (Figs. 5i). In 2012 these transitions take place over the distance of a single grid cell (5.5 km), whereas in 2017, IMAU-FDM shows gradual transitions without discontinuities. In 2012, MAR shows no discontinuities in albedo and melt across $\max\Upsilon_{\mathrm{rcm}}^{\mathrm{MAR}}$ (Figs. 5c and e) but it exhibits step-wise changes in runoff and refreezing (Fig. 5g and i). These discontinuities are somewhat less pronounced than for IMAU-FDM. In 2017, simulated refreezing of MAR and IMAU-FDM are rather similar along the transect (Fig. 5k), regardless of $\max\Upsilon_{\mathrm{rcm}}^{\mathrm{MAR}}$ being located at higher elevation.

In 2012, $\Upsilon_{\mathrm{rcm}}^{\mathrm{IMAU-FDM}}$ remained stable over an extended time period (e.g. Fig. 6c). The sharp increase in total refreezing, observed in Fig. 5i, is the result of intense refreezing that took place during the prolonged time period when the IMAU-FDM runoff limit was at its maximum (Fig. 6e). The refreezing raised 10 m firn temperatures to 0 °C (Fig. 6g), which is unique for the decade 2010 to 2020 (Fig. C3). In 2012, MAR refreezing was also focused to directly above $\max\Upsilon_{\mathrm{rcm}}^{\mathrm{MAR}}$ (Fig. 6l), but not as clearly as IMAU-FDM. The peak in MAR summed refreezing is thus less pronounced (Fig. 5i). We notice that MAR refreezing fluctuates somewhat randomly along the transect. These fluctuations can be observed in both years and occur mainly in-between the $\max\Upsilon_{\mathrm{rcm}}$ of the two RCMs (Figs. 5i, k). The fluctuations can also be seen in Figs. 6l and m.

In MAR, there is less influence of refreezing on 10 m firn temperatures (Fig. 6n and o) and firn temperatures below the 2012 $\max\Upsilon_{\mathrm{rcm}}^{\mathrm{MAR}}$ were already very close to 0 °C. The relatively intense 2012 refreezing results in moderate firn warming above $\max\Upsilon_{\mathrm{rcm}}^{\mathrm{MAR}}$ which then persists (Fig. C3).

Figure 7 serves to assess whether $\max\Upsilon_{\mathrm{rcm}}$ are related to simulated firn structure. In 2012, $\max\Upsilon_{\mathrm{rcm}}^{\mathrm{IMAU-FDM}}$ coincides with the uppermost grid cell where the top 20 m of the firn consist of ice. MAR $\max\Upsilon_{\mathrm{rcm}}$ is underlain by less dense firn and is located much higher than the uppermost grid cell of uniform ice. Furthermore, we notice that the IMAU-FDM firn profile shows an ice slab, a zone of icy firn in the top $\sim$5 m of the firn profile overlying material of lower density. The slab is most pronounced directly uphill of the 2012 $\max\Upsilon_{\mathrm{rcm}}^{\mathrm{IMAU-FDM}}$. The MAR firn profile shows a more weakly developed zone of increased near-surface density around and above the 2012 $\max\Upsilon_{\mathrm{rcm}}^{\mathrm{MAR}}$.

Firn properties simulated by MAR and IMAU-FDM differ in the vicinity of the $\max\Upsilon_{\mathrm{rcm}}$ (Fig. 7), which mandates a more detailed comparison of firn properties. Along the K-Transect, KAN_U is the optimal site for such a comparison because (i) the site is located at 1840 m a.s.l. which places it above the highest $\max\Upsilon_{\mathrm{rcm}}^{\mathrm{IMAU-FDM}}$ and close to the average $\max\Upsilon_{\mathrm{rcm}}^{\mathrm{MAR}}$, and (ii) the site features repeated measurements of firn density (Rennermalm et al., 2021) and firn temperatures (e.g. Charalampidis et al., 2016; Vandecrux et al., 2023). Figure C4 visualizes simulated MAR and IMAU-FDM firn density evolution for the top 20 m over the time period 1980-2020 at KAN_U and Fig. C5 shows simulated firn temperature profiles and a comparison to measured 10 m depth firn temperatures. IMAU-FDM firn density evolution shows annual layers getting buried and an ice slab forming in summer 2012. Afterwards, the slab gets buried under accumulating snow and firn at the same rate as the annual layers. In contrast to this, the field observed depth of the top of the ice slab (Fig. C4a) remains close to the surface. The coarser vertical resolution of the MAR outputs makes it more difficult to follow horizons as they get buried. Simulated temperatures

vary strongly at the site, being close to -15 °C in IMAU-FDM and around 0 °C in MAR. The former match measured 10 m temperatures (around -11 °C; Fig. C5c) more closely.

## 4.4 $\Upsilon_{\mathrm{rcm}}$ and its relevance for RCM simulated runoff

Along the K-Transect, but also for most other regions of the ice sheet (e.g. Fig. 3), the MAR runoff zone is larger than for IMAU-FDM because $\max\Upsilon_{\mathrm{rcm}}^{\mathrm{MAR}}$ is located at higher elevations than $\max\Upsilon_{\mathrm{rcm}}^{\mathrm{IMAU-FDM}}$. The question arises to what degree this is relevant to overall runoff. On the example of the K-transect we quantify by how much total simulated runoff is influenced by differences between $\max\Upsilon_{\mathrm{rcm}}^{\mathrm{IMAU-FDM}}$ and $\max\Upsilon_{\mathrm{rcm}}^{\mathrm{MAR}}$.

For each year from 1980 to 2020 we calculate total annual RCM runoff (i) below and (ii) above $\max\Upsilon_{\mathrm{rcm}}^{\mathrm{IMAU-FDM}}$ along the
305 K-transect (see the inset in Fig. 8). The first value, termed $\int_{\Downarrow} Q$, can be calculated from both RCMs. The second value, $\int_{\Uparrow} Q$, can only be calculated from MAR whose $\max\Upsilon_{\mathrm{rcm}}$ is always higher than IMAU-FDM along the K-transect.

Exponential regression of the two parameters $\int_{\Downarrow} Q_{\mathrm{MAR}}$ and $\int_{\Uparrow} Q_{\mathrm{MAR}}$ yields $R^2 = 0.83$ (Fig. 8), which means the amount of MAR runoff above $\max\Upsilon_{\mathrm{rcm}}^{\mathrm{IMAU-FDM}}$ increases exponentially as a function of the MAR runoff below $\max\Upsilon_{\mathrm{rcm}}^{\mathrm{IMAU-FDM}}$. If MAR and IMAU total runoff below $\max\Upsilon_{\mathrm{rcm}}^{\mathrm{IMAU-FDM}}$ limit were similar (see the following paragraph), this implies that
the difference in simulated runoff between MAR and IMAU-FDM increases in high-melt seasons. The reason for the disproportional growth is that the more intense the melt season, the further apart the two $\max\Upsilon_{\mathrm{rcm}}$. If $\int_{\Uparrow} Q_{\mathrm{MAR}}$ is expressed as a percentage of $\int_{\Downarrow} Q_{\mathrm{MAR}}$, we find that for 2012 $\int_{\Uparrow} Q_{\mathrm{MAR}}$ corresponds to 20 % of $\int_{\Downarrow} Q_{\mathrm{MAR}}$. For the year 2017, the percentage is 3.2 % which is somewhat lower than the mean of all years (5.7 %).

The above statistics are based on $Q_{\mathrm{MAR}}$ alone and the question arises how relevant $\int_{\Uparrow} Q_{\mathrm{MAR}}$ is, given that simulated runoff
of the two RCMs *below* $\max\Upsilon_{\mathrm{rcm}}^{\mathrm{IMAU-FDM}}$ are not identical. We label the area below $\max\Upsilon_{\mathrm{rcm}}^{\mathrm{IMAU-FDM}}$ as the "common runoff area" and we find that over this area, MAR simulates 5.3±7.1 % (mean±1 std. dev.) more runoff than IMAU-FDM. This means that along the K-transect, and during normal melt seasons, the differences in RCM runoff caused by the diverging $\max\Upsilon_{\mathrm{rcm}}$ are similar to the differences in runoff over the common runoff area. In extraordinary melt seasons such as 2012 and 2019, however, the influence of the differing $\max\Upsilon_{\mathrm{rcm}}$ clearly exceeds the differences in RCM runoff over the common runoff area.
In 2012, total MAR runoff along the K-transect exceeds IMAU-FDM by 29 %, out of which three quarters are due to MAR runoff above $\max\Upsilon_{\mathrm{rcm}}^{\mathrm{IMAU-FDM}}$; in 2019, the difference in total runoff is 16 % out of which almost four fifths are due to MAR runoff from above $\max\Upsilon_{\mathrm{rcm}}^{\mathrm{IMAU-FDM}}$.

## 5 Discussion

There are fundamental differences between runoff processes detected from remote sensing and their simulation. Optical satellite
imagery primarily detects *lateral* runoff, visible in slush fields and meltwater streams at the surface; sub-surface runoff cannot be sensed. In contrast, current state-of-the-art dedicated firn models or RCM firn modules simulate runoff through *vertical* percolation alone; lateral flow is not simulated. Nevertheless, we have compared modelled and remotely sensed runoff limits on the Greenland Ice Sheet because (i) modelled runoff has the purpose of mimicking the actual, strongly lateral, process. Thus

we here tested whether the mimicking approximates the effects of the actual hydrological processes. (ii) The remotely sensed visible runoff limit approximates the actual (invisible) runoff limit reasonably well at the peak of the melt season (Holmes, 1955; Clerx et al., 2022; Tedstone and Machguth, 2022). (iii) We observe the most remarkable differences not between $\Upsilon_{\text{obs}}$ and $\Upsilon_{\text{rcm}}$, but between the two $\Upsilon_{\text{rcm}}$.

## 5.1 Comparing MODIS and simulated runoff limits

We observe a relationship between $\max\Upsilon_{\text{obs}}$ and $\max\Upsilon_{\text{rcm}}$ that is in broad agreement to Ryan et al. (2019) who compared *snow lines* simulated by MAR, RACMO and observed from remote sensing (cf. Fig. 4 herein and Fig. 5 in Ryan et al., 2019). Runoff limits and snow lines simulated by MAR are often high, but differences between melt seasons are in qualitative agreement with MODIS observations. On average, $\max\Upsilon_{\text{rcm}}^{\text{IMAU}-\text{FDM}}$, as well as RACMO snow lines, fall below MODIS and variability from year to year appears suppressed.

We find that $\max\Upsilon_{\text{rcm}}$ in RACMO 1 km are somewhat higher than for IMAU-FDM, which could be an effect of downscaling and bias correction. RACMO 1 km exhibits the same reduced temporal variability as $\max\Upsilon_{\text{rcm}}^{\text{IMAU}-\text{FDM}}$. RACMO's firn module and IMAU-FDM are very similar, apart from the coarser vertical resolution of the former, and for the remainder of the discussion, we focus on IMAU-FDM to establish the main causes for the differences in $\max\Upsilon_{\text{rcm}}$ between MAR and the RACMO family of models.

At the scale of individual melt seasons, daily MAR data shows strong drops in $\Upsilon_{\text{rcm}}$ during cold spells (Fig. 2). IMAU-FDM shows only moderate drops but the smoother curve is due to the coarser 10-day temporal resolution of the data. Sudden drops are not present in MODIS $\Upsilon_{\text{obs}}$ because the actual routing of meltwater is a much slower process than the instantaneous vertical routing in bucket schemes. In slush fields and streams water can flow along the surface for tens of kilometers (Holmes, 1955; Poinar et al., 2015; Yang et al., 2021), at speeds of a few meters per hours in slush (Clerx et al., 2022) or a few kilometers per hour in surface streams (Gleason et al., 2016). Holmes (1955) observed that it took about two weeks after the end of melting before streams ran dry and froze over.

## 5.2 Why do simulated runoff limits differ?

The very substantial differences between runoff limits simulated by MAR and IMAU-FDM (e.g. Figs. 3 and 4) could be caused by (i) differences in RCM simulated accumulation or melt, or (ii) differences in the parameterizations of firn and firn hydrology. On the example of the K-transect we have shown that RCM simulated accumulation and melt (Fig. B1) are generally similar. However, $\max\Upsilon_{\text{rcm}}^{\text{IMAU}-\text{FDM}}$ are situated at lower elevations than $\max\Upsilon_{\text{rcm}}^{\text{MAR}}$ and because of their lower elevations, melt at $\max\Upsilon_{\text{rcm}}^{\text{IMAU}-\text{FDM}}$ is substantially larger than at $\max\Upsilon_{\text{rcm}}^{\text{MAR}}$ (Appendix B). Because the condition applies that for all $\max\Upsilon_{\text{rcm}}$ there can be no runoff directly above the runoff limit, IMAU-FDM simulated refreezing at $\max\Upsilon_{\text{rcm}}^{\text{IMAU}-\text{FDM}}$ is substantially larger than MAR refreezing at $\max\Upsilon_{\text{rcm}}^{\text{MAR}}$. Hence, we argue that differences in the models' parameterizations of firn and firn hydrology are mainly responsible for the differences between their runoff limits.

IMAU-FDM's large refreezing potential is the main reason for its low runoff limits. The refreezing potential is large due to (i) the relatively low firn temperatures, (ii) the relatively high irreducible water saturation at higher firn densities, and (iii)

the thick firn layer (up to 100 m) which offers ample amounts of firn air content in which meltwater can refreeze. The large refreezing potential, and IMAU-FDM's condition that runoff can only occur at the bottom of the firn pack, is also responsible for the runoff limit being relatively immobile. Before $\max \Upsilon_{\mathrm{rcm}}^{\mathrm{IMAU-FDM}}$ can propagate to higher elevations, the pore space of the thick firn pack needs to be filled. Consequently, $\max \Upsilon_{\mathrm{rcm}}^{\mathrm{IMAU-FDM}}$ migrates uphill only slowly. However, once a grid cell's firn has lost its pore space, this grid cell will nearly always remain runoff area, even during weak melt years: apart from the pore space in the seasonal snow, there is no more possibility to store meltwater. This explains (i) why in IMAU-FDM the uppermost elevation of fully icy firn roughly coincides with $\max \Upsilon_{\mathrm{rcm}}^{\mathrm{IMAU-FDM}}$ (Fig. 7), (ii) why in moderate melt years $\max \Upsilon_{\mathrm{rcm}}^{\mathrm{IMAU-FDM}}$ does not drop substantially below the elevation of fully icy firn, and (iii) why high-elevation melt in extreme melt years cannot run off and instead percolates to depth and refreezes, as indicated by the strong firn warming in 2012 (Fig. C3).

RACMO's surface albedo parameterization further contributes to immobilizing $\max \Upsilon_{\mathrm{rcm}}^{\mathrm{IMAU-FDM}}$. During intense melt seasons, RACMO shows a pronounced step change in surface albedo that coincides with $\max \Upsilon_{\mathrm{rcm}}^{\mathrm{IMAU-FDM}}$ (see Fig. 5c for the situation in the summer of 2012). The higher albedo above that step change reduces melt and also the likelihood of percolation to the bottom of the firn where runoff could take place. Furthermore, reduced melt above $\max \Upsilon_{\mathrm{rcm}}^{\mathrm{IMAU-FDM}}$ reduces the amount of water available for refreezing which slows down the loss of firn pore space. The albedo step change is caused by RACMO's ELA (defined here as the elevation where the climatic mass balance, see Cogley et al., 2011, equals zero) coinciding with $\max \Upsilon_{\mathrm{rcm}}^{\mathrm{IMAU-FDM}}$, a situation which occurred every fourth melt season during the time period 1990-2020. Below the ELA, RACMO albedo is prescribed based on MODIS imagery (see Section 3.2.2). Above the ELA, the albedo is calculated based on snow albedo parameterizations independent of MODIS data. MAR does not show discontinuities in albedo, also not in 2012 (Fig. 5c) which is the only melt season where MAR's ELA coincides with $\max \Upsilon_{\mathrm{rcm}}^{\mathrm{MAR}}$. It appears that MAR's albedo parameterization, which does not use remote sensing data, allows for a more smooth transitions of surface albedo across $\max \Upsilon_{\mathrm{rcm}}^{\mathrm{MAR}}$.

MAR's firn temperatures are warmer than IMAU-FDM (Fig. C5), the irreducible water saturation below 1 m depth is smaller than in IMAU-FDM and the simulated firn pack is more shallow reaching only to 20 m depth. This means that MAR's refreezing potential is smaller and allows for stronger fluctuations in $\max \Upsilon_{\mathrm{rcm}}^{\mathrm{MAR}}$, as compared to IMAU-FDM. Runoff in MAR occurs also from areas of porous firn (Fig. 7), which is not possible in IMAU-FDM. The reason is MAR's parameterization which states that $1/3$ of meltwater reaching an ice lens runs off immediately while the remaining $2/3$ are routed further to depth. This parameterization mimics lateral runoff of meltwater on top of low-permeability ice slabs (MacFerrin et al., 2019) and allows $\max \Upsilon_{\mathrm{rcm}}^{\mathrm{MAR}}$ to fluctuate in-between the elevation of depleted firn pore space (where similarly to IMAU-FDM pore space exists only in the seasonal snow) and the highest elevation where ice layers are simulated in the otherwise porous firn.

It remains unclear why MAR firn temperatures are warmer and show a less smooth spatial distribution than RACMO (e.g. Fig. C3). MAR's irregular spatial pattern could be partially caused by the coarser vertical resolution of MAR's firn and the dynamic vertical discretisation where adjacent layers of similar properties are merged in depth to keep a higher number of layers available to represent the first meter of snow. It can occur that individual MAR pixels have only one layer of $\sim$19 m in thickness situated below 19 thin layers resolving the first meter of the snowpack. As a result, in some pixels the 10 m depth temperature refers to the temperature of a layer covering a large depth interval, for other pixels to a much thinner layer close to

10 m depth. In IMAU-FDM, the firn is much finer resolved and a comparison to measured firn temperatures at a certain depth (Fig. C5) always compares to a thin model layer very close to the same depth. An alternative explanation for the colder IMAU-FDM firn temperatures would be that the Figures 6, C3 and C5 give a wrong impression because latent heat in IMAU-FDM is released at depths greater than the max. 20 m shown in the figures. If this were the case, then IMAU-FDM depth-integrated firn temperatures would be warmer than shown due to warm firn below the visualized depths. However, this is not the case: During the strongest melt season of 2012, IMAU-FDM meltwater percolation reached a maximum of $\sim$15 m depth directly above $\max\Upsilon_{\mathrm{rcm}}^{\mathrm{IMAU-FDM}}$ and $\sim$5 m at KAN_U. IMAU-FDM's relatively high irreducible water saturation hinders deep percolation.

## 5.3 Simulated runoff limits influence total runoff

We find that in intense melt years, MAR simulates up to 29 % more runoff than IMAU-FDM along the K-Transect. This difference is mainly due to MAR runoff from above $\max\Upsilon_{\mathrm{rcm}}^{\mathrm{IMAU-FDM}}$. All $\max\Upsilon_{\mathrm{rcm}}^{\mathrm{MAR}}$ are located further inland than $\max\Upsilon_{\mathrm{rcm}}^{\mathrm{IMAU-FDM}}$ and in the year 2012, the distance between the two runoff limits reaches $\sim$75 km (Fig. 5a). While MAR runoff between the two runoff limits is on average modest and small compared to runoff over the RCM's common runoff area (Fig. 5e), the considerable distance causes total MAR runoff from above $\max\Upsilon_{\mathrm{rcm}}^{\mathrm{IMAU-FDM}}$ to become relatively large. In melt seasons of intermediate intensity, MAR and IMAU-FDM $\max\Upsilon_{\mathrm{rcm}}$ are located closer to each other (Fig. 5b) and total runoff between them is relatively small.

Although 2012 and 2019 appear as outliers when compared to most other melt seasons, the trend towards larger differences in strong melt seasons is a logical consequence of $\max\Upsilon_{\mathrm{rcm}}^{\mathrm{IMAU-FDM}}$ varying weakly with melt intensity while $\max\Upsilon_{\mathrm{rcm}}^{\mathrm{MAR}}$ fluctuates strongly. This causes differences in runoff to grow as a result of increasing melt season intensity. The ice sheet hypsometry contributes to this effect. As the ice sheet surface becomes increasingly flatter towards higher elevations, elevation differences between the two $\max\Upsilon_{\mathrm{rcm}}$ translate into large horizontal offsets. In strong melt seasons, $\max\Upsilon_{\mathrm{rcm}}^{\mathrm{MAR}}$ are located at elevations where the surface slope is shallow and horizontal distance between the two $\max\Upsilon_{\mathrm{rcm}}$ is large (e.g. Figs. 5a and b). The runoff area simulated by MAR therefore grows substantially, unlike IMAU-FDM, whose runoff limit, and consequently also runoff area, is insensitive to strong melting.

## 5.4 Implications

Our analysis focuses on the K-transect, which is located where differences in $\max\Upsilon_{\mathrm{rcm}}$ are at their maximum (Fig. 3). However, other studies indicate our findings are valid elsewhere on the ice sheet. Spatial discontinuities in MAR firn temperatures, for example, were already shown to exist Greenland-wide by Vandecrux et al. (2023). Tedstone and Machguth (2022) focused on firn areas that experience surface runoff and found that 1985-2020 MAR and RACMO simulated cumulative runoff above a certain reference elevation differ by a factor of two. Given the relationship shown in Fig. 8 and our explanation why the difference between the two $\max\Upsilon_{\mathrm{rcm}}$ increases with melt season intensity, one expects runoff limits to diverge further in a warmer future climate. Indeed, Glaude et al. (2024) show that by the year 2100, under identical SSP5-8.5 high emissions forcing, the runoff limits of RACMO and MAR differ strongly over most of the ice sheet. The consequence is a twofold larger simulated annual surface mass loss in MAR than in RACMO (Glaude et al., 2024).

Uncertainty in future Greenland surface mass balance will grow with continued warming, and uncertainties in simulating Greenland's firn area contribute strongly to overall uncertainty. As both models demonstrate strengths and weaknesses in reproducing MODIS $\Upsilon_{obs}$ and $\max\Upsilon_{obs}$, it is unknown which simulates total runoff more accurately. Nevertheless, combining the strengths of the models might be a first step to improve the simulation of the surface mass balance of Greenland's firn area.

RACMO, and consequently also IMAU-FDM, might benefit from a revised bare-ice albedo parameterization. The existing parameterization leads to step-like changes in albedo at the runoff limit during intense melt seasons.

IMAU-FDM simulates a finely resolved and deep firn column, but this leads to a relatively immobile runoff limit when combined with a standard bucket scheme where runoff can take place only at the base of the firn. While RACMO has much fewer firn layers, the runoff limit is similarly immobile (Figs. 3 and 4) because RACMO uses a very similar bucket scheme with the same parameterization of irreducible liquid water saturation and a similarly deep firn column. In a first step, IMAU-FDM and RACMO could include a parameterization that mimics lateral runoff whenever percolating water encounters an ice layer, akin to the parameterization included in MAR.

MAR might benefit from simulating a deeper and more finely resolved firn column. The current coarse resolution and the merging of layers impede comparisons to measurements and challenge assessment of model performance.

MAR includes a parameterization mimicking the effect of ice slabs on runoff. However, The comparison to MODIS $\max\Upsilon_{obs}$ indicates that MAR's fluctuations of $\max\Upsilon_{rcm}^{MAR}$ are too large and strongly exceed $\max\Upsilon_{obs}$ in intense melt seasons. The strong fluctuations are mainly caused by the parameterization for ice slab runoff, which indicates the need for calibration. The minimum thickness, required for an ice layer to trigger runoff, could be set based on Jullien et al. (2025) who provide first empirical evidence for the minimum ice slab thickness supporting lateral runoff. Altering the runoff ratio from $1/3$ to another value would not directly influence $\max\Upsilon_{rcm}^{MAR}$, but controls how much water percolates to depth and thus also influences refreezing and firn structure, such as the formation or thickening of ice layers.

Beyond these initial, albeit not trivial modifications, the models could replace the bucket scheme with more physical simulations of snow and firn as applied by Wever et al. (2014, 2016); Langen et al. (2017); Vandecrux et al. (2020). Besides the inclusion of preferential percolation, these approaches also allow for temporary storage of meltwater in snow and firn, which plays an important role in shaping firn structure. Observations since 2012 at the KAN_U site show that, unlike the IMAU-FDM simulation, the ice slab is not getting buried. Instead, the depth of its surface remained roughly constant (Fig. C4a). The ice slabs are of low permeability which causes meltwater to pond in slush at their surface (Clerx et al., 2022) and to refreeze partially, over the course of a melt season, as superimposed ice (Tedstone et al., 2025). This mechanism, by which ice slabs mainly thicken, is absent in an instantaneous bucket scheme. Both RCMs currently do not permit slush formation and even thick ice layers must remain "permeable" for meltwater to be routed vertically. Removing these constraints by adopting more physical firn simulations might improve the models' representation of melting firn. However, this potential can only be tapped if sufficient empirical data exist to calibrate and evaluate firn parameterizations. So far few studies have focused on measuring the processes and changes in Greenland's melting firn area.

# 6 Conclusions

We developed a flexible method to detect visible runoff limits from MODIS and compared the results to modelled runoff limits from IMAU-FDM and MAR. We found large differences not only between remotely sensed and modelled data, but also between the two models. IMAU-FDM simulated runoff limits are on average somewhat lower than MODIS, and variability from year to year is strongly reduced. On average, MAR simulates substantially higher runoff limits than MODIS, but the magnitude of yearly fluctuations of MAR's runoff limits are similar to MODIS, except for some areas where the inter-annual variability exceeds MODIS. Both MAR and IMAU-FDM use a bucket scheme that routes water vertically through the firn in an attempt to mimic the strongly lateral water flux of the actual firn hydrology. Differences in the implementation of the bucket schemes are the main reasons for the deviations between MAR and IMAU-FDM runoff limits: (i) in MAR a fraction of the meltwater runs off when it encounters an ice layer inside the firn, (ii) the amount of pore space and cold content varies between the two models because they simulate different firn depths (iii) IMAU-FDM allows for a higher irreducible water saturation, and (iv) the firn layer in MAR is warmer near the runoff limit which promotes runoff.

We compare total simulated RCM runoff along the K-transect and we find that MAR total runoff exceeds IMAU-FDM by up to 29 %. We show that in strong melt seasons MAR and IMAU-FDM runoff limits are separated by large horizontal distances, which is the main reason for the difference in total runoff. Any differences in ablation area runoff, simulated by the two RCMs, are eclipsed by the amount of runoff that MAR simulates, in strong melt years, above the IMAU-FDM runoff limit. Ice sheet hypsometry is partially responsible for the large horizontal distance between the two runoff limits: the ice sheet surface slope becomes increasingly shallow with altitude and relatively small differences in the elevation of the runoff limits translate into large horizontal distances.

Increased melting is anticipated for the future. This means the situation where the two models diverge the most will become more frequent, simulated runoff will further diverge and uncertainty grow. We conclude that a reliable simulation of the surface mass balance in a melting firn zone is key to faithfully anticipate Greenland's future surface mass balance. Newly formed runoff areas will play a major role in Greenland's future mass balance. Understanding of the physical processes in firn, firn hydrology and superimposed ice formation is essential to improve model performance.

*Code and data availability.* The code and most data used in this manuscript are available at https://zenodo.org/doi/10.5281/zenodo.13332326. The RCM data are too volumous for the repository and can be obtained directly from the authors.

*Author contributions.* HM designed the study, wrote most of the code and carried out most of the analysis. AT contributed to study design, analysis and provided code as well as input data. The study was written by HM with contributions by AT, BN, MB, MvdB, PKM and XF (in alphabetic order). BN, MB, MvdB, PKM and XF provided the MAR, RACMO and IMAU-FDM data. All authors discussed the results and commented on the text.

*Competing interests.* At least one of the (co)authors is an editor at the Cryosphere. The authors declare that they have no other competing interests.

*Acknowledgements.* This study is funded by the European Research Council (ERC) under the European Union's Horizon 2020 research and innovation programme (project acronym CASSANDRA, grant agreement No. 818994). MB, PKM and MvdB acknowledge support from the Netherlands Earth System Science Centre (NESSC). BN is a Research Associate of the Fonds de la Recherche Scientifique de Belgique– F.R.S.-FNRS.

## Appendix A: Calculation of $\Upsilon_{\mathrm{obs}}$ along flowlines

### A1 Calculation along flowlines

We create polygons by (i) calculating actual flowlines which are (ii) buffered by $p_w/2$. This approach creates polygons of arbitrary shape and direction, here termed flowline-polygons. Even in complex topography, the direction of the flowline-polygons is always roughly perpendicular to the surface slope.

We chose to calculate flowlines based on surface velocity fields rather than surface slope (cf. Machguth and Huss, 2014).
The advantage is a straightforward algorithm, as described in the following. We calculate flowlines following Fig. 3 in Cabral and Leedom (1993), using Greenland ice sheet surface velocity fields in x and y direction. Our algorithm starts at seed-points and then progresses downhill from gridcell to gridcell. A flowline enters a cell at a certain point along its margins and based on entry point, flow direction within the cell and location of the cell margins, the algorithm then calculates the point where the flowline leaves the cell and enters the following cell. A flowline ends when it reaches the ice sheet margin.
There are cases where flow directions of neighboring cells are conflicting and the algorithm would send the flowline immediately back to the cell where it came from. Such conflicts are solved by calculating the average flow direction of the two grid cells in question. The flow line then continues in average flow direction through one of the two cells.

Seed-points are created by first drawing a polygon that follows roughly the 2400 m a.s.l. elevation contour in the south of the ice sheet and descends towards the 1800 m a.s.l. contour in the north. Along the polygon, seed-points are created automatically
every 15 km. Eventually, all flowlines are buffered by $p_w/2 = 10$ km to create flowline-polygons. Given the width of the flowline-polygons ($p_w = 20$ km) and 15 km spacing of the seed points, a certain overlap of the polygons occurs and is wanted (Fig. A1). More closely spaced polygons provide a higher spatial resolution of $\Upsilon_{\mathrm{obs}}$ and make it easier to detect outliers. On outlet glaciers polygons overlap due to confluence (Fig. A1). There are also cases where polygons overlap for most of their length due to a combination of specific flow patterns and location of the seed points. The polygons were sifted manually to
remove such polygons. The result is a set of 510 flowline-polygons (see Fig. 1).

## A2 Accounting for background spatial variability of albedo

Our algorithm uses daily MODIS MOD10A1 albedo maps to assess spatial variability of albedo $\sigma_\alpha$. MODIS records changes in $\alpha$ and $\sigma_\alpha$ as surface characteristics and hydrology evolve over the duration of a melt season. However, the satellite images also capture pattern in $\alpha$ that are persistent in space and time. Such persistent albedo features typically originate from topographic undulations or rock outcrops. Where persistent albedo features are frequent, they impact $\sigma_\alpha$ and interfere with detecting $\Upsilon_{\mathrm{obs}}$. The original approach by Machguth et al. (2023) did not include any correction for the potential impact of persistent albedo features on $\Upsilon_{\mathrm{obs}}$. The updated approach used here now includes a simple correction as described in the following.

We calculate a Greenland-wide map of background $\sigma_\alpha$, based on daily arrays of $\sigma_\alpha$ from before the start of the melt season. (i) From each spring of the 22 years 2000 to 2021, 20 daily arrays of $\sigma_\alpha$ are selected. (ii) We then calculate grid cell values of an initial background $\sigma_\alpha$ array as the median of up to 440 (22 years $\times$ 20 days) daily values (the actual number of data points is smaller due to frequent clouds or data issues). The large north-south extent of Greenland requires to vary the 20-day time-window across latitudes. Up to $\sim$75.5 °N the time window are the days of year (DoY) 110 – 130, between $\sim$75.5 °N and $\sim$80 °N DoY 120 – 140, and north of $\sim$80 °N DoY 130 – 150. (iii) The final array of background $\sigma_\alpha$ is calculated by subtracting the mean of all grid cells, calculated from the initial background $\sigma_\alpha$ array, from each grid cell. Any resulting negative values are replaced by zero.

In detecting daily $\Upsilon_{\mathrm{obs}}$, the final array of background $\sigma_\alpha$ is subtracted from every daily array of $\sigma_\alpha$. The thresholds for $\sigma_\alpha$, used in the original algorithm by Machguth et al. (2023), remain unchanged as the background $\sigma_\alpha$ array consists mostly (82 %) of zeros.

## A3 Modified filtering for outliers

Candidates for $\Upsilon_{\mathrm{obs}}$ require filtering to remove false positives (Machguth et al., 2023). We apply the same automated approach in two stages but the filtering of the last valid candidates has been simplified (Section 4.4 in Machguth et al., 2023). If a suspicious last candidate is detected, then the updated algorithm searches for valid detections within a time window of $\pm6$ days and a circle of 75 km. The suspicious candidate is labeled invalid if it exceeds the median elevation of all nearby valid detections by >75 m. If the number of nearby valid detections is too small to calculate a median, the suspicious candidate is labeled 'valid'. The number of removed candidates remains similar under the updated filter algorithm, but there is no more risk of consulting distant $\Upsilon_{\mathrm{obs}}$ when evaluating reliability of candidates.

## A4 Comparison to Landsat-derived visible runoff limits

We compared MODIS $\Upsilon_{\mathrm{obs}}$ to annual maxima of Landsat visible runoff limits $RL$, using annual maximum $RL$ at 1 km posting (see methods in Tedstone and Machguth, 2022). We first iterated through each flowline polygon, identifying all the Landsat $RL$ which fall inside it, then generated median Landsat $RL$ for all data in that polygon on a particular day. We only compare MODIS and Landsat on days when retrievals were made by both approaches and comparisons were only done for those flowline

polygons located in areas for which Tedstone and Machguth (2022) applied their Landsat algorithm. Among smaller excluded areas on the west coast and in the north, no comparison was possible for the entire east coast south of $\sim 76\,°\text{N}$.

The comparison is shown in Fig. A2 and yields a linear regression that falls very close to the line of identity. The bias between the two datasets is small, on average MODIS $\Upsilon_{\text{obs}}$ falls 26 m below Landsat $RL$. The comparison yields $R^2 = 0.81$, which is somewhat lower than the $R^2 = 0.87$ of the evaluation of the Machguth et al. (2023) algorithm against Landsat visible runoff limits. However, the comparison in Machguth et al. (2023) was restricted to the west coast which is the area where MODIS and Landsat visible runoff limits are most reliable. Furthermore, the comparison shown in Fig. A2 focuses on Landsat annual maximum $RL$ while Machguth et al. (2023) used all individual Landsat visible runoff limit retrievals followed by detection and removal of likely erroneous Landsat visible runoff limits. Here we do not apply any cleaning to the Landsat $RL$.

Qualitatively, we conclude that the improved MODIS algorithm compares similarly to Landsat $RL$ as did the original MODIS algorithm by Machguth et al. (2023). The latter, however, was restricted in its applicability to the western flank of the ice sheet. We find the largest deviations between the improved MODIS algorithm and Landsat at the north-eastern flank of the ice sheet. For example, the point cloud located below the line of identity at $\Upsilon_{\text{obs}} \approx 850\,\text{m a.s.l.}$ (see Fig. A2) concerns MODIS and Landsat retrievals from the vicinity of flowline NE (Fig. 1).

## Appendix B: Differences in accumulation and melt near and at $\max\Upsilon_{\text{rcm}}^{\text{IMAU}-\text{FDM}}$ and $\max\Upsilon_{\text{rcm}}^{\text{MAR}}$

We explore differences in melt $M_{\text{rcm}}$ and accumulation $C_{\text{rcm}}$ at $\max\Upsilon_{\text{rcm}}$ and investigate whether they could explain the differences in modelled runoff limits. For clarity, we focus the analysis on the K-Transect. First we compare annual accumulation sums in RACMO ($C_{\text{RACMO}}$) and MAR ($C_{\text{MAR}}$). Thereby we sum up $C_{\text{rcm}}$ over one year (1 September to 31 August) and average over a zone that encompasses all annual $\max\Upsilon_{\text{rcm}}$ of IMAU-FDM and MAR. We focus on this zone rather than the entire K-Transect as we want to examine RCM differences close to the $\max\Upsilon_{\text{rcm}}$. We observe a high correlation of annual accumulation simulated by the two RCMs ($C_{\text{RACMO}} = 0.09 + 0.93 C_{\text{MAR}}$; $R^2 = 0.92$, $p < 0.001$). Average $C_{\text{RACMO}}$ (0.44±0.08 m w.e.) exceeds average $C_{\text{MAR}}$ (0.37±0.09 m w.e.). Next we regress annual $\max\Upsilon_{\text{rcm}}^{\text{MAR}}$ vs. $C_{\text{MAR}}$ and $\max\Upsilon_{\text{rcm}}^{\text{IMAU}-\text{FDM}}$ vs. $C_{\text{RACMO}}$. Both regressions do not yield statistically significant relationships, indicating that differences in $C_{\text{rcm}}$ cannot explain the differences between the models' runoff limits.

Second, we compare melt for the same zone and summed up over the melt season, defined as 1 June to 31 August. We find that $M_{\text{RACMO}}$ and $M_{\text{MAR}}$ are highly correlated but RACMO melt is biased low in comparison to MAR (Fig. B1a). However, the bias is small or close to zero for moderate and low melt seasons, respectively. The differences in $M_{\text{rcm}}$ might be explained by RACMO having on average a higher surface albedo (0.79±0.02) as MAR (0.77±0.02). Regressing annual $\max\Upsilon_{\text{rcm}}$ against $M_{\text{rcm}}$ reveals a stark contrast between the two RCMs (Fig. B1b). For a given amount of melt, $\max\Upsilon_{\text{rcm}}^{\text{MAR}}$ is up to $\sim 450\,\text{m}$ higher than $\max\Upsilon_{\text{rcm}}^{\text{IMAU}-\text{FDM}}$. The latter shows a weak dependency on $M_{\text{RACMO}}$ while $\max\Upsilon_{\text{rcm}}^{\text{MAR}}$ depends more strongly on $M_{\text{MAR}}$. Differences between $M_{\text{RACMO}}$ and $M_{\text{MAR}}$ apparently cannot explain the large differences in $\Upsilon_{\text{rcm}}$ either.

Third, we compare $C_{\text{rcm}}$ and $M_{\text{rcm}}$ simulated at the RCM grid cells that coincide with each annual $\Upsilon_{\text{rcm}}$. We find rather similar average $C_{\text{RACMO}}$ at $\max\Upsilon_{\text{rcm}}^{\text{IMAU}-\text{FDM}}$ (0.40±0.07 m w.e.) and $C_{\text{MAR}}$ at $\max\Upsilon_{\text{rcm}}^{\text{MAR}}$ (0.37±0.09 m w.e.). Average

$M_{\mathrm{rcm}}$ at $\max \Upsilon_{\mathrm{rcm}}$ is higher in RACMO ($0.59\pm0.21$ m w.e.) than in MAR ($0.34\pm0.12$ m w.e.). This is consistent with the above established low bias of $M_{\mathrm{RACMO}}$ because the $\max \Upsilon_{\mathrm{rcm}}^{\mathrm{IMAU-FDM}}$ are located at substantially lower elevations where melt is higher. The comparison of $C_{\mathrm{rcm}}$ and $M_{\mathrm{rcm}}$ at annual $\max \Upsilon_{\mathrm{rcm}}$ reveals an important difference between the models: in IMAU-FDM, the runoff limit is typically located where summer melt exceeds annual accumulation ($C_{\mathrm{RACMO}} - M_{\mathrm{RACMO}} = -0.19 \pm 0.25$ m w.e.); in MAR melt and accumulation at $\max \Upsilon_{\mathrm{rcm}}^{\mathrm{MAR}}$ are similar ($C_{\mathrm{MAR}} - M_{\mathrm{MAR}} = 0.03 \pm 0.14$ m w.e.).

**Appendix C: Additional Figures**

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

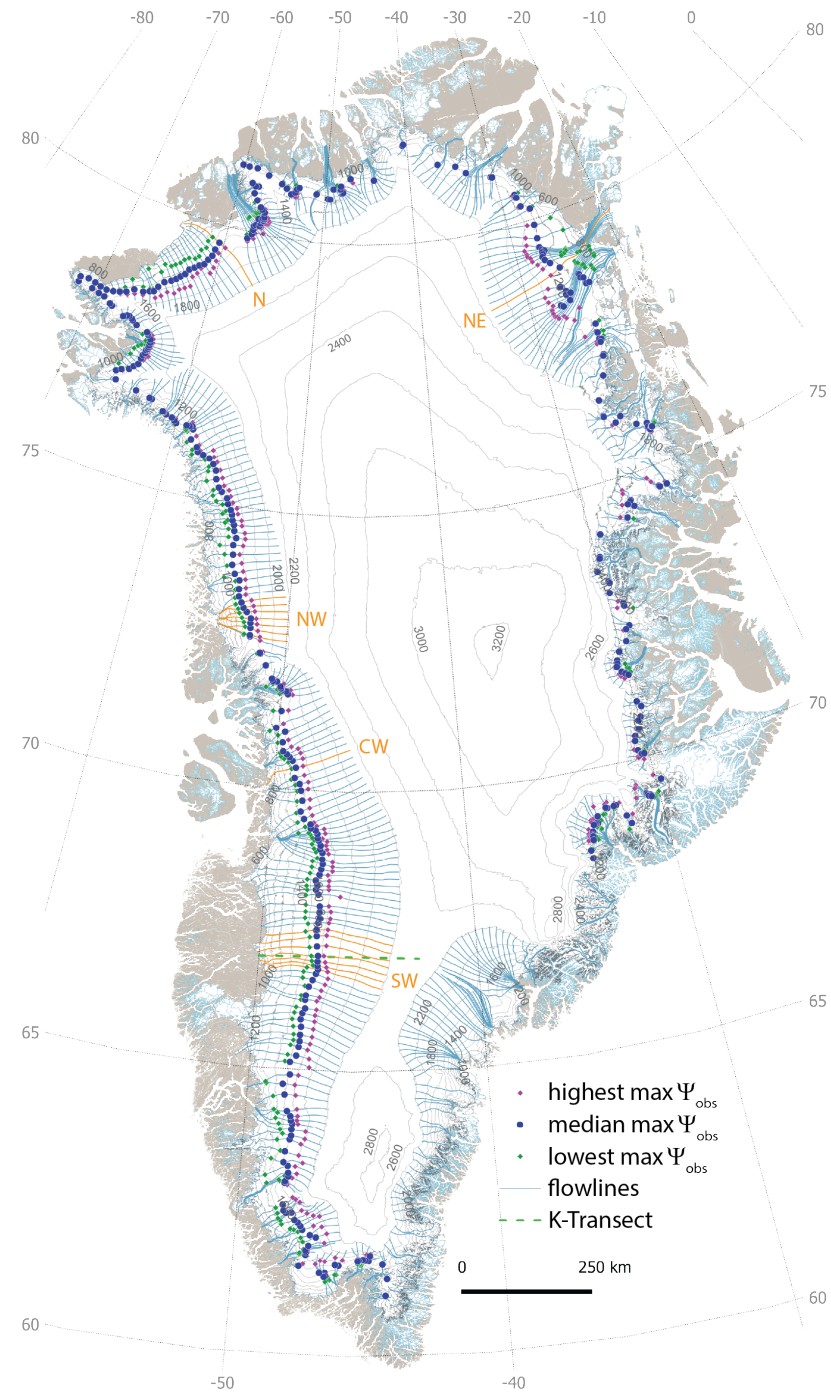

**Figure 1.** Median, highest and lowest of all annual MODIS $\max\Upsilon_{obs}$ for the time period 2000 to 2021. Retrievals at the east coast between $60\,^\circ$ to $68.4\,^\circ N$, where the hydrological regime is dominated by firn aquifers, have been masked. Flowlines highlighted in orange indicate the locations for which detailed results are shown in Figs. 2, 4, C1 and C2. The location of the K-transect (Figs. 5, 6 and C3) is indicated as well.

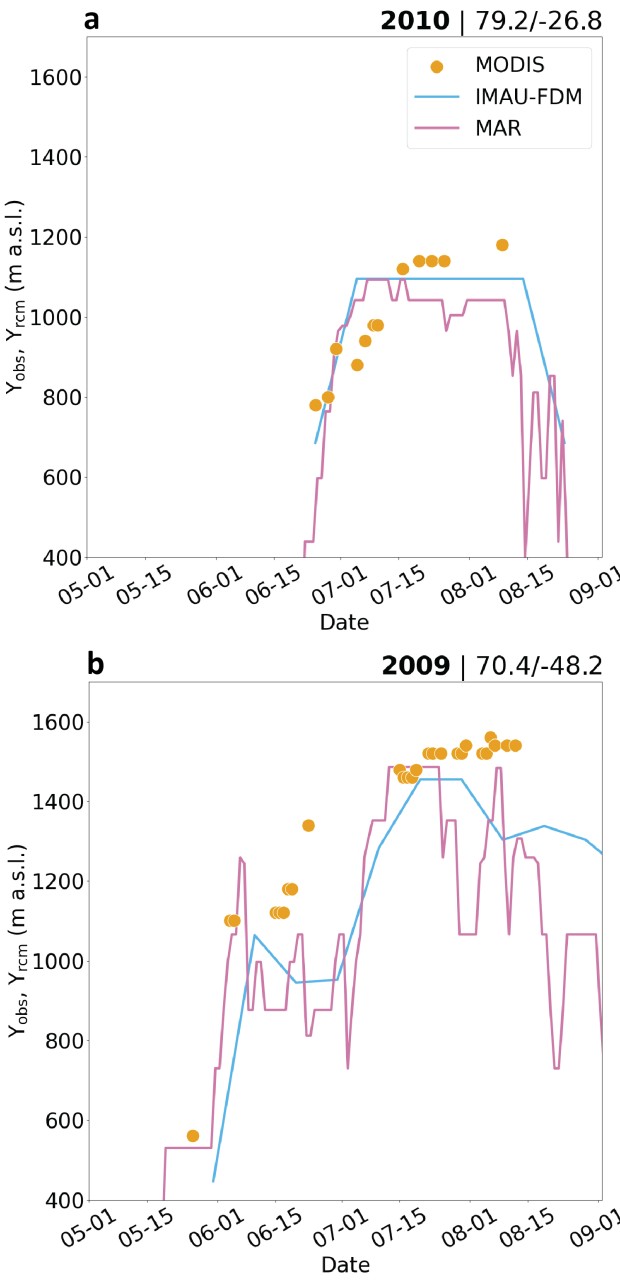

**Figure 2.** Evolution of MODIS viible runoff limits $\Upsilon_{\rm obs}$ and RCM simulate runoff limit $\Upsilon_{\rm rcm}$ over two selected melt seasons and flowlines. Solid lines show RCM runoff limits at daily resolution for MAR and in 10-day steps for IMAU-FDM. Subplot **a** shows data for the transect NE for the year 2010, **b** shows the transect CW for the year 2009. See Fig. 1 for the location of the two transects shown. Coordinates are provided to indicate the approximate location of the two transects.

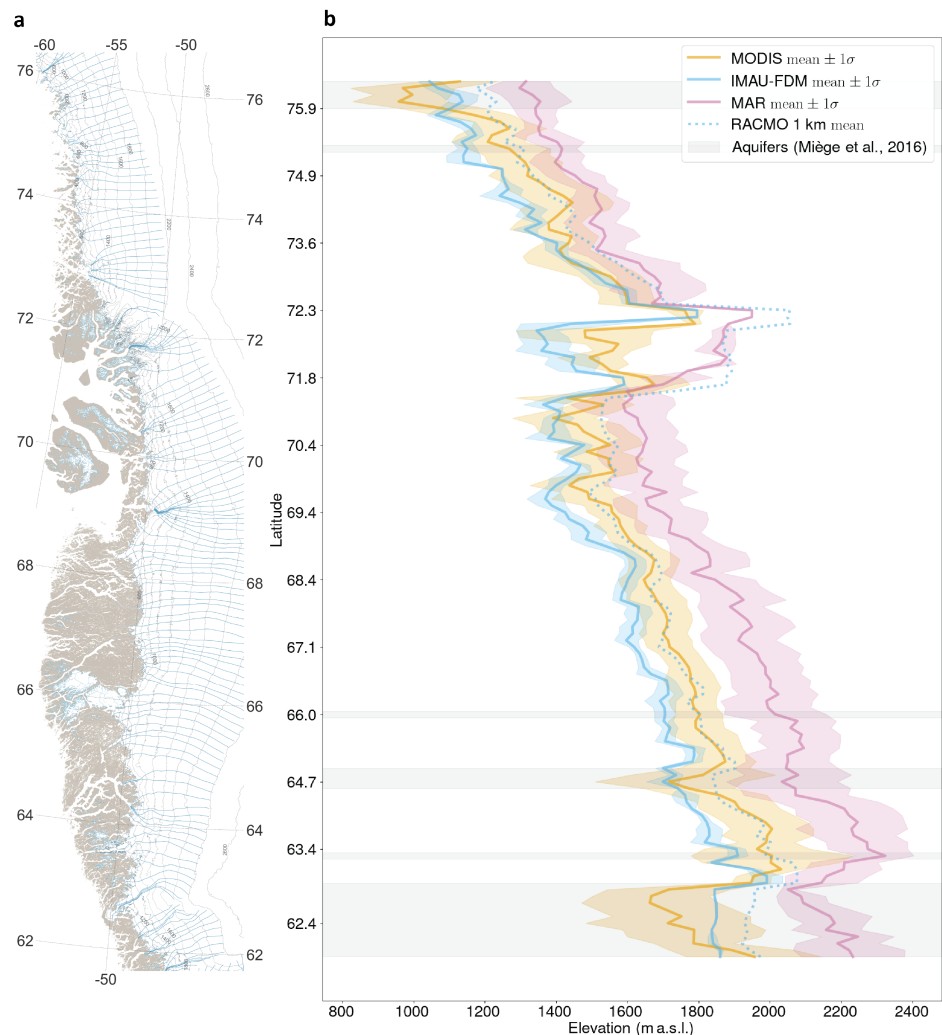

**Figure 3.** The western slope of the Greenland ice sheet and mean MODIS, MAR and IMAU-FDM runoff limits, averaged over the time period 2000 to 2021. **a)** Map of Greenland's west coast showing also the flowlines along which the runoff limits have been calculated. **b)** Mean and standard deviation of MODIS $\max\Upsilon_{\mathrm{obs}}$ and $\max\Upsilon_{\mathrm{rcm}}$ of MAR and IMAU-FDM for all flowlines that fall into the area shown. RACMO 1 km mean $\max\Upsilon_{\mathrm{obs}}$ are also shown but without standard deviation to optimize clarity of the figure. Gray shading indicates latitudes where firn aquifers occur Miège et al. (2016).

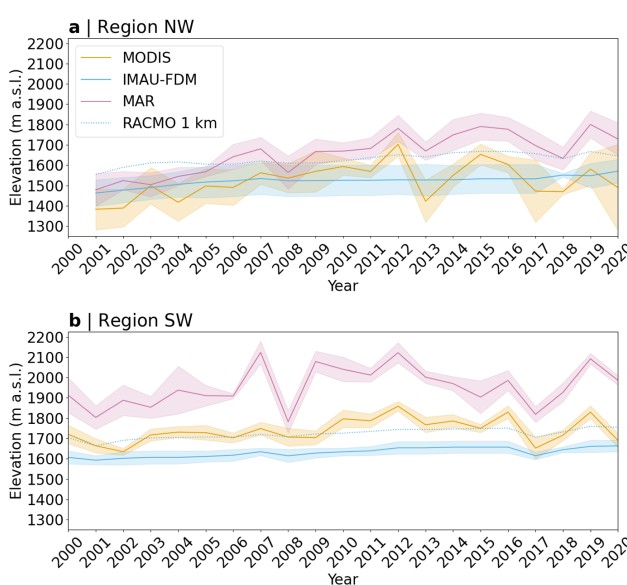

**Figure 4.** Annual mean of MODIS $\max\Upsilon_{\mathrm{obs}}$ and $\max\Upsilon_{\mathrm{rcm}}$ of MAR, RACMO 1 km and IMAU-FDM. **a)** Averaged over the six flowlines of region NW and **b)** averaged over the six flowlines at around the K-transect (region SW, see Fig. 1). Shading illustrates annual variability ($\pm 1\sigma$) of $\max\Upsilon_{\mathrm{obs}}$ or $\max\Upsilon_{\mathrm{rcm}}$ within the two groups of six neighboring flowlines and is omitted for RACMO 1 km to optimize clarity of the figure.

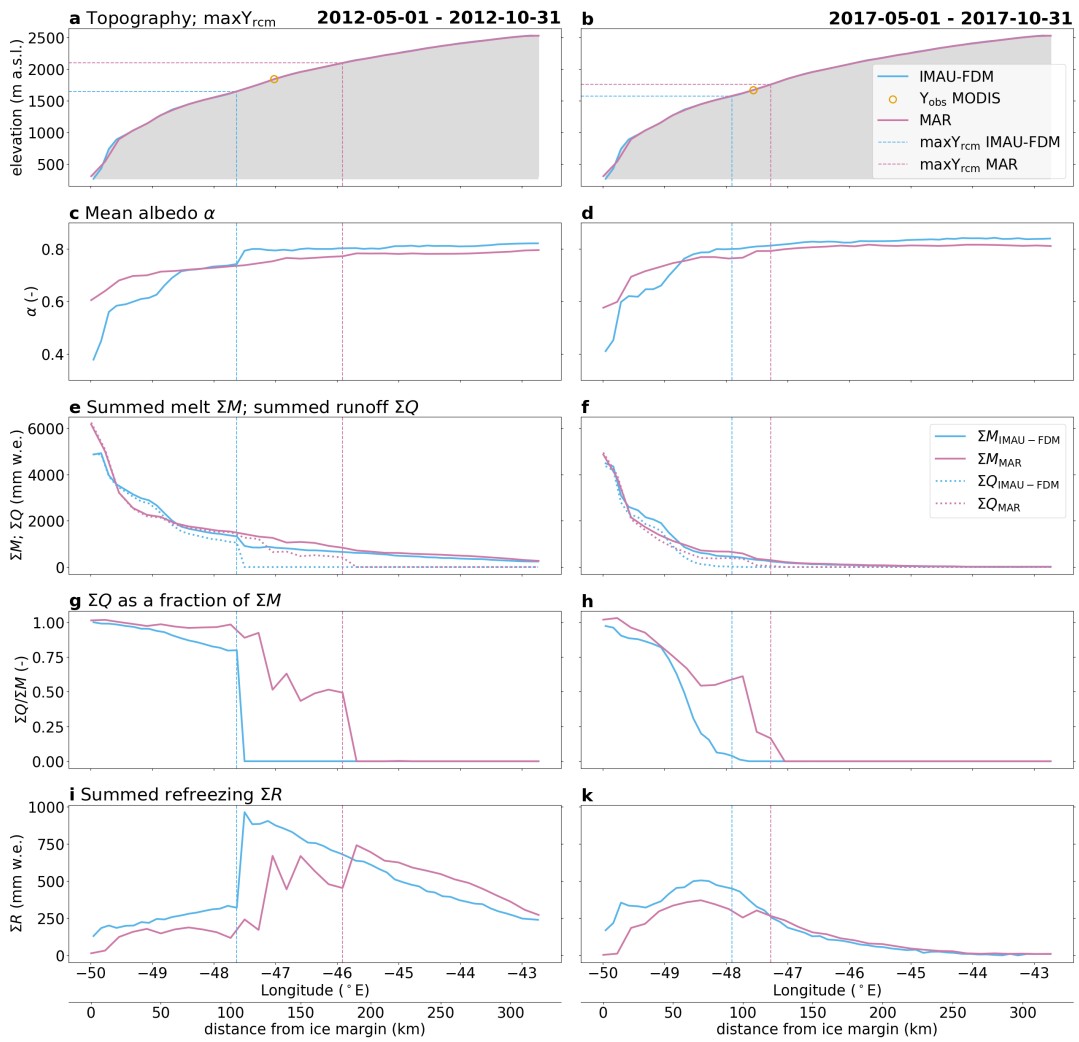

**Figure 5.** Comparison of RCM simulated parameters along the K-transect. The left column of subplots refers to the 2012 melt season; 2017 is to the right. The parameters shown in each row of subplots are explained in the plot titles to the left. Summed values in subplots e to k are summed over the time frames indicated at the top; runoff and refreezing are furthermore depth integrated over the first 20 m of the firn pack.

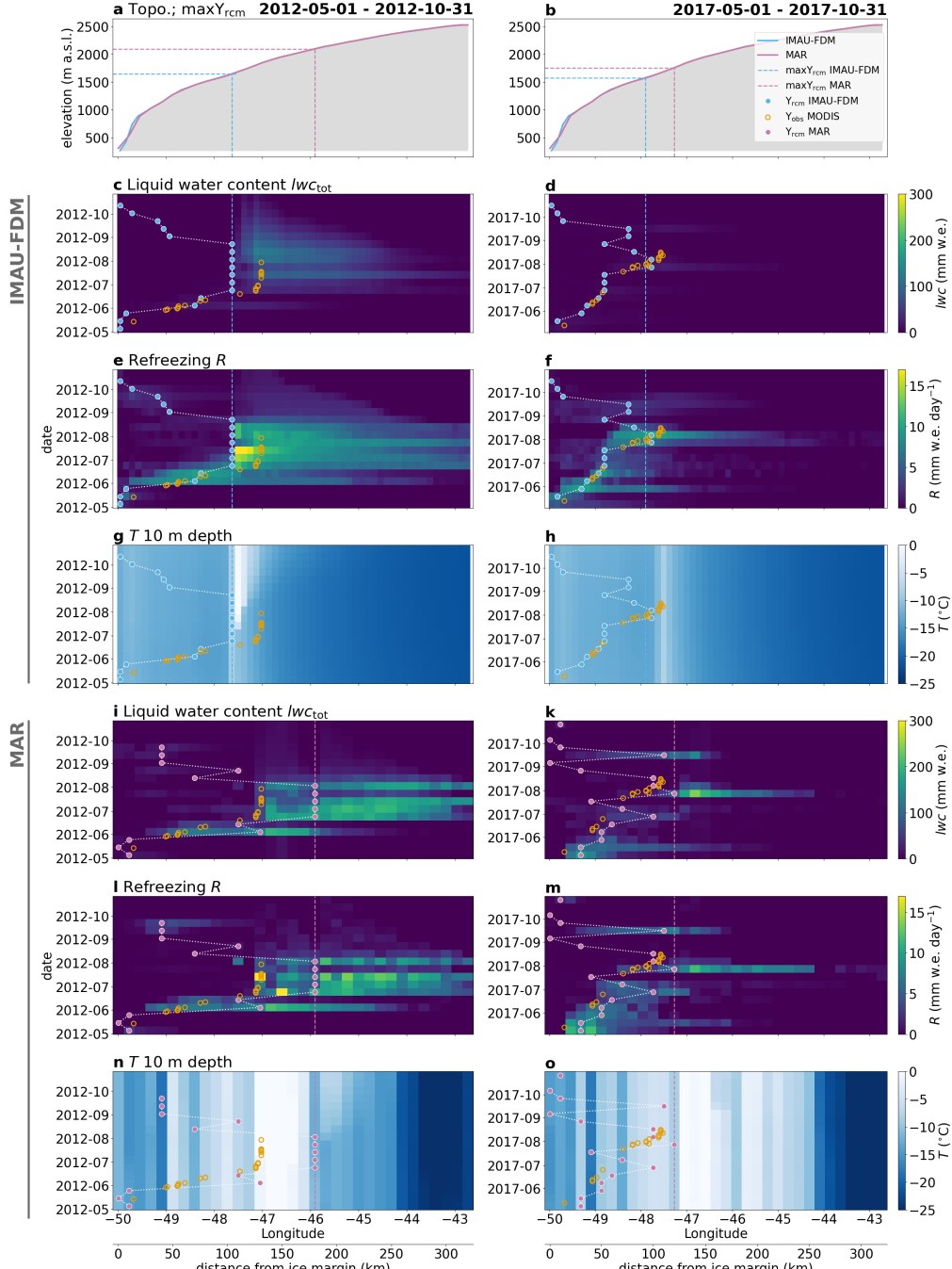

**Figure 6.** Comparison of IMAU-FDM (subplots **c** to **h**) and MAR (**i** to **o**) simulated parameters along the K-transect. Subplots to the left refer to the 2012 melt season; 2017 is to the right. Refreezing and liquid water content (subplots c to f and i to m) are depth integrated over the top 20 m of the firn column. Blue and pink dots denote RACMO and MAR simulated seasonal evolution of the runoff limit, respectively. Orange circles show MODIS-mapped seasonal evolution of the visible runoff limit. All heat maps are given at 10-day temporal resolution. The parameters shown in each row of subplots are explained in the plot titles to the left.

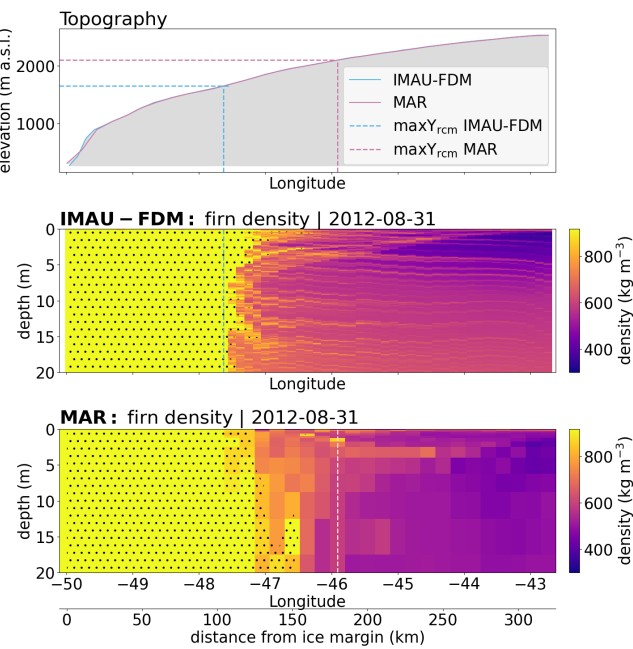

**Figure 7.** Comparison of RCM simulated firn structure along the K-transect and for the year 2012. Dotted areas signify depth intervals where $\rho > 830 \, \text{kg m}^{-3}$ and exceeds pore close-of density. Runoff limits are also shown for the 2012 melt season.

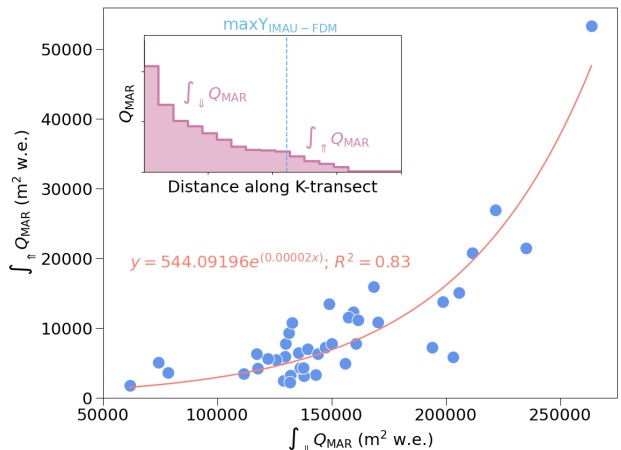

**Figure 8.** Regression of 1980 to 2020 MAR simulated runoff below and above the IMAU-FDM $\Upsilon_{\text{rcm}}$. Every point corresponds to one year and the two runoff values for each year are integrated along parts of the K-transect as illustrated in the inset.

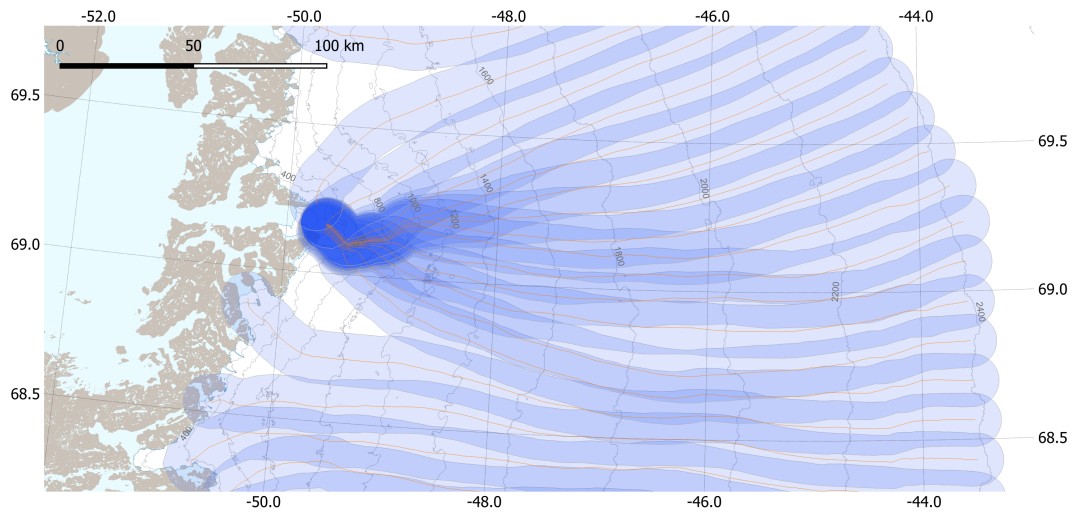

**Figure A1.** Flowlines (orange) and flowline polygons (blue shaded areas) at Sermeq Kujalleq (Jakobshavn Isbræ). Darker shades of blue indicate overlapping polygons.

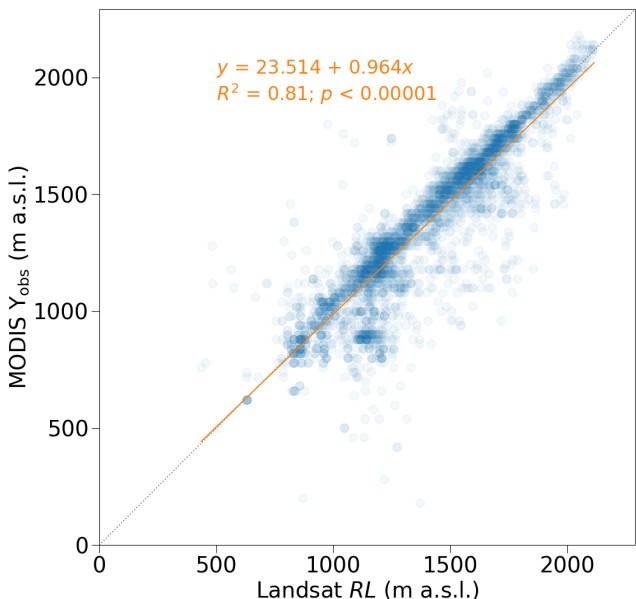

**Figure A2.** Comparison of daily MODIS $\Upsilon_{obs}$ and Landsat derived visible runoff limits ($RL$; Tedstone and Machguth, 2022). The number of samples is $n = 3880$.

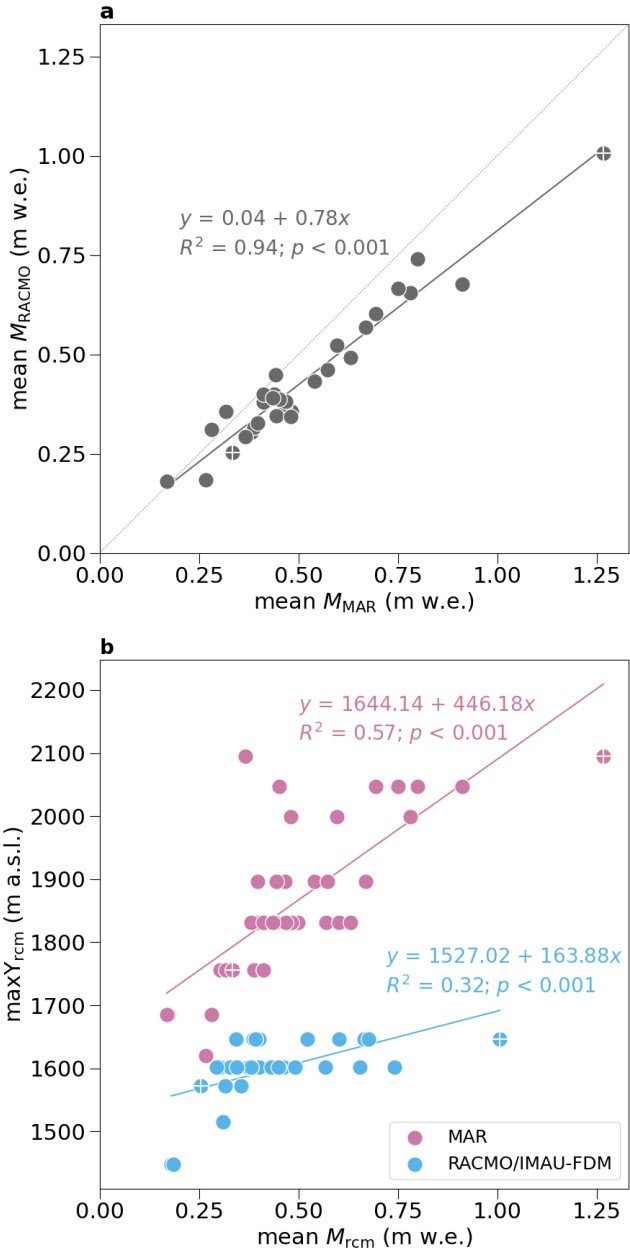

**Figure B1.** Comparison of seasonal simulated melt in MAR and RACMO along the K-transect. Melt is averaged over the grid cells located in-between the lowest and highest of all $\Upsilon_{\text{rcm}}$ and over 1 June to 30 September of each year. **a** Linear regression of MAR and RACMO seasonal melt. **b** Scatterplot of seasonal melt in MAR and RACMO vs. $\Upsilon_{\text{rcm}}^{\text{MAR}}$ and $\Upsilon_{\text{rcm}}^{\text{IMAU}-\text{FDM}}$, respectively. In both subplots the 2012 and 2017 melt seasons are marked with a plus sign.

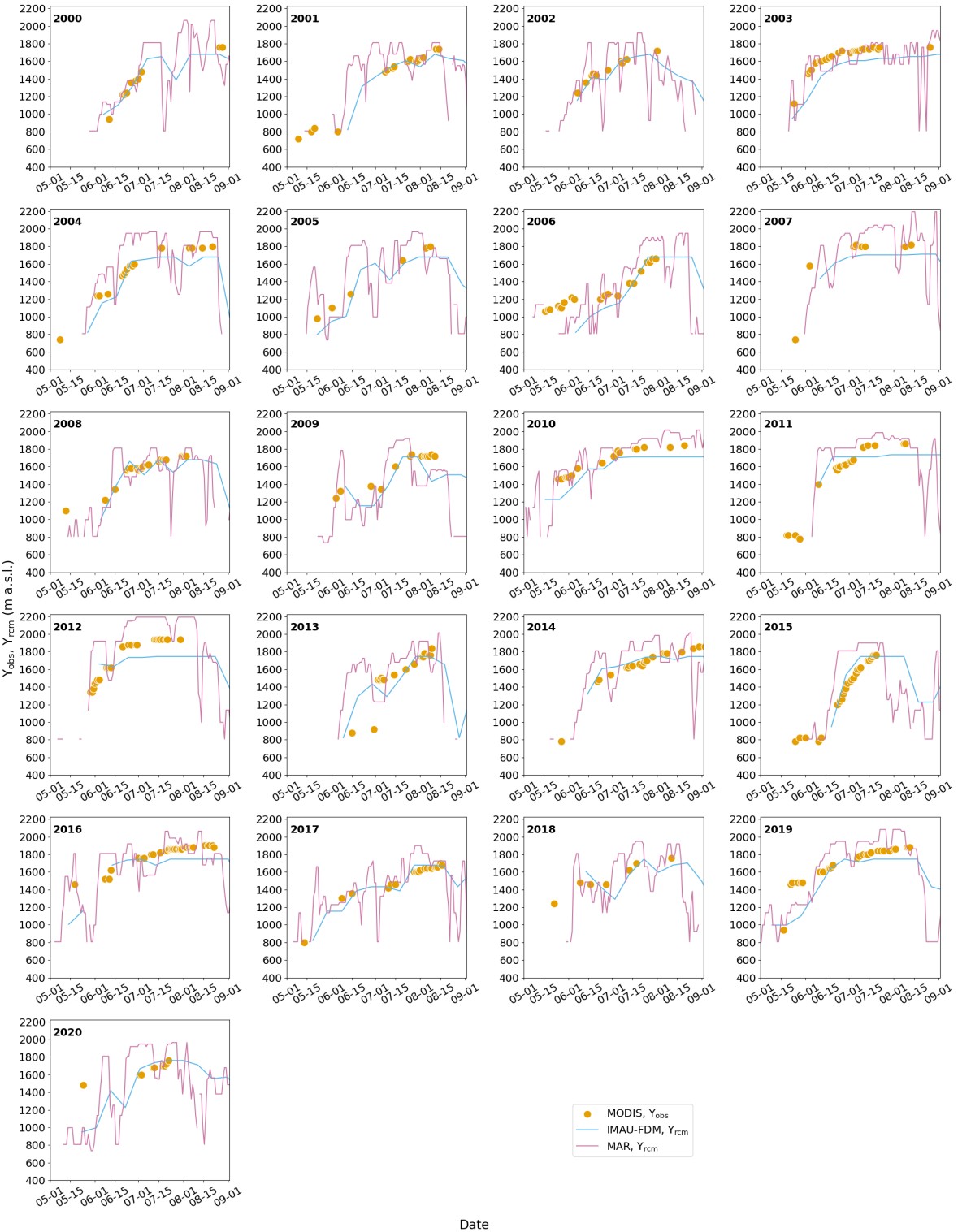

**Figure C1.** Seasonal evolution of $\Upsilon_{rcm}$ simulated by MAR and IMAU-FDM, as well as $\Upsilon_{obs}$ detected from MODIS. The comparison is shown for a flowline-polygon located at around 66 °N on the west coast (region SW, see Fig. 1).

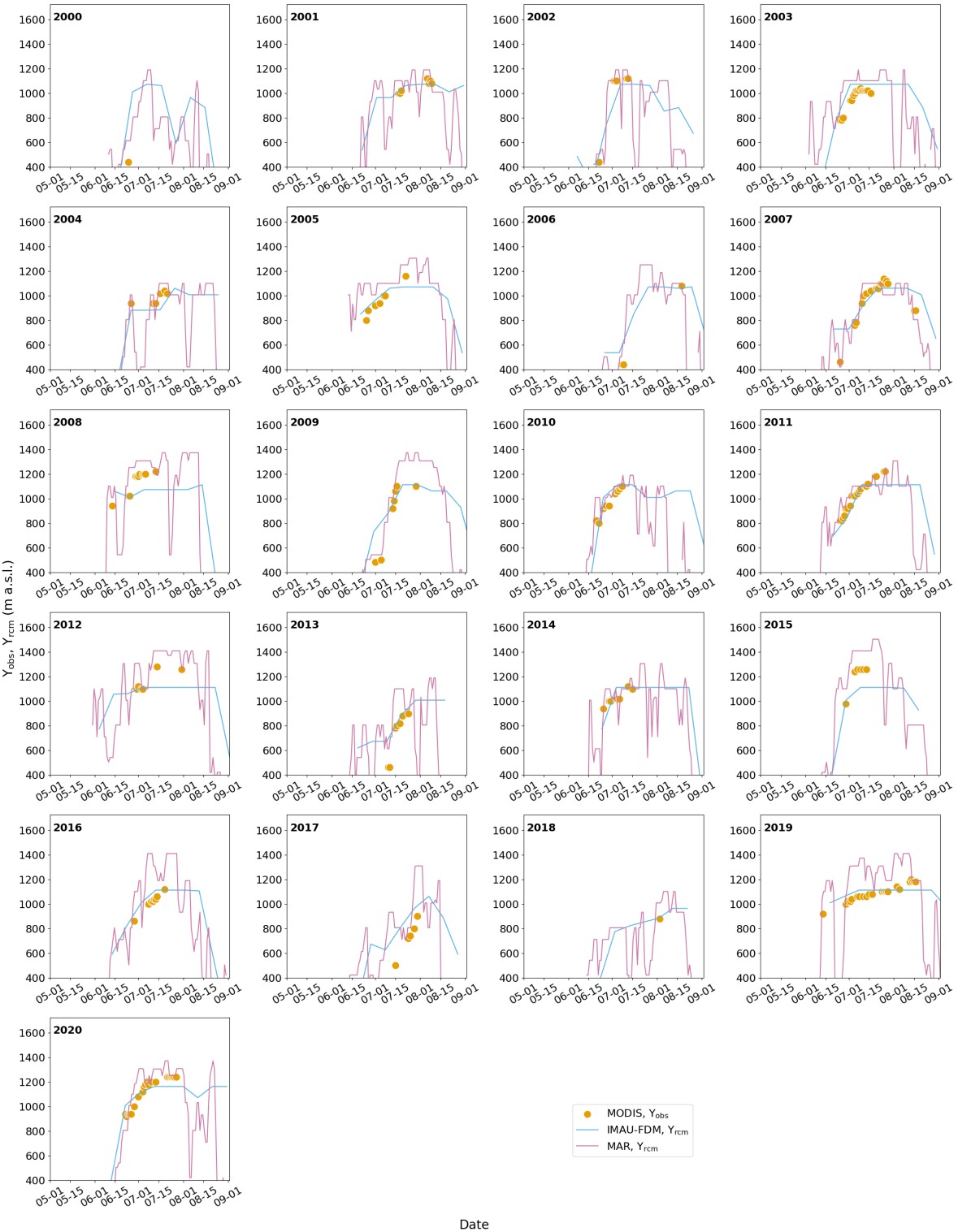

**Figure C2.** Seasonal evolution of $\Upsilon_{rcm}$ simulated by MAR and IMAU-FDM, as well as $\Upsilon_{obs}$ detected from MODIS. The comparison is shown for a flowline-polygon located at around 80 °N (region N, see Fig. 1).

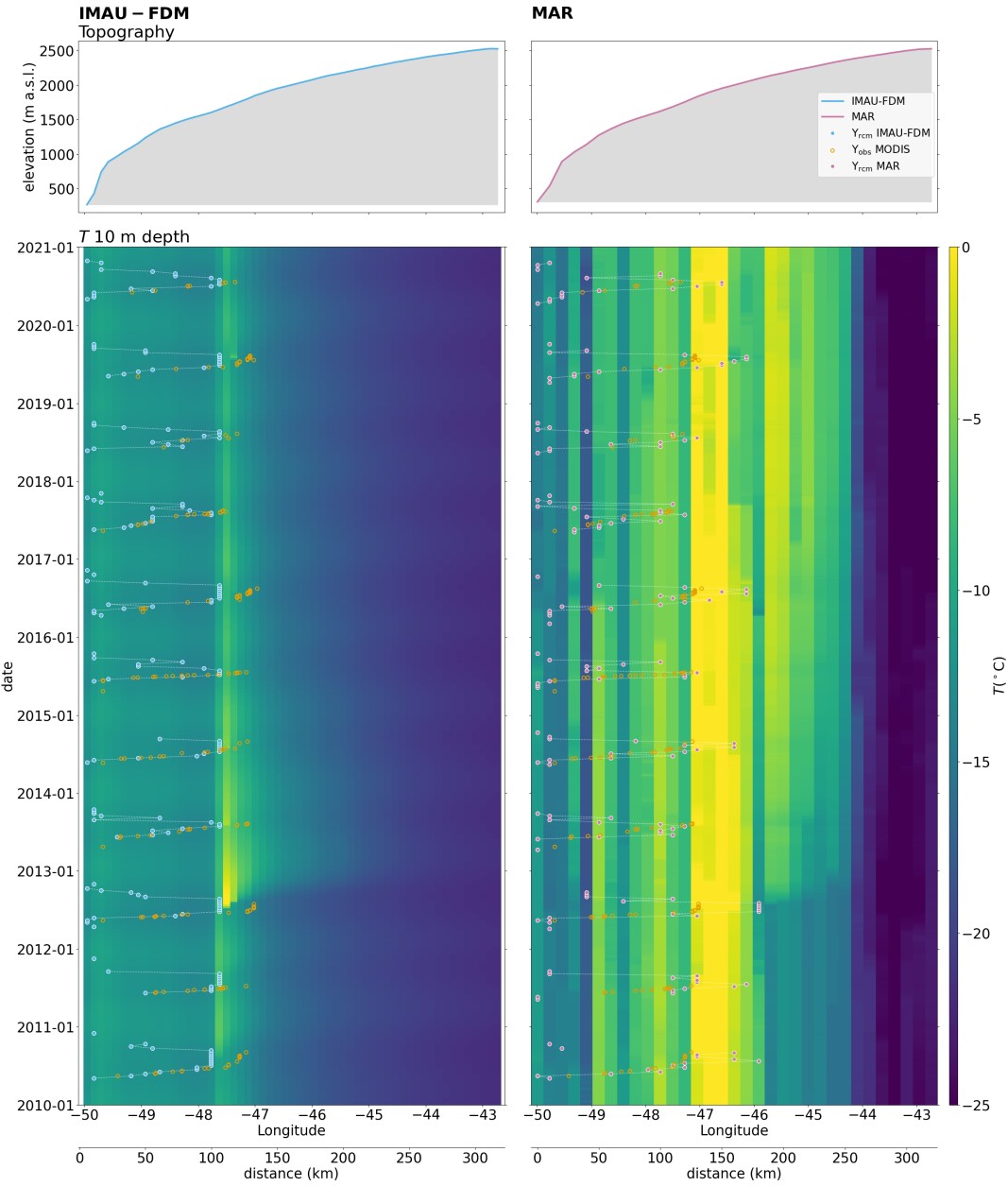

**Figure C3.** Comparison of RCM simulated 10 m firn temperatures along the K-transect, 2010 to 2020. Data to the left are simulated by IMAU-FDM; MAR data are shown to the right.

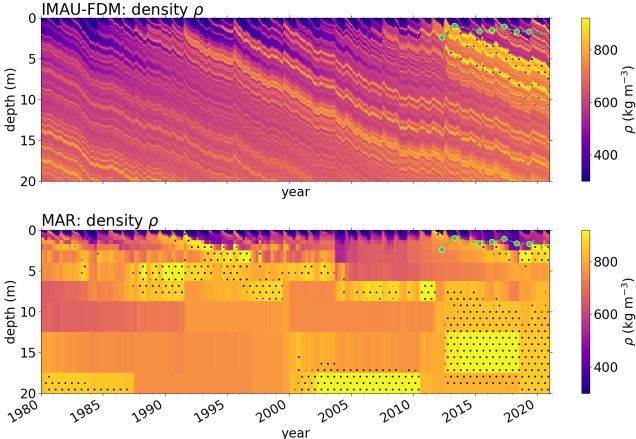

**Figure C4.** Evolution 1980–2020 of RCM simulated firn density $\rho$ in the vicinity of the KAN_U site (K-transect at 1840 m a.s.l.). Dotted areas show where $\rho > 830\,\mathrm{kg\,m^{-3}}$, i.e. exceeding pore close-off density. Green dots mark *in situ* measured depths of the top of the ice slab.

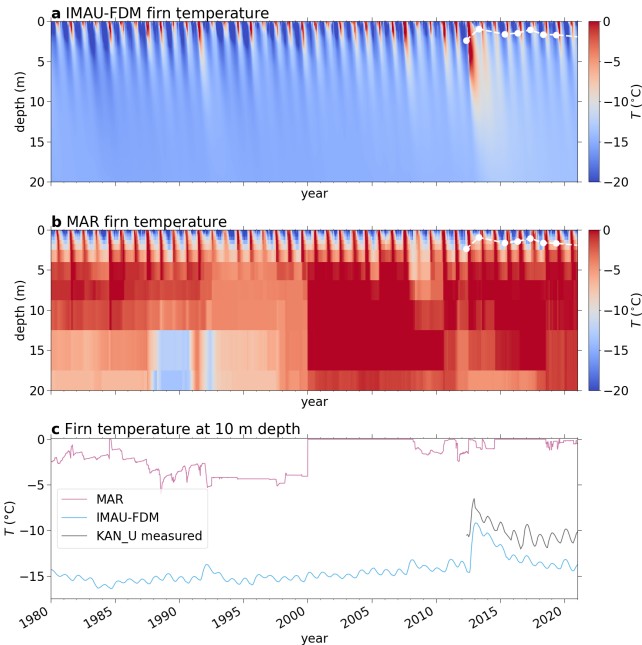

**Figure C5.** Comparison of RCM simulated and measured firn temperatures at the KAN_U site (1840 m a.s.l.) and for the years 1980 to 2020. **a** Firn temperatures for the top 20 m simulated by FDM; **b** top 20 m firn temperatures modelled by MAR; **c** comparison of modelled and measured firn temperatures at 10 m depth (Charalampidis et al., 2016; How et al., 2022; Vandecrux et al., 2023; Vandecrux, 2023). White dots in subplots **a** and **b** denote the top of the ice slab surface according to the measurements summarized in Rennermalm et al. (2021).