# Peer review of "Runoff from Greenland's firn area – why do MODIS, RCMs and a firn model disagree?"

_EGUsphere, 2024_

## Referee Comment (RC2)

Review of: "Runoff from Greenland's firn area – why do MODIS, RCMs and a firn model disagree?"

By Horst Machguth et al.

**Summary**

This paper examines and compares meltwater runoff limits in Greenland (i) derived from MODIS imagery and (ii) predicted by two regional climate models (RCMs), RACMO and MAR. The authors find that in general the runoff limits predicted by RACMO are lower than he observations, and those from MAR are higher than observed. The variability in the MAR limits is more closely aligned with the observations, while the RACMO results show comparatively little interannual variability. The authors attribute much of the difference between the models to differences in the meltwater schemes (bucket schemes). The higher runoff limits in MAR leads to higher predicted runoff volume (up to 29% along the KAN-U transect) than in RACMO.

In general, I found this paper to be scientifically sound and well written. It will make a quality contribution to our understanding of meltwater processes on ice sheets (and limitations in modeling them). There are several issues that should be addressed prior to publication, but I expect that most of these should be quite tenable. I have split my comments into "general comments" and "specific and line by line comments"; note that the line-by-line section includes both small comments (e.g., typos) and larger questions I had as I read the paper.

**General Comments**

1. My primary comment is that the discussion (particularly Section 5.2) needs to be a bit more substantive and based on the results presented. One of the chief takeaways of the paper seems to be that the formulation of the bucket scheme, but this section only sparsely uses the results to support their claim that the bucket scheme is responsible for the differences. For example, is there information that I can glean from figure 5 to support this?

In part this concern comes from the fact that the comparison uses two RCMs, and there is much more in RCMs than the bucket scheme. For example, are RACMO and MAR predicting similar amounts of snowfall and similar winter temperatures to each other, which could change the melt dynamics? The authors also present on albedo differences between the two, but are there other differences in the terms in the energy balance (e.g. calculation of the turbulent fluxes, downwelling longwave, etc.) that could be different between the two? What about how the models handle heat transfer, especially with phase change and associated latent heat?

I don't think that the authors need to do a ton of additional analysis to this end, as I agree that the formulation of the bucket scheme is likely to make a difference. But, I do think it would be appropriate to include a paragraph or two about what other factors might

contribute to the differences between the model, and why the bucket scheme formulation is the most important one.

2. This could be my ignorance, but throughout the paper the authors seem to use RCM and firn model somewhat interchangeably. I've previously operated thinking that the RCM is an atmospheric model, which is coupled to a subsurface snow/firn model. A bit of language (in section 2.2 perhaps?) clarifying this may be useful.

3. The implications of the paper seem to be mostly limited to a vague paragraph at the end of section 5. Would it be a reasonable amount of work to include the total GrIS runoff from both MAR and RACMO and discuss the uncertainty in runoff with a more detailed discussion of the implications on our understanding of GrIS SMB?

4. Clarity: mostly the paper is well written and clear, but there are a number of instances (especially in the discussion) that were not written clearly. I note some of these in my specific comments below. My recommendation is to try to avoid writing in passive voice.

5. Model settings: can you provide more detail about the firn model settings that were used? How were the firn columns spun up? Are model settings the same to the extent possible, e.g. surface snow density, etc.? Do both models use similar surface energy balance schemes?

6. The figures are creative and well made, but I generally found that soft colors were hard to see and that text (e.g., legends, axis labels) is too small. (There are also specific comments below that I made while reading the paper.)

**Specific and Line by line comments**

L31 paragraph: It may be worth mentioning here that the ELA and runoff limit vary from year to year (or calling it a "zone" to encompass that variability?), to differentiate between shifts in ELA and runoff limit that change the long term mean SMB.

46: "oppose" – I would choose a different word here. Intercompare? Or just "compare"? "analyze the differences between the runoff limits..."?

47: remove period after 2021

Section 2.2 (related to my general comment 2 above): Can you provide slightly more detail/clarification about the differences between RACMO2.3p2 and IMAU-FDM, especially in the context of how you use them in this paper? My previous impression was that IMAU-FDM was coupled to RACMO as the subsurface scheme – is this not the case? Here, is the only reason you are bringing in the offline version of IMAU-FDM to evaluate model physics and outputs that are not provided from the coupled model, or are you running IMAU-FDM as well for comparison? (Table 1 clarifies this somewhat, but I think it would be helpful to clarify the text slightly as well.)

83: "agree to" → "agree with"

90: I get the gist of what you are doing as described here; however, is it possible to use Figure 1 to help illustrate the polygons?

Section 3.2.2 – regarding the IMAU bucket scheme – can you provide more specific detail about how it handles ice lenses and slabs, as you do for MAR? (this will help a lot with clarity of discussion section.)

151: Please clarify: Is the annual maximum runoff limit the highest elevation where runoff occurs for each year? Given the definition of runoff limit provided in the introduction, why is the $\max\Upsilon_{rcm}$ not simply where runoff is greater than zero? Also, would it make more sense to consider a $\max\Upsilon_{rcm}$ threshold as a percentage of the annual snowfall than just a value?

183: can you speculate why the approach does not work well in that terrain?

Figure 2: The caption and accompanying text needs more detail (text especially) of how to read the various lines. It may help to label the pale lines "IMAU-FDM daily" and "MAR daily". I initially missed the fact that you say where the transects are at the end of the caption. Perhaps include that detail in the first section of the caption, and make the flowlines bolder in figure 1? Figure 1 legend could also include the flowlines. Also, the axis labels on this figure are too small, and the figure would be easier to view if the colors were bolder. I don't quite understand the lat/lon labeled at the tops of the figures, as this is a transect and not a point?

200: What difference are you referring to specifically here?

Figure 3: This is a neat figure. However, the text size throughout and the soft colors make it difficult to read. In panel a, is it possible to darken the flowlines as well?

Figure 3/Line 200 – It was surprising to see RACMO1k and IMAU-FDM differing so much. I wrote this comment before seeing the text about this later in the paper. I don't think you need to add more in this paper, but it is surprising to see and I hope you will investigate further in the future.

Figure 4: perhaps include the region along with the panel label at the top to add clarity, i.e. a: NW and b: K-transect

204: can you quantify this variability with a simple correlation?

208: "shorter in the north than in the south" – is this robust, or just the case for the areas you picked?

212: It May be useful in some cases when talking about a specific RCM to use e.g. $\Upsilon_{rcm}^{MAR}$ or $\Upsilon_{rcm}^{RAC}$.

217: "appears more step-wise": Is this just an artifact of the gridding?

219: This looks like it is only true for MAR?

Figure 6: the direction of the y-axis is opposite what I would expect (I would expect time to proceed downward in the direction of reading) – so I recommend switching that or noting this in the caption.

Figure 6: It is not clear to me why the 10m firn temperature at the lower elevations much cooler than at the runoff limit (e.g., Figure 6g). Why does the melt cause 10m temperature at runoff limit to increase to near the freezing point, whereas at lower elevations this does not occur? I suspect this is due to there not being firn at all (figure 7), but it would be good to clarify here. (Perhaps label the figure $T_{10m}$ instead of $T_{firn}$ $10\ m$, as there isn't actually firn there?

245: Can you explain the odd (discontinuous in distance) 10-m temperatures in MAR? I would expect a (more or less) monotonic change as distance increases.

250: does this mean that in IMAU-FDM, ELA and $\max\Upsilon_{RCM}$ are effectively the same thing?

266: "MAR simulates runoff between the two runoff limits" – not exactly clear what you mean by this. I think you mean the additional runoff simulated by MAR above the IMAU-FDM runoff limit and below the MAR limit, but perhaps there is a clearer way of stating this.

273: "the larger the difference in total runoff simulated by the two RCMs" – even with the below paragraph I think is phrased as too strong of a claim, as I don't see anything in this regression based on the total runoff in each. I think the claim needs to be a bit more nuanced, along the lines of "the amount of MAR melt above the IMAU-FDM runoff limit increases exponentially as a function of the MAR melt below the IMAU-FDM runoff limit. Assuming the MAR and IMAU total melt below the IMAU-FDM runoff limit are similar, this implies that the difference in predicted melt between MAR and IMAU increases in high-melt seasons". (or rearrange the section a bit so that the comparison of common area is not at the end.

279: is this $\max\Upsilon_{RCM}^{IMAU}$?

285: "16 % out of which almost four fifths" (and previous sentence) – these statistics are hard to follow – i.e., 80% of 16% - can you make it a bit clearer for the reader (perhaps adding the actual volumes would do this?)

288: "regardless of fundamental differences between runoff processes detected from remote sensing and their simulations." – clarify the sentence structure here.

306: Aren't bucket schemes instant (all routing occurs within a model time step)? That could be mentioned here instead of "RCM vertical routing is much faster".

311: It is not clear to me how what is described in this paragraph is lack of inertia – can you elaborate what you mean?

315: "This feedback mechanism, by which ice slabs thicken, is challenging to mimic through a relatively instantaneous bucket scheme." I don't necessarily agree with this – in the model domain, once there is meltwater percolating to the slab all snow/firn above the slab is temperate. If the slab is below freezing, some of the meltwater can refreeze (according to the heat transfer scheme in the model). Why would an instantaneous bucket scheme prevent the model from simulating this process correctly, albeit in a single time step rather than over some timescale? If the slab is being buried in IMAU-FDM and not in reality, a simpler explanation to me is that there is too much snowfall and not enough melt in RACMO at KAN-U. Wouldn't this be consistent with your findings that IMAU-FDM has lower-elevation $\max\Upsilon_{RCM}$?

316: "In particular, both RCMs do not permit any slush formation and even thick ice layers must remain "permeable" for meltwater to be routed vertically." I am not sure what your point is with this sentence – can you expand a bit to clarify what you are saying?

330: "in the absence of pore space, even moderate amounts of melt will run off" – isn't this what would be expected in reality? If there is no pore space available to store the water shouldn't all water run off?

332 – 335: in general this paragraph is hard to follow, in part because (as I noted earlier) the description of the bucket scheme in IMAU-FDM is not fully described. Here it is implied that any meltwater is allowed to pass through ice slabs – is that correct?

336: "pronounced step change in surface albedo in 2012" Is this shown anywhere? Is this in time or space? Not exactly clear what you mean here.

337: Why does higher albedo reduce likelihood of "meltwater percolating to the bottom of the firn where it would run off"? Or are you trying to say that the higher albedo above that step change reduces melt and thereby reduces the volume of meltwater percolating to the bottom of the firn and running off?

340: on one hand/on the other – colloquial phrase

358: "As the primary cause we identify the discrepancies between the two maxYrcm": this could be rephrased to be clearer, e.g. the primary cause is the fact that MAR consistently predicts a higher runoff limit (and thereby a larger area producing runoff) than IMAU-FDM"

359: why is this surprising? If both predict similar runoff in the ablation zone, wouldn't the one with higher runoff limit be expected to produce more runoff?

393: "strongly later water flux" – can you clarify what this means, or perhaps a typo (layered?)?

Figure A4: this is pretty wild to see this much disagreement – I hope you'll continue work like the present paper to figure out what is going on with the models here.

---

## Author Comment (AC1)

**Reviewer #1**

**Review of Machguth et al. 2024**

This paper provides an update to a methodology to determine runoff limits on the Greenland Ice Sheet from MODIS and in addition investigates differences in runoff from these observations as well as MAR and IMAU-FDM models, partially in RACMO.

While there is clearly some new science in this paper I would argue it needs some major revision if it seeks to answer the question posed in the paper title. The research goal of the paper is unclear, is it to introduce an improved method for using satellite data, or to compare different methods for calculating runoff? I would argue that for either improvements are needed, due to the following reasoning.

Thank you for your comment. Our main goal is to answer the question, to what degree do RCM simulations and MODIS observations of the runoff limit agree? This required us to make some improvements to our remote sensing methodology originally introduced in *Machguth et al.* (2022), mainly increasing its flexibility to be used ice-sheet-wide. The modifications are not substantial enough to merit a dedicated publication. We have made our motivation clearer throughout the text and modified the wording, where needed.

1. The new methodology for improving detection of runoff limits from MODIS is not compared with other remote sensing methods, or with the previous method in Machguth 2022. Although the improvement here is that the method can now be used in more areas, without any validation or comparison it is impossible to judge the validity of the method.

   We considered our changes as minor in the sense that the basic method of detecting the runoff limit remains the same. The actual runoff limit retrievals will not be influenced by whether they are being derived along east-west transects or in flowline polygons. However, we understand that these considerations are not obvious to the reader, so we now provide a similar comparison to *Tedstone and Machguth* (2023) as was provided in *Machguth et al.* (2022).

2. The authors make suggestions as to why MAR and RACMO/IMAU-FDM may differ but can't actually evidence this. An RCM and a firn model are very different things, and while it is clear that there is a difference in the implementation of vertical water percolation between the two, that is not the only difference between them. A firn model is run on a very different vertical resolution and will be dependent on RACMO forcing in this case, which will also influence the runoff. The bucket method may only be part of the story as the paper does not compare like for like, or truly compare the two RCMs, or explain why they differ from MODIS completely.

   We believe there are two aspects that need to be distinguished.

   (1) We agree that an RCM is not the same thing as a firn model. However, the polar RCMs we investigate here have firn models embedded. These embedded firn models do not differ fundamentally from dedicated firn models (such as IMAU-FDM). We compare an RCMs firn module to a firn model. We made this clearer throughout the text.

   (2) We disagree that differences such as model vertical resolution prevent simulations of firn from being compared to one another. For instance, two given firn models can be

more different from each other than two firn modules in two RCMs (such potential is indicated in *Vandecrux et al.,* 2020). Specifically concerning vertical resolution, even if we were able to make the resolutions of both models identical, this would likely create new issues that would severely hamper comparison. For example, MAR includes a parameterization that a certain percentage of water runs off immediately when it encounters a layer of the density of ice. If we were able to run MAR at the same high resolution as the RACMO (or IMAU-FDM) firn module, this parameterization would only allow water to reach the bottom of the model domain very rarely thanks to thin layers of ice in the simulated firn column. The parameterization is thus optimized for the coarse MAR firn layers. There are more parameterizations in MAR, RACMO and IMAU-FDM that are optimized to work in the context of the modelling framework they are in.

Furthermore, we emphasize that MAR's and RACMO's firn modules, as well IMAU-FDM, are used to simulate the same processes, one of them being the interaction of firn and meltwater. Given that it is not possible to run all of these models with, for instance, the same common resolution and inputs, our solution is to compare model output and to assess qualitatively which parameterizations likely contribute most to differences in output.

While we believe that our findings show that differences in the implementation of the bucket scheme strongly influence simulated runoff limits, we agree that interpretation needs to be done cautiously. To do so, we have carried out further analysis with a main focus on simulated melt and how MAR and IMAU-FDM simulated runoff limits react to changes in melt. We present and discuss these additional analyses.

**Below I provide line by line comments to add detail to the above.**

Title- The paper doesn't really compare RCMs (plural). It jumps from sometimes including RACMO, sometimes IMAU-FDM forced by RACMO, but to really compare RCMs a full comparison of MAR and RACMO would be needed.

We agree, such a comparison would be favourable. However, RACMO detailed firn output is not written to output files. RACMO does provide depth-integrated variables at a daily resolution (firn air content, total water content). However, other variables such as vertical density, temperature and liquid water profiles are only available from IMAU-FDM. IMAU-FDM outputs these data at 10-daily (for the uppermost 20 m) or 30-daily resolution (for the entire firn column) because otherwise the file sizes become unmanageable. Furthermore, also the IMAU-FDM firn output is used in numerous studies. Hence, we decided to compare MAR and IMAU-FDM. We use plural RCMs because certain parameters need to be compared between MAR and RACMO.

Line 9- The paper does not demonstrate that the difference in implementation of the bucket scheme are responsible for the disparity, only that this is a possible explanation posited by the authors.

We refer to our detailed answer to the reviewer's major comment #2. We have revised the text to more clearly argue that the percolation schemes are the main reasons for the observed differences. At the same time we also now provide a more detailed analysis of the influence of differences in melt on simulated runoff limits (see also our answers to reviewer #2).

Line 18- Suggest addition of 'among' our most advanced tools here. Remote sensing, firn models etc. are all advanced tools that contribute to our understanding.

The statement has been removed (see our reply to the following reviewer comment).

Line 19- Do all these papers actually say RCMs are our most advanced tool? E.g. are they not a part of some of the methodologies described in IMBIE?

These papers were cited as examples of assessing "past, present and future surface mass balance of the Greenland and Antarctic ice sheets". As this was unclear, we revised the sentence to simply say "are widely used to assess ..."

Line 46- Confused by the use of the word 'oppose' here, how do you oppose the runoff limits?

Replaced by "compared"

Line 46- Landsat is mentioned here- why would MODIS be used instead of Landsat? The context is missing here, and a reference for any work done with Landsat as well as justification for why the results in this paper are not compared to any results using Landsat.

This was unclear. We now introduce our MODIS study (*Machguth et al.,* 2023) earlier in the introduction and then state that we use MODIS runoff limits in this study because of their higher temporal resolution.

Line 75- This is further evidence that MAR vs IMAU-FDM is not a straightforward comparison and that there may be other reasons for differences between them.

We agree that it would be better to have all data at the same temporal resolution, optimally 1 day. Unfortunately, the data are provided at different temporal resolutions. Nevertheless, we believe that a comparison is valid (see our detailed answer to the reviewer's major comment #2) and that we provide clear evidence that differences in the implemented of the bucket schemes contribute to the observed differences in modelled runoff limits. We have clarified our argumentation, also to address comments by reviewer #2.

Line 80- Again this makes me more confused why Landsat isn't used, or at least compared against for the MODIS methodology.

We hope this has now become clearer as we have modified the text to better explain why we here prefer to use MODIS data.

Line 94- Does the new method give the same results for the areas covered in Machguth 2022?

We now show the comparison of MODIS (improved methodology) to Landsat (*Tedstone and Machguth,* 2022) Landsat runoff limits.

Line 105-'used no more' or better might be 'not used'.

Agreed.

Line 105- Is the difference between clean and dirty ice a function of water depth? This doesn't quite make sense, clean ice, dirty ice and ponded water are different things.

The aim of this parametrisation was to represent the supraglacial lakes (the albedo of water) as explained in *Lefebre et al*. (2003).

Originally, the bare ice albedo could vary between 0.15 and 0.55 as a function of water depth but in view of the spatial resolution of MAR (not explicitly resolving the presence of lakes) and the lack of observations to validate such a parametrisation, we have reduced a lot the possible

range of bare ice albedo values (0.5-0.55) to decrease a lot the importance of this parametrisation in the recent MAR versions. We have added an explanation in the text.

Line 137- Could the differences in albedo scheme also contribute to the differences in runoff found between this model and MAR?

The different albedo schemes certainly have an influence. For this reason, albedo is compared in Figs. 5c and d and melt (directly affected by albedo schemes) and runoff (not directly affected by albedo schemes) are compared in Fig. 5e and f. The figure shows that at higher elevations the differences in runoff are much larger than the differences in melt. This is one of the key reasons why we conclude that differences in parameterizations of firn meltwater percolation, rather than melting, are mainly responsible for the differences in runoff. At this occasion we also point out that for low elevations MAR shows higher surface albedo and higher melt rate (Fig. 5e and f). This is contradictory but exploring this issue is beyond the scope of this study. Nevertheless, differences in simulated melt also have an influence which we show now in an extended analysis.

Section 3.2 Please state clearly how IMAU-FDM does (or doesn't) deal with ice lenses given this is a key difference with MAR.

Agreed, will be done

Line 172- Could the lack of masking for smaller aquifers influence the results?

Where smaller aquifers are present, the reliability of MODIS visible runoff limits is negatively affected. In such areas, our detection method based on MODIS (as well as the approach based on Landsat) probably underestimate the *actual* runoff limits as optical remote sensing can only detect the *visible* runoff limit. We speak of a probable underestimation as there is not other "ground truth" available. The effect is especially clear in Figure 3 and is mentioned on lines 195-196.

Line 184- How do the detections in the Tedstone paper compare to those made here?

We hope this question is now clarified by the direct comparison of the two approaches.

Section 4.2.1 I found this section and the jumping between RACMO and IMAU-FDM confusing. I would suggest either having a full comparison of MAR, RACMO and IMAU-FDM, or removing the RACMO here as it's not clear what it adds when it's only partially included.

We have modified the text, trying to make the arguments clearer. We agree with the reviewer that including the downscaled RACMO 1 km adds to the complexity, but we would like to keep it as these data are widely used.

Figure 7- This figure to me is the one that really made me question the comparisons made here. It shows very clearly that MAR and IMAU-FDM are working on very different scales and thus capturing different processes, and the suggestion that differences are due to bucket methodology over simplifies this.

As we note in our response major comment #2, we believe that the two models *need* to be compared despite (and indeed because of) their scale differences. Ultimately, they both numerically calculate the same quantity: Greenland surface mass balance including simulating firn and its reaction to melt. They both are used in numerous studies to assess how the

Greenland ice sheet is currently changing and will react to climate change, they both are widely used to assess runoff, sea level rise contribution and so on.

A recently published study (*Glaude et al.,* 2024) shows that under identical forcing, three RCMs differ very strongly in simulated Greenland runoff extent by the year 2100. The difference in simulated runoff areas is particularly striking. The study points out that there is a factor two difference in simulated mass balance. Hence, models need to be compared to understand why their output differs, whether due to e.g. spatial resolution (for which we find no evidence) or firn parameterizations (for which we find evidence). We have addressed the reviewer's general critique as outlined in our replies to their general comments. We have also added reference to the study by *Glaude et al.* (2024).

Line 311- Is this a lack of inertia or just that processes are not accounted for?

The lack of inertia is the result of processes not being accounted for. We have accordingly changed the wording.

Line 318- This paragraph is confusingly written. It starts by comparing RACMO and IMAU-FDM, then compares MAR and RACMO, but not IMAU-FDM and MAR. Stating that RACMO and IMAU-FDM show similar temporal patterns here is hardly surprising given that one forces the other. The RACMO firn model is also mentioned here but hasn't been detailed before.

We agree that the paragraph was unclear and have modified it for clarity. It is correct that RACMO forces IMAU-FDM, but the fact that both show the same reduced temporal variability of the runoff limits is not due to identical meltwater input but due to their firn modules being very similar, apart from a substantial difference in the number of vertical layers. If meltwater input would control temporal variability of the runoff limit, then RACMO, IMAU-FDM and MAR should all show similar temporal variability because the amount of melt in MAR's accumulation area is similar to RACMO (Fig. 5e and f). We investigate and discuss these aspects now in our revised analysis. We also have added more detail explanations of RACMO's firn module.

Section 5.2- As stated above I don't think this paragraph shows the bucket scheme is the main cause of deviations. Several other differences between IMAU-FDM and MAR are mentioned but without justification as to why they are less important.

Please see our answers to earlier comments. It is potentially impossible to *quantitatively* compare the effects of all differences between the two models. We qualitatively estimate that the way the bucket scheme is implemented is of major importance to the simulated runoff limits. At the same time, we state that the two models have substantial differences unrelated to firn simulation.

Also in reply to comments of reviewer #2 we clarified our argumentation why the implementation of the bucket schemes is a major contributor to the observed differences. We also now provide an extended analysis to investigate the contributions of other differences between the RCMs.

Line 386- Proof that this is an improved method is missing. It might cover more areas but it is not validated.

Indeed, we do not want to claim that the method is improved in the sense that its results are more accurate. But it is improved in the sense of being applicable much more flexibly. As

already mentioned, we have added a validation and we also have revised the text to more accurately reflect the nature of the improvement.

Section A2- Please make clearer how this differs from the 2022 paper.

This was unclear, we have revised the text to make the differences clear.

**References cited**

Glaude, Q., Noel, B., Olesen, M., Van den Broeke, M., van de Berg, W. J., Mottram, R., et al. (2024). A factor two difference in 21st-century Greenland Ice Sheet surface mass balance projections from three regional climate models under a strong warming scenario (SSP5-8.5). *Geophysical Research Letters,* **51**(22). https://doi.org/10.1029/2024gl111902

Lefebre, F., Gallée, H., van Ypersele, J.-P., & Greuell, W. (2003). Modeling of snow and ice melt at ETH Camp (West Greenland): A study of surface albedo. *Journal of Geophysical Research: Atmospheres,* **108**(D8). https://doi.org/10.1029/2001jd001160

Machguth, H., Tedstone, A., & Mattea, E. (2023). Daily variations in western Greenland slush limits, 2000 to 2021, mapped from MODIS. *Journal of Glaciology,* **69**(273), 191–203. https://doi.org/10.1017/jog.2022.65

Tedstone, A., & Machguth, H. (2022). Increasing surface runoff from Greenland's firn areas. *Nature Climate Change,* **12,** 672–676. https://doi.org/10.1038/s41558-022-01371-z

Vandecrux, B., Mottram, R., Langen, P. L., Fausto, R. S., Olesen, M., Stevens, C. M., et al. (2020). The firn meltwater Retention Model Intercomparison Project (RetMIP): evaluation of nine firn models at four weather station sites on the Greenland ice sheet. *The Cryosphere,* **14**(11), 3785–3810. https://doi.org/10.5194/tc-14-3785-2020

---

## Author Comment (AC2)

**Reviewer #2**

Review of: "Runoff from Greenland's firn area – why do MODIS, RCMs and a firn model disagree?"

By Horst Machguth et al.

**Summary**

This paper examines and compares meltwater runoff limits in Greenland (i) derived from MODIS imagery and (ii) predicted by two regional climate models (RCMs), RACMO and MAR. The authors find that in general the runoff limits predicted by RACMO are lower than he observations, and those from MAR are higher than observed. The variability in the MAR limits is more closely aligned with the observations, while the RACMO results show comparatively little interannual variability. The authors attribute much of the difference between the models to differences in the meltwater schemes (bucket schemes). The higher runoff limits in MAR leads to higher predicted runoff volume (up to 29% along the KAN-U transect) than in RACMO.

In general, I found this paper to be scientifically sound and well written. It will make a quality contribution to our understanding of meltwater processes on ice sheets (and limitations in modeling them. There are several issues that should be addressed prior to publication, but I expect that most of these should be quite tenable. I have split my comments into "general comments" and "specific and line by line comments"; note that the line-by-line section includes both small comments (e.g., typos) and larger questions I had as I read the paper.

We thank the reviewer for their feedback. We reply to all comments below.

**General Comments**

1.      My primary comment is that the discussion (particularly Section 5.2) needs to be a bit more substantive and based on the results presented. One of the chief takeaways of the paper seems to be that the formulation of the bucket scheme, but this section only sparsely uses the results to support their claim that the bucket scheme is responsible for the differences. For example, is there information that I can glean from figure 5 to support this?

In part this concern comes from the fact that the comparison uses two RCMs, and there is much more in RCMs than the bucket scheme. For example, are RACMO and MAR predicting similar amounts of snowfall and similar winter temperatures to each other, which could change the melt dynamics? The authors also present on albedo differences between the two, but are there other differences in the terms in the energy balance (e.g. calculation of the turbulent fluxes, downwelling longwave, etc.) that could be different between the two? What about how the models handle heat transfer, especially with phase change and associated latent heat?

I don't think that the authors need to do a ton of additional analysis to this end, as I agree that the formulation of the bucket scheme is likely to make a difference. But, I do think it would be appropriate to include a paragraph or two about what other factors might contribute to the differences between the model, and why the bucket scheme formulation is the most important one.

This comment is similar to the overall critique of reviewer #1. We refer to our answer there, please also see our response to the major comment number 4 of reviewer #3. Differences in

melt input, caused by differences in the surface energy balance, would obviously trigger differences in meltwater percolation. This was partially considered in the manuscript (mainly Fig. 5 left and right columns) but differences in melt input have now given more attention. The figure shows that both RCMs have similar melt. Albedo is also similar but there are differences in 2012 albedo directly above the RACMO-FDM runoff limit. This sudden change in RACMO-FDM albedo is discussed in the manuscript. Nevertheless, Fig. 5 shows that the main difference is not in melt or albedo, but in a major disparity in refreezing, which happens in the subsurface.

Apart from the revisions already mentioned to reviewer #1 we have more clearly emphasized the above argumentation.

2.      This could be my ignorance, but throughout the paper the authors seem to use RCM and firn model somewhat interchangeably. I've previously operated thinking that the RCM is an atmospheric model, which is coupled to a subsurface snow/firn model. A bit of language (in section 2.2 perhaps?) clarifying this may be useful.

Agreed, we have revised the text to use a clearer terminology

3.      The implications of the paper seem to be mostly limited to a vague paragraph at the end of section 5. Would it be a reasonable amount of work to include the total GrIS runoff from both MAR and RACMO and discuss the uncertainty in runoff with a more detailed discussion of the implications on our understanding of GrIS SMB?

We agree that the importance of runoff in Greenland's firn area for the ice sheet's total mass balance needs to be estimated. However, we would like this study to focus on exploring the reasons behind the models disagreement and quantifying the relevance of firn area runoff to total mass balance is beyond the scope of our study. However, we now highlight the study by *Glaude et al.* (2024) in the introduction and discussion as their work shows that there is very substantial uncertainty in simulations of future mass balance of the Greenland firn region. In particular, they demonstrate large differences in RCM simulated future melt extent which leads to pronounced uncertainties in future projections of Greenland mass balance.

4.      Clarity: mostly the paper is well written and clear, but there are a number of instances (especially in the discussion) that were not written clearly. I note some of these in my specific comments below. My recommendation is to try to avoid writing in passive voice.

We thank the reviewer for pointing out these issues which we have corrected as described below.

5.      Model settings: can you provide more detail about the firn model settings that were used? How were the firn columns spun up? Are model settings the same to the extent possible, e.g. surface snow density, etc.? Do both models use similar surface energy balance schemes?

We have added additional detail and references to publications describing the models.

6.      The figures are creative and well made, but I generally found that soft colors were hard to see and that text (e.g., legends, axis labels) is too small. (There are also specific comments below that I made while reading the paper.)

We have modified axis fonts and colors, were possible.

**Specific and Line by line comments**

L31 paragraph: It may be worth mentioning here that the ELA and runoff limit vary from year to year (or calling it a "zone" to encompass that variability?), to differentiate between shifts in ELA and runoff limit that change the long term mean SMB.

Agreed, added.

46: "oppose" – I would choose a different word here. Intercompare? Or just "compare"? "analyze the differences between the runoff limits…"?

Agreed, done.

47: remove period after 2021

Done

Section 2.2 (related to my general comment 2 above): Can you provide slightly more detail/clarification about the differences between RACMO2.3p2 and IMAU-FDM, especially in the context of how you use them in this paper? My previous impression was that IMAU- FDM was coupled to RACMO as the subsurface scheme – is this not the case? Here, is the only reason you are bringing in the online version of IMAU-FDM to evaluate model physics and outputs that are not provided from the coupled model, or are you running IMAU-FDM as well for comparison? (Table 1 clarifies this somewhat, but I think it would be helpful to clarify the text slightly as well.)

We have added more detail, as also mentioned in our reply to point 2 above. Furthermore, the description of the firn models and modules in Section 3.2 has been extended.

83: "agree to" -> "agree with"

Done.

90: I get the gist of what you are doing as described here; however, is it possible to use Figure 1 to help illustrate the polygons?

Showing all polygons directly in Fig. 1 will make the figure unreadable and the polygons would be small. However, we added a supplementary figure to illustrate the concept.

Section 3.2.2 – regarding the IMAU bucket scheme – can you provide more specific detail about how it handles ice lenses and slabs, as you do for MAR? (this will help a lot with clarity of discussion section.)

Agreed, we have rewritten this paragraph in order to provide more detail.

151: Please clarify: Is the annual maximum runoff limit the highest elevation where runoff occurs for each year? Given the definition of runoff limit provided in the introduction, why is the maxY!"# not simply where runoff is greater than zero? Also, would it make more sense to consider a maxY!"# threshold as a percentage of the annual snowfall than just a value?

On the ice sheet, the runoff limit is indeed defined as the elevation where runoff starts. However, this definition cannot be directly applied to the RCMs. The RCMs simulation of runoff is fundamentally different from the actual processes and there can be numerical issues where tiny amounts of runoff always occur. In discussion with RCM modellers we decided to use a fixed threshold. Using a threshold that is a function of the amount of annual snowfall would

make it very difficult to understand RCM runoff positions. Furthermore, the larger the amount of annual snowfall, the less likely near surface lateral runoff and instead aquifers will form, which are not the topic of this study.

183: can you speculate why the approach does not work well in that terrain?

Yes, we added a brief discussion in the appendix.

Figure 2: The caption and accompanying text needs more detail (text especially) of how to read the various lines. It may help to label the pale lines "IMAU-FDM daily" and "MAR daily". I initially missed the fact that you say where the transects are at the end of the caption. Perhaps include that detail in the first section of the caption, and make the flowlines bolder in figure 1? Figure 1 legend could also include the flowlines. Also, the axis labels on this figure are too small, and the figure would be easier to view if the colors were bolder. I don't quite understand the lat/lon labeled at the tops of the figures, as this is a transect and not a point?

Thank you for this feedback, of which some refers to Fig. 1, if we understand correctly? We have addressed these issues and hope that the new figure is more readable. The coordinates are provided as a rough indication to where the transect is located. This allows the reader to immediately understand whether a transect is in the far north or south. We have mentioned this now and also reduced the coordinate precision to one digit.

200: What difference are you referring to specifically here?

We have removed this sentence.

Figure 3: This is a neat figure. However, the text size throughout and the soft colors make it difficult to read. In panel a, is it possible to darken the flowlines as well?

Done.

Figure 3/Line 200 – It was surprising to see RACMO1k and IMAU-FDM differing so much. I wrote this comment before seeing the text about this later in the paper. I don't think you need to add more in this paper, but it is surprising to see and I hope you will investigate further in the future.

Thank you for the comment.

Figure 4: perhaps include the region along with the panel label at the top to add clarity, i.e. a: NW and b: K-transect

Done.

204: can you quantify this variability with a simple correlation?

Unfortunately, we do not fully understand the reviewer's comment. There are two variabilities mentioned on this line, (1) variability with intensity of the melt season and (2) differences in interannual variability between MODIS and MAR. Both were based on qualitative assessment, which we have now replaced with a quantitative assessment.

208: "shorter in the north than in the south" – is this robust, or just the case for the areas you picked?

This was a qualitative assessment, we agree it would need to be undermined with statistics. As it is not of particular relevance in the context of this study, we have removed the statement.

212: It May be useful in some cases when talking about a specific RCM to use e.g. Y$%& or &%'!"#

Yes, this has been changed

217: "appears more step-wise": Is this just an artifact of the gridding?

We removed this statement as our analysis does not demonstrate this statement quantitatively.

219: This looks like it is only true for MAR?

The challenge is the different temporal resolution of the available data: MAR at 1 day, RACMO-FDM at 12 days. Unfortunately, the latter are unavailable at 1-day resolution. We now briefly comment on this in the text.

Figure 6: the direction of the y-axis is opposite what I would expect (I would expect time to proceed downward in the direction of reading) – so I recommend switching that or noting this in the caption.

We added a comment in the figure caption.

Figure 6: It is not clear to me why the 10m firn temperature at the lower elevations much cooler than at the runoff limit (e.g., Figure 6g). Why does the melt cause 10m temperature at runoff limit to increase to near the freezing point, whereas at lower elevations this does not occur? I suspect this is due to there not being firn at all (figure 7), but it would be good to clarify here. (Perhaps label the figure $T()$# instead of $T^*$+!, 10 $m$, as there isn't actually firn there?

This is correct, there is no firn at elevations much lower than the runoff limit. Consequently, meltwater runs off without releasing any latent heat in the subsurface. We have relabelled the figure as suggested.

245: Can you explain the odd (discontinuous in distance) 10-m temperatures in MAR? I would expect a (more or less) monotonic change as distance increases.

The snow layer of each pixel is evolving independently of the other ones from the initialisation. Moreover, the snow pack vertical discretisation could be very different following the considered pixel. There are thin snow layers close the surface but thick layers (of several meters) at depth. Therefore, the 10 m temperature shown here is the temperature of the snow layer including the 10 m depth. For some pixels, it could be the temperature of a unique ice layer of 15 m thick or to a 1 m thick snow layer close to 10m depth. To reduce this problem of spatial discontinuity (identified in this paper) in the MAR snow pack, a very light spatial smoothing/filtering is now applied over the 1st layers (the deeper layer) of each pixel in the latest version (3.14.1) of MAR (this study uses 3.14.0). We have commented on this issue in the text.

250: does this mean that in IMAU-FDM, ELA and maxY&'$ are effectively the same thing?

This could be the case, but we have not investigated the ELA as simulated by the RCMs.

266: "MAR simulates runoff between the two runoff limits" – not exactly clear what you mean by this. I think you mean the additional runoff simulated by MAR above the IMAU-FDM runoff limit and below the MAR limit, but perhaps there is a clearer way of stating this.

Thank you for pointing this out, we have tried to clarify the wording.

273: "the larger the difference in total runoff simulated by the two RCMs" – even with the below paragraph I think is phrased as too strong of a claim, as I don't see anything in this regression based on the total runoff in each. I think the claim needs to be a bit more nuanced, along the lines of "the amount of MAR melt above the IMAU-FDM runoff limit increases exponentially as a function of the MAR melt below the IMAU-FDM runoff limit. Assuming the MAR and IMAU total melt below the IMAU-FDM runoff limit are similar, this implies that the difference in predicted melt between MAR and IMAU increases in high-melt seasons". (or rearrange the section a bit so that the comparison of common area is not at the end.

The total runoff in RACMO-FDM is represented in the regression by MAR runoff below the runoff limit of RACMO-FDM. The RCMs' similarities in "ablation area runoff" are discussed in the following paragraph. We have changed the wording in line with the suggestion of the reviewer.

279: is this maxY-$%.?

Yes, this information was missing, updated.

285: "16 % out of which almost four fifths" (and previous sentence) – these statistics are hard to follow – i.e., 80% of 16% - can you make it a bit clearer for the reader (perhaps adding the actual volumes would do this?)

Adding volumes can lead to more confusion as the analysis focuses on a transect where runoff has, in a strict sense, the unit $m^2$. We have reworded to clarify the message.

288: "regardless of fundamental differences between runoff processes detected from remote sensing and their simulations." – clarify the sentence structure here.

Done.

306: Aren't bucket schemes instant (all routing occurs within a model time step)? That could be mentioned here instead of "RCM vertical routing is much faster".

Correct, done.

311: It is not clear to me how what is described in this paragraph is lack of inertia – can you elaborate what you mean?

The word "inertia" does not add to clarity. In line with the above comment, we also use "instantaneous" here.

315: "This feedback mechanism, by which ice slabs thicken, is challenging to mimic through a relatively instantaneous bucket scheme." I don't necessarily agree with this – in the model domain, once there is meltwater percolating to the slab all snow/firn above the slab is temperate. If the slab is below freezing, some of the meltwater can refreeze (according to the heat transfer scheme in the model). Why would an instantaneous bucket scheme prevent the model from simulating this process correctly, albeit in a single time step rather than over some timescale? If the slab is being buried in IMAU-FDM and not in reality, a simpler explanation to me is that there is too much snowfall and not enough melt in RACMO at KAN-U. Wouldn't this be consistent with your findings that IMAU-FDM has lower-elevation maxY&'$?

Our statement was misleading or incomplete. The issues lie in the RCMs limiting meltwater storage in firn to irreducible water content. No slush formation is allowed, which by definition is meltwater exceeding the irreducible water content. The ice slabs thicken primarily by accretion of superimposed ice on top of the slabs, which requires slush to persist on top of the slabs for

longer time periods. This cannot be simulated by the RCMs at the moment. Too much snowfall could also cause the effect of the slab getting buried, but this is not the problem here. We have modified the text for the message to become clearer.

316: "In particular, both RCMs do not permit any slush formation and even thick ice layers must remain "permeable" for meltwater to be routed vertically." I am not sure what your point is with this sentence – can you expand a bit to clarify what you are saying?

If the RCMs would treat layers of the density of ice (or any other threshold, such as density of pore close-off) as impermeable, then this would also require to have percolating meltwater ponding on top of such an impermeable layer. The ponding water would completely fill the pore space of the overlying porous layers which would mean that slush is formed in an RCM's firn layer. Both RCMs do not form slush because any water exceeding the irreducible water content is instantaneously routed downwards.

We have reworded the criticised sentence to make the statement clearer in line with the above explanation.

330: "in the absence of pore space, even moderate amounts of melt will run o-" – isn't this what would be expected in reality? If there is no pore space available to store the water shouldn't all water run off?

Yes, this behaviour is intended. The question is more whether the pore space, as simulated by RACMO-FDM, corresponds to the spatial distribution of pore space in reality. We changed the wording to make this clearer.

332 – 335: in general this paragraph is hard to follow, in part because (as I noted earlier) the description of the bucket scheme in IMAU-FDM is not fully described. Here it is implied that any meltwater is allowed to pass through ice slabs – is that correct?

Yes, this is correct. As explained above, no water can pond on any ice layer regardless of who thick that layer is. Hence the water passes through ice layers. However, it does this without any interaction with the ice. For any interaction of percolating meltwater with a layer in the firn model, a layer needs to contain pore space and existing water content in the layer needs to be below the maximum irreducible water content. We have added text to clarify this behaviour.

336: "pronounced step change in surface albedo in 2012" Is this shown anywhere? Is this in time or space? Not exactly clear what you mean here.

Yes, this is shown in Figure 5c. We have added a reference to the figure for clarity.

337: Why does higher albedo reduce likelihood of "meltwater percolating to the bottom of the firn where it would run off"? Or are you trying to say that the higher albedo above that step change reduces melt and thereby reduces the volume of meltwater percolating to the bottom of the firn and running off?

Yes, this is what we are trying to say. We changed the wording to make the message clearer.

340: on one hand/on the other – colloquial phrase

We have adapted the wording.

358: "As the primary cause we identify the discrepancies between the two maxYrcm": this could be rephrased to be clearer, e.g. the primary cause is the fact that MAR consistently predicts a higher runoff limit (and thereby a larger area producing runoff) than IMAU-FDM"

Done

359: why is this surprising? If both predict similar runoff in the ablation zone, wouldn't the one with higher runoff limit be expected to produce more runoff?

Our message was unclear. The "surprising" refers to the fact that melt is rather small in the vicinity of the runoff limit as compared to the ablation area. Hence, one could expect that differences in modelled ablation area runoff dominate total difference in runoff. As this is unclear, we have reworded the sentence, avoiding the word "surprising".

393: "strongly later water flux" – can you clarify what this means, or perhaps a typo (layered?)?

Thank you for pointing this out, it should read "strongly lateral", changed.

Figure A4: this is pretty wild to see this much disagreement – I hope you'll continue work like the present paper to figure out what is going on with the models here.

Thank you for the comment. Yes, the disagreement is substantial. It is challenging to simulate the interaction of firn and meltwater. We hope our work motivates further research on this topic.

**References**

Glaude, Q., Noel, B., Olesen, M., Van den Broeke, M., van de Berg, W. J., Mottram, R., et al. (2024). A factor two difference in 21st-century Greenland Ice Sheet surface mass balance projections from three regional climate models under a strong warming scenario (SSP5-8.5). *Geophysical Research Letters,* **51**(22). https://doi.org/10.1029/2024gl111902

---

## Author Comment (AC3)

**Reviewer #3**

The manuscript "Runoff from Greenland's firn area – why do MODIS, RCMs and a firn model disagree?" by Machguth et al. describes a study where the authors developed an improved algorithm to estimate the runoff limit using MODIS. This result is compared to simulation output from firn models in RCMs. A mismatch is found in the extent of the runoff area between MODIS and the modelled runoff area, as well as discrepancies that are present between firn models. This is then investigated further by analyzing detailed firn model output along a transect, which reveals that the way water retention is treated is the main cause for the discrepancies.

The study is very relevant: meltwater runoff from the ice sheets is a major souce of sea level rise, and is in fact quite uncertain (as clear from this study). The study is informative for the further firn model development acitivites in the firn community, and I think it's well suited for publication in The Cryosphere. Nevertheless, revisions are necessary, to improve the substantiation and clarity of the results. I think that a bit more analysis may be required to make the study more relevant. Now, the detailed analysis of possible causes for the model discrepancy is restricted to 1 transect only, which makes it uncertain how well the findings translate to the Greenland ice sheet runoff area as a whole.

We thank the reviewer for their comments. Below we answer all of the reviewer's comments.

**Main concerns:**

1. My biggest concern with the study is that basically only 1 transect is used to substantiated the conclusions. The argument that this transect has been used very frequently in studies, given the wealth of detailed field observations available is a bit weak, because very little field observations are in fact used. For example, no density-depth profile is shown to indicate if the density profile simulated by FDM or by MAR is in closer agreement with field observations. The majority of the results section describes the discrepancies, rather than explain them (as the title "why..." would suggest). RACMO FDM shows a strong transition from basically full ice to firn with substantial pore space around -47.5 longtidude. Immediately, this raises the question if this is general behaviour from FDM, also found further north? I think that the authors should try to find a way to make the discussion and results more robust, by analyzing larger parts of the ice sheet. Similarly, Fig. 2, A1 and A2 basically show one flowline. Are the results robust and can be extrapolated to other flowlines?

We agree that it would be good to analyse more profiles. However, we feel that we cannot do this. The reviewers also request additional explanations and interpretation of our K-transect analysis. To keep the manuscript at a manageable length we would rather address these requests than show more transects. However, we now discuss our findings in the context of the study by *Glaude et al.* (2024) which shows that the two RCMs studied here strongly divert in their predicted runoff areas for the year 2100 and that the differences are large in all regions. These independent results (i) confirm that under strong melting the two RCMs divert strongly in simulated extent of their runoff areas and (ii) that all regions of the ice sheet show this effect.

2. An aspect that I think is slightly difficult to grasp for readers not familiar with the topic, is how the different definitions of runoff are used. When it comes to modelling, a very clear definition is water that leaves the firn column. But here, a crucial difference is that in FDM, water can only leave at the bottom, whereas in MAR, it can leave at the bottom or laterally. Furthermore, bottom in FDM is much deeper down than in MAR. When it comes to MODIS, runoff is estimated from the slush limit, an assumption that is reasonable, but doesn't come without caveats.

Maybe a sketch can help to establish clear definitions. Note that in the text, it is is not clearly defined what runoff is for the used firn models. For FDM, it is not mentioned, and for MAR, only lateral runoff is explained. But it should also be explicitly mentioned that water leaving the firn column at the base is considered runoff.

We thank the reviewer for pointing out this inconsistency and we have added text to clarify the definitions.

3. Fig. 5: Here, I with panels l and k (what happened to panel i and j?) should also include water retained in the firn column as liquid, not only the refrozen part. I think this needs to be included to better explain how the differences in water retention parameterization are responsible for the differences. Moreover, I wonder to what extent it plays a role that the FDM simulated firn column is much deeper. For example, the additional refreezing in FDM shown in 5l is partly caused by the fact that the firn column is deeper in FDM than in MAR. Maybe these figures should be standardized to only show the uppermost 10, or 20m.

It is panel "i", though the way the character is rendered makes it a bit difficult to distinguish from "l". Furthermore, we omitted the letter "j" as is often done in labels due to its visual similarity with "i". The figures are already standardized and show data only for the top 20 m. No data is available for IMAU-FDM liquid water content below 20 m depth as these data are not written to output. We have now clarified to which depth the summed values refer. Liquid water contents for the top 20 m are already shown in Figs. 6c,d (RACMO-FDM) and 6i,k (MAR). The plots show that liquid water content is substantially higher in MAR. We note that Fig. 5 shows time integrated values and the meaning of time integrated liquid water content would be less clear than e.g. time integrated refreezing or runoff. For this reason, we prefer to show the temporal evolution of liquid water content in Fig. 6. We noticed that there was no indication to what depth the summed liquid water content and refreezing refer, we have added this information.

4. A bit in line with my previous comment: I find it hard to understand how the thermodynamics are different between MAR and FDM. Given that both are driven by ERA5, I assume that the overall energy balance should be more or less similar. The 10m-depth temperature, however, varies largely between MAR and FDM (Fig. A3). I find it very hard to grasp if this is only due to the percolation scheme, or that the firn is generally warmer, or if the surface energy balance is higher in MAR. Also, I think it easily gets confounding that the firn layer in FDM is so much deeper than MAR. This would allow for much more cold content to be stored at depth in FDM than can ever be captured in MAR. I wonder if the authors can give a bit more insight in this. Maybe calculate the uppermost 10m cold content of the firn layer, to see to what extent surface energy balance, and firn temperatures in general differ?

Both models simulate similar amounts of surface melt (Figs. 5e,f). This means that overall, the surface energy balance between the two models cannot differ strongly (the evidence based on melt is valid for summer, but Fig. A4 shows that winter surface temperatures are similar too). In 2012, there is a remarkable jump in RACMO albedo, directly above the 2012 runoff limit (Fig. 5c). This sudden increase in albedo also leads to a sudden drop of melt (Fig. 5e). This sudden change in RACMO albedo has been discussed in detail in the manuscript and it is not present in e.g. 2017 (Fig. 5e).

The thermodynamics of firn involve a complex interplay of thermal conductivity and latent heat release from percolating meltwater, which are both strongly influenced by firn density. Positive and negative feedback mechanisms are also present, such as the percolation depth of water

depending on firn temperature and density. If for some reason a model transports meltwater in summer to too deep a depth, then latent heat release cannot be fully undone in winter through heat conduction (porous snow and firn having relatively poor heat conduction). The amount of surface melt, controlled by the surface energy balance, plays an important role in controlling the temperature of a firn pack. However, as shown in the manuscript, the differences in melt are relatively small between the two models while firn temperatures differ very strongly (Fig. A4). For these reasons, we state that the water percolation scheme is mainly responsible for the observed differences. As mentioned in our replies to Reviewers #1 and #2, we now analyse differences in melt in more detail and provide an extended discussion of the reasons behind the large differences in simulated runoff limits.

We agree that the thickness of the firn pack plays a role. However, the depth of zero annual temperature amplitude in firn is typically around 15 m. Firn deeper than 20 m contributes to near-surface firn temperatures only through slow heat conduction.

- I would check for consistency in using the terms "saturation" and "water content". For me, saturation is the part of the pore space taken up by water. Thus, 100% saturation means all pore space filled by water. In contrast, liquid water content is most often defined per volume. Thus, 100% saturation in firn with density ~450, would mean a liquid water content of ~50%. So for example, in L111, both water content and saturation are used in the same sentence, and I'm not sure if the 7% refers to saturation, or to liquid water content. In L126, a percentage of 13% is mentioned, but it's not clear if it refers to saturation or liquid water content, since it's only written "irreducible water". Given the numbers, I think 13% is a value for saturation, not liquid water content. Anyway, I would encourage the authors to thoroughly check this throughout the manuscript, because I think now the percentages given are a mix between saturation and liquid water content values.

We have revised the text to improve clarity.

**Minor comments:**

- L8: "where meltwater is routed" --> "to route meltwater vertically"

Done

- L25: "found to perform well" is too general. Performs well in terms of? Or on what variables did it perform well? In terms of mass balance, or calculated melt?

We have added more details to the statement.

- L36: "over which mass loss takes place". I would add "mass loss through runoff", because sublimation and wind erosion are also mass loss terms.

Agreed.

- L39-40: Obviously, the choice of forcing model can have an affect as well. Forcing both RCMs with the same boundaries removes a large part of uncertainty that would come from the GCMs. This is not really a drawback, because it allows to compare RACMO and MAR on more equal footing. But maybe a brief remark on how well ERA reproduces Greenland climate and is suitable to use as forcing along the model boundaries could be justified here.

Certainly, the choice of driving re-analysis is important. However, we do not fully understand why an assessment of the various ERA products over Greenland is relevant here, with respect to

the study by *Tedstone and Machguth* (2022). Our study shows that the RCM simulated runoff limit either overestimates the MODIS runoff limit (in the case of MAR) or underestimates (in the case of RACMO-FDM). Thus, there is no systematic bias between RCM (modelled) and MODIS (observed) runoff limits. If there were a systematic bias then the RCM forcing (ERA-5) would need to be investigated, among other potential issues. We have carried out additional analysis with focus on simulated meltwater input and added a more detailed argumentation, explaining the role of meltwater and why the main issues are the differences in the firn parameterisations.

- L74: I suggest to specify that these concern output time resolution. Like: "MAR output and RACMO 1 km downscaled data" and "Output from RACMO and IMAU-FDM are at ..."

Done

- L134: "150 vs 3000 layers". Maybe specify if this covers equal total firn depth?

Both models cover the entire firn column, whose depth varies in space and time. The difference in the number of layers thus reflects their different vertical resolution, which is most pronounced at greater depths. We have added this information to the text.

- L153: "is rather insensitive": this is actually an interesting point. A figure and a full blown sensitivity study is not necessary, but maybe a statement like: using a threshold of X compared to Y affected the runoff limit only by Z kilometers would be useful here.

A sensitivity analysis on this subject is already included in *Tedstone and Machguth* (2022), visualized in Extended Data Fig. 8 and discussed therein in the Methods section under the subsection "Runoff Volumes". The sensitivity analysis shows a very low sensitivity for MAR and a somewhat larger sensitivity for RACMO. However, the latter is mainly due to the 1 km resolution of the downscaled product used in *Tedstone and Machguth* (2022) while MAR is at 10 km resolution. Hence, the low sensitivity of MAR is more representative for this study where RACMO-FDM is at 5.5 km resolution. We have added more quantitative information from *Tedstone and Machguth* (2022) rather than only a reference.

- L205: "The plateau is shorter in the north than in the south." is a too general statement given what is shown in the figures. In fact only one transect in the north or in the south is shown. Can this indeed be generalized?

We have removed this statement as it was also questioned by reviewer #2. The duration of the plateau is not relevant for the present study.

- L171: The way it was written, this actually confused me a bit, because I was looking for the masked points in Fig. 1. I would maybe write it more explicitly, like "Fig. 1 does not show retrievals from 60 to 68 ...."

Thank you, we have changed the wording.

- L245-246: "and thus could not warm further" is a bit poorly phrased, since this would also limit refreeze once temperatures reach 0 degC.

Agreed, it is obvious that 0 °C firn cannot warm further, we have modified the sentence.

- L311: Is it really inertia? The example given in the next sentence sounds to me more like non-linear behaviour.

The term was also criticised by reviewer #1. We have changed the wording, avoiding the confusing term inertia.

- L347: Please correct: "A secondary reasons"

Done

- L393: "later water flux" --> "lateral water flux" (I assume?)

Done

- Please avoid the use of green and red in one figure for color blinds (Fig. 1 and A4)

Changed

- Generally, I think the figure captions are too short.

We have added more information where needed, also with respect to comments from the other two reviewers.

**References**

Glaude, Q., Noel, B., Olesen, M., Van den Broeke, M., van de Berg, W. J., Mottram, R., et al. (2024). A factor two difference in 21st-century Greenland Ice Sheet surface mass balance projections from three regional climate models under a strong warming scenario (SSP5-8.5). *Geophysical Research Letters,* **51**(22). https://doi.org/10.1029/2024gl111902

Tedstone, A., & Machguth, H. (2022). Increasing surface runoff from Greenland's firn areas. *Nature Climate Change,* **12,** 672–676. https://doi.org/10.1038/s41558-022-01371-z

---

## Referee Report (RR1)

Review of revised "Runoff from Greenland's firn area – why do MODIS, RCMs and a firn model disagree?"

By Horst Machguth et al.

**General comments:**

I thank the authors for their care in addressing my previous concerns about the paper. While many of those concerns have been adequately addressed, I think the paper needs additional revisions to be ready for publication. The major remaining issues I identify:

**1) Introduction of RACMO vs. IMAU-FDM vs. RACMO 1km.** In my opinion, the introduction of the RACMO/IMAU products in sections 2.2 and 3.2.2., as well as Table 1, confuse the overall message of the paper. For instance, line 64 implies that the analysis will include runoff limits from IMAU-FDM, RACMO2.3p2, and RACMO 1km. In reality, the analysis focuses on results from IMAU-FDM, with brief mentions of RACMO 1km. To simplify and clarify the message, I suggest paring down these sections and simply state that you are comparing outputs from IMAU-FDM forced with RACMO2.3p2 to MAR. For example, the discussion of differences between RACMO and IMAU-FDM are not actually germane to the message of the paper and thus only serves as a distraction to the reader. For simplicity I would also suggest removing the RACMO 1km results – there are no detailed analyses of why it performs as it does.

As an example of a lack of clarity in these sections: on line 73: "Various parameters are unavailable from RACMO2.3p2 and are instead obtained from the offline firn model IMAU-FDM v1.2G henceforth IMAU-FDM. The model is forced in offline mode by RACMO2.3p2 and is run on an identical spatial grid. In the following we refer to 'MAR' for MARv3.14, to 'RACMO' for RACMO2.3p2 at native resolution of 5.5 km and we use 'RACMO 1 km' when we refer to downscaled and bias corrected RACMO2.3p2 data. But, it turns out in your analyses that RACMO2.3p2 results are not actually used, correct? And on line 80, "whose output is not available at a sufficient level of detail for the present study." Again, what does this mean? The above text had indicated that RACMO2.3p2 was part of the study, and here there it seems contradictory that it is not being used.

Also, what does it mean (line 73) that "parameters are unavailable"?

**2) The issue of the firn temperature difference between RACMO and MAR is inadequately examined.** As a central focus of the paper is on the difference in runoff between the models, the temperature of the modeled firn is a very important contributor to the amount of refreezing that occurs, and therefore warrants thorough examination. Figure 6 shows a substantial difference in firn temperature (RACMO much colder) which would indicate that RACMO has substantially more cold content available to refreeze meltwater.

Line 398 says, "An alternative explanation for the colder IMAU- FDM firn temperatures would be that the Figures 6, C3 and C5 give a wrong impression because latent heat in IMAU-FDM is released at depths greater than the max. 20 m shown in the figures." This does not make sense to me – doesn't percolating meltwater in the bucket scheme first warm the firn in a given layer, refreezing as much water as needed to bring the temperature to 0, before percolating to the next layer below? So yes, latent heat can be released at deeper temperatures, but it does not bypass releasing latent heat in shallower firn as it percolates to those greater depths.

Likewise, the abstract and conclusion state that the implementation of the bucket scheme is the cause of the disparity in modeled runoff limits, but the results and discussion section feature no discussion of the bucket scheme – if this is indeed a main conclusion of the paper, I would expect the analysis and discussion to make the scientific argument that this is the case. As it is, it comes across as rather speculative. I do think that it is a bit more nuanced than just the bucket scheme, as temperature seems like it should play a role? Or, is heat transfer considered part of the bucket scheme? In that case, a more granular conclusion (process based vs. simply using bucket scheme as a catch-all) would be appropriate. The conclusion gets a bit at that granularity (470-474), but categorizes those under "bucket scheme", while I would contend that "(iv) the firn layer in MAR is warmer" is not necessarily a bucket scheme issue.

**3) Qualitative results:** In sections 4.2.1 and 4.2.2, the results are entirely qualitative. While qualitative results are not necessarily bad, they should be accompanied by quantitative results. Phrases like "very similar" and "generally good" do not provide adequate description in a results section.

**4) Clarity issues**. I have tried to note some of these below in line-by-line comments. There are numerous places that meanings are obscured by overly verbose explanations. This may sound contradictory, but there are also numerous instances of statements being made without explanation. But, in both cases the paper loses clarity. I suggest a thorough read-through to add clarity to descriptions and to remove statements that are not germane to the topic of a particular paragraph. This is not an issue that should prevent publication but would improve the paper greatly.

**Line by line comments:**

**Line 72/Table 1:** There seems to be a discrepancy between this list and what is analyzed in the paper. For example, the list includes fac_10m and lwc_1m, but as far as I can see those are not actually included anywhere in the analyses.

**84:** Why are you considering this "relatively coarse" when it is significantly finer than you RCM/FDM grid?

**109:** "(the maximum density of pure snow)" – this claim seems unfounded. Does snow need to have impurities to have higher density? Or are you suggesting that at that density it is no longer snow? In either case this seems to be a dubious claim; for example, snow at the base of an Alaskan snowpack will regularly exceed that value.

**114:** I realize that this is not the appropriate venue for a complete description of the changs in MAR3.14, but if you are going to list them it would be appropriate to list how those changes are germane to the present work, e.g. how do the bug fixes in the cloud scheme and the rain/snow partitioning changes affect the runoff limit and/or meltwater production?

**195:** "search for parameters that show peculiar or unexpected values in the broader elevation range": related to clarity comment above – what constitutes peculiar or unexpected? How do you search for parameters?

**218:** Specify which Appendix A (1-4)

**Figure 2:** I noted in my previous review as well – in Figure 2 the names of the transects that the panels are illustrating are buried in the caption. It would be much easier if the panels had the transect name printed on the panel itself – e.g. next to or in place of the coordinates listed. If you are keeping the coordinates, than I recommending adding the degree symbol and direction, i.e. 79.2°N, -26.8°E. I realize the present formatting may be obvious to some readers but I suspect that is not the case for all.

**238:** Consider adding note in text stating the SD for RACMO 1k is not shown; I read this paragraph and went to figure 3 specifically to look at 1km SD.

**309:** "this implies that the difference in simulated runoff between MAR and IMAU-FDM increases in high-melt seasons" – can't you calculate this directly by summing that total runoff from MAR and IMAU-FDM for each year, and comparing the difference to the total runoff? That would make a much stronger argument here, especially for extending the argument beyond the K-transect.

**Paragraph 315-320:** I've read and reread this paragraph several times, and the reality is that the second half of it is not clearly written. What is the core takeaway meant to be? I think that it is that in high-melt years the additional area of the runoff zone in MAR creates a disproportionate increase in total runoff relative to RACMO (i.e., MAR usually has more runoff than RACMO, but in high-melt years it has a lot more runoff due to the increased size of the runoff area). I don't dispute the claim, but please work on the language to make your point clearer. I flagged this issue in my previous review, but a look at the changes document

shows no meaningful change. Describing a fraction of a percentage as done presently obscures the meaning here. I disagree with the assertion in the response that "Adding volumes can lead to more confusion as the analysis focuses on a transect where runoff has, in a strict sense, the unit m2". Why is it confusing to add actual values that readers can reference, even if the units are a bit odd? Regarding the unit issue – you could simply assume a unit width of the transect. Or, you could change to something like, "In 2012, total MAR runoff along the K-transect exceeds IMAU-FDM by 29 %. However, the partitioning of that difference is disproportional: in the common runoff area, MAR runoff exceeds IMAU-FDM runoff by X%, meaning that most (75%) of the additional runoff in MAR is generated in the zone above $\max Y_{rcm}^{IMAU-FDM}$.

**Paragraph 315-320:** Introducing 2019 as an example here is a bit confusing. I get that it serves as another example, but the results section mostly focuses on contrasting 2012 and 2017. I fear that a reader who is reading quickly will just assume this is another 2012/2017 comparison. If you want to keep the 2019 information, I suggest being more explicit that you are pivoting and discussing another high-melt year, e.g., first discuss 2012, then say something like, "to examine the robustness of this finding, we also examined the runoff in 2019, which was another above average melt year. Consistent with the 2012 results, for 2019 MAR predicts total runoff that is X%".

**416:** It may be appropriate to cite Van As, 2017 here:

van As, D., Bech Mikkelsen, A., Holtegaard Nielsen, M., Box, J. E., Claesson Liljedahl, L., Lindbäck, K., Pitcher, L., and Hasholt, B.: Hypsometric amplification and routing moderation of Greenland ice sheet meltwater release, The Cryosphere, 11, 1371–1386, https://doi.org/10.5194/tc-11-1371-2017, 2017.

**482:** "This means the situation where the two models diverge the most will become more frequent, simulated runoff will further diverge and uncertainty grow." I don't think this statement is entirely backed up by the analyses done in the paper – this is a claim about the transient response of the models to warming, but the manuscript does not include a rigorous analysis of the temporal trends in the modelled maximum runoff elevation (rather, it provides snapshots in time). While the statement is likely true, the paper does not provide evidence that the simulated runoff will further diverge. I suggest a simple change to acknowledge the speculative nature, such as, "This means the situation where we observe the two models diverge the most will become more frequent. We hypothesize that as a result, simulated runoff will further diverge and uncertainty will grow."

**Data availability:** I recognize that the RCM data are too big for a Zenodo repository, but there are other options for making data available. From the Copernicus Publications Data

Policy: "The best way to provide access to data is by depositing them (as well as related metadata) in FAIR-aligned reliable public data repositories, assigning digital object identifiers, and properly citing data sets as individual contributions" and "In rare cases where the data cannot be deposited publicly (e.g., because of commercial constraints), a detailed explanation of why this is the case is required." In the modern era of open science, "data can be obtained directly from the authors" does not meet the standard of publicly accessible, nor does this statement comprise a detailed explanation.

**Appendix B, Line 587:** "in IMAU-FDM, the runoff limit is typically located where summer melt exceeds annual accumulation (CRACMO − MRACMO = −0.19 ± 0.25 m w.e.); in MAR melt and accumulation at maxYMAR are similar (C − M = 0.03 ± 0.14 m w.e.)."

I don't get this – isn't this saying that the runoff limit is below the ELA in RACMO, and at the ELA in MAR? If accumulation is less than melt, you are in the ablation zone by definition?

---

## Author Response (AR2)

**Review #1:**

Suggest publication as is.

Reply: We thank the reviewer for their valuable contribution with the earlier revision. This has resulted in substantial improvement to our manuscript.

**Review #2: Suggestions for revision or reasons for rejection**

(visible to the public if the article is accepted and published)

The authors presented a thoroughly revised manuscript, with most of my concerns taken into account. I appreciate the improved description of how models define runoff, the improved explanations and expanded figure captions.

However, I do find the response to my concern about the huge discrepancy between MAR and FDM firn temperatures a bit underwhelming (page 18 in the rebuttal document). The authors basically provide some hypothesis and argumentation, but it feels like that they have to conclude that they can not explain this. However, it feels like there should be more explanation possible, and such explanations would be key to understanding the results. Can it be validated somehow that it is not an effect of spinup, or lower boundary conditions? Or maybe that there is not a bug in one of the models? I don't know. Maybe if the other 2 reviewers don't mention this, it's not too much of a concern, but I'm not really convinced.

Reply: We agree with the reviewer that we provide hypotheses and argumentation. We cannot exactly point out the reason behind differences in firn temperatures. We added the comparison of melt amounts to rule out that the disagreement is simply due to differences in the amount of simulated melt (it was also requested by a reviewer). It could well be that spin-up plays a role or bugs in the model code (see below). However, this would require series of dedicated model runs, potentially even modifications to the model codes (of IMAU-FDM and/or MAR) to enable an optimal comparison. It would also require an in-depth analysis of the code of both models. Such analysis is outside the scope off our manuscript which aims at pointing out previously unnoticed discrepancies in simulated runoff limits by MAR and IMAU-FDM. Finally, it has been shown in Vandecrux et al., (2024) focusing on the temperature at 10 m below the surface that models disagree substantially, and that there seem to be problems in MAR. It is currently assumed that they are due to numerical instabilities (Vandecrux et al., 2024) but the exact reason is not yet known. We have further emphasized this point in the manuscript.

So the authors write in the rebuttal document: "However, as shown in the manuscript, the differences in melt are relatively small between the two models while firn temperatures differ very strongly (Fig. A4). For these reasons, we state that the water percolation scheme is mainly responsible for the observed differences."

This implies that MAR routes meltwater down faster than FDM. Is this really the case? I tried to trace back a bit the values for liquid water content, because intuitively 2% saturation (not volumetric content), as mentioned in L120 is very low. Intuitively, I would think it's a volumetric contents. But it is not easy to find in the literature. Can you maybe add a citation to L120? It looks like there is no specific citation for MARv3.14?

Reply: The larger in the firn model the allowed liquid water content in the firn, the faster the warming of the firn, the faster the densification of the snowpack and the conversion from percolation zone to bare ice zone. In Glaude et al. (2024), it has been shown that the climate sensitivity of MAR (and HIRHAM) seems to be higher than RACMO. Therefore, the irreducible water saturation has been reduced in MAR below 1 m knowing that it does not impact the SMB simulated by MAR. 2% remains within the range of acceptable values.

So higher firn temperature could be caused by refreezing of deeper percolation. But first of all, MAR retains more water in the surface layers (7%, L120) than FDM (5.8%, L300). Second, in MAR, the runoff from the middle of the domain over ice layers would remove water available for downward percolation. On top of that, in Alexander et al., 2019, it is written: "The value of the irreducible water saturation in MAR is relatively high (7–10%) compared to the RACMO2.3p2 RCM (Noël et al., 2018; 1%), and could potentially lead to overestimated liquid water retention." But that paper is quite a bit old now, and maybe MAR has been modified since that?

Reply: The values indicated in the present study are correct. Alexander et al. (2019) used a former MAR version (MARv3.12). We have made this clearer in the text.

So, does MAR really route meltwater down faster, such that freezing heats up the deeper firn? If this was the case, then one would expect higher firn densities in MAR, which is not really the case in the regime where MAR is much warmer than FDM (comparing Fig. 7 with Fig. 6). Note that the mentioning from Alexander et al., 2019 of 1% for RACMO in the quote above, is not consistent with the 15% of pore space for 800 kg/m3 mentioned in L147 of the manuscript, which would be 2% volumetric content. Also, the quote from Alexander et al. 2019 is not consistent with L120 in the manuscript. Is this because of later model changes? It is actually a bit difficult to trace back the exact values and definitions, because it is not always clear from published literature, and irreducible water content/saturation probably is a parameter that modellers use to tune their models (and is also often inconsistently defined or confused in meaning). So it's not that I want to blame the authors for the confusion, but that doesn't remove the fact that I'm a bit confused. From the presented information, I simply cannot put the pieces together and understand how MAR and FDM really differ, such that MAR would be so much warmer than FDM.

Reply: As mentioned above, the value in Alexander et al. (2019) are outdated for MAR and wrong for RACMO. If MAR is too warm in depth, it is also due to the problems identified in Vandecrux et al., (2024) which concerns the whole ice sheet and not the difference in irreducible water content. In the dry snow area, for example, MAR is up to 5 °C warmer than the other models while there is no melt in this area. We have further emphasized these issues in the revised manuscript (so far, they were already mentioned on line 423). At this occasion, we have also replaced erroneous citations of the preprint of Vandecrux et al. (2024) with citations of the final published study.

So how much mm of meltwater must be refrozen in MAR to explain the higher temperature? That is an easy calculation when density and temperatures are known. Is that amount consistent with what can be expected from the produced melt, and the faster downward percolation?

I think that the warmer firn temperatures are key to understanding the differences in runoff limits, but currently, I do not manage to understand the connection with how the models treat vertical percolation. Again, I'm not sure about the view of the other reviewers, but maybe the authors can try to substantiate this aspect a bit more. It maybe also depends if the other

reviewers have concerns here as well, which would then obviously make it a more substantial problem.

Reply: Figs. 6 and Fig 7 in Vandecrux et al., (2024) convincingly show that the difference of temperature between MAR and FDM is not only linked to the irreducible water content threshold as there are also discrepancies in the bare ice area (where there is no percolation) and in the dry snow zone (where there is no surface melt).

**Minor comments**

L111, Maybe check the range 0.55 to 0.5, because in Lefebre 2003, it is written: If snow depth is zero, surface albedo varies exponentially between the ice (α = 0.58) and water (α = 0.15)

Reply: We confirm that it is well between 0.5 and 0.55. The values of Lefebre (2003) are outdated. We also clarified this in the text.

L144: Maybe use (i), (ii) and (iii) to separate the three conditions and to make the sentence easier to follow.

Reply: done

Typo caption Fig. 2 "viible"

Reply: done

**References**

Vandecrux, B., Fausto, R. S., Box, J. E., Covi, F., Hock, R., Rennermalm, Å. K., Heilig, A., Abermann, J., van As, D., Bjerre, E., Fettweis, X., Smeets, P. C. J. P., Kuipers Munneke, P., van den Broeke, M. R., Brils, M., Langen, P. L., Mottram, R., and Ahlstrøm, A. P. (2024). Recent warming trends of the Greenland ice sheet documented by historical firn and ice temperature observations and machine learning, The Cryosphere, 18, 609–631, https://doi.org/10.5194/tc-18-609-2024.

Glaude, Q., Noel, B., Olesen, M., Van den Broeke, M., van de Berg, W. J., Mottram, R., et al. (2024). A factor two difference in 21st-century Greenland ice sheet surface mass balance projections from three regional climate models under a strong warming scenario (SSP5-8.5). Geophysical Research Letters, 51, e2024GL111902. https://doi.org/10.1029/2024GL111902

**Review #2: Suggestions for revision or reasons for rejection**

General comments:

I thank the authors for their care in addressing my previous concerns about the paper. While many of those concerns have been adequately addressed, I think the paper needs additional revisions to be ready for publication. The major remaining issues I identify:

1) Introduction of RACMO vs. IMAU-FDM vs. RACMO 1km. In my opinion, the introduction of the RACMO/IMAU products in sections 2.2 and 3.2.2., as well as Table 1, confuse the overall message of the paper. For instance, line 64 implies that the analysis will include runoff limits from IMAU-FDM, RACMO2.3p2, and RACMO 1km. In reality, the analysis focuses on results from IMAU-FDM, with brief mentions of RACMO 1km. To simplify and clarify the message, I suggest paring down these sections and simply state that you are comparing outputs from IMAU-FDM forced with RACMO2.3p2 to MAR. For example, the discussion of differences between RACMO and IMAU-FDM are not actually germane to the message of the paper and thus only serves as a distraction to the reader. For simplicity I would also suggest removing the RACMO 1km results – there are no detailed analyses of why it performs as it does.

Reply: We have completely removed the RACMO 1 km results from the manuscript. However, we do not believe that removing RACMO2.3p2 parameters from the analysis would increase clarity. In the previous rounds of reviews, for example, the question was asked whether the differences between the firn models are due to different amounts of melt or accumulation. Answering this question, for example, requires analysing parameters which are only available from RACMO2.3p2, and not from IMAU-FDM. The same is the case with albedo.

As an example of a lack of clarity in these sections: on line 73: *"Various parameters are unavailable from RACMO2.3p2 and are instead obtained from the offline firn model IMAU- FDM v1.2G henceforth IMAU-FDM. The model is forced in offline mode by RACMO2.3p2 and is run on an identical spatial grid. In the following we refer to 'MAR' for MARv3.14, to 'RACMO' for RACMO2.3p2 at native resolution of 5.5 km and we use 'RACMO 1 km' when we refer to downscaled and bias corrected RACMO2.3p2 data."* But, it turns out in your analyses that RACMO2.3p2 results are not actually used, correct? And on line 80, *"whose output is not available at a sufficient level of detail for the present study."* Again, what does this mean? The above text had indicated that RACMO2.3p2 was part of the study, and here there it seems contradictory that it is not being used.

Reply: As mentioned above, this is incorrect as we use parameters from RACMO2.3p2. The RACMO2.3p2 simulated parameters ($\alpha$, C and M) are used widely in the sections results and discussion as well as in the appendix. As mentioned above, we have removed the 1 km RACMO data, and we hope that this will further increase clarity.

We agree that the statement on RACMO2.3p2 output having insufficient detail was unclear. We have removed the statement on "insufficient level of detail for this study" because this is now better explained on line 73 (see our answer below).

Also, what does it mean (line 73) that *"parameters are unavailable"*?

Reply: This means that these parameters are not written to output by RACMO2.3p2 and are unavailable for that reason. We have clarified the statement accordingly.

2) The issue of the firn temperature difference between RACMO and MAR is inadequately examined. As a central focus of the paper is on the difference in runoff between the models, the temperature of the modeled firn is a very important contributor to the amount of refreezing that occurs, and therefore warrants thorough examination. Figure 6 shows a substantial difference in firn temperature (RACMO much colder) which would indicate that RACMO has substantially more cold content available to refreeze meltwater.

Reply: The differences in firn temperatures are now discussed in more detail, further highlighting the results by Vandecrux et al. (2024). However, the exact cause of the warmer MAR firn temperatures has not yet been identified (see our replies further below) and can thus not be described in the present study.

Line 398 says, *"An alternative explanation for the colder IMAU- FDM firn temperatures would be that the Figures 6, C3 and C5 give a wrong impression because latent heat in IMAU-FDM is released at depths greater than the max. 20 m shown in the figures."* This does not make sense to me – doesn't percolating meltwater in the bucket scheme first warm the firn in a given layer, refreezing as much water as needed to bring the temperature to 0, before percolating to the next layer below? So yes, latent heat can be released at deeper temperatures, but it does not bypass releasing latent heat in shallower firn as it percolates to those greater depths.

Reply: In a bucket scheme meltwater can bypass layers if there is insufficient pore space. This leads to the situation where water can bypass thick and even cold ice layers without any interaction. This is a general limitation of bucket schemes.

Likewise, the abstract and conclusion state that the implementation of the bucket scheme is the cause of the disparity in modeled runoff limits, but the results and discussion section feature no discussion of the bucket scheme – if this is indeed a main conclusion of the paper, I would expect the analysis and discussion to make the scientific argument that this is the case. As it is, it comes across as rather speculative. I do think that it is a bit more nuanced than just the bucket scheme, as temperature seems like it should play a role? Or, is heat transfer considered part of the bucket scheme? In that case, a more granular conclusion (process based vs. simply using bucket scheme as a catch-all) would be appropriate. The conclusion gets a bit at that granularity (470-474), but categorizes those under "bucket scheme", while I would contend that "(iv) the firn layer in MAR is warmer" is not necessarily a bucket scheme issue.

Reply: We agree that the bucket scheme is not the sole reason behind the disparities. It is also correct that the term "bucket scheme" did not appear in the discussion. The latter, however, referred in detail to various key parameters of bucket schemes such as (i) the choice of irreducible water content, (ii) the chosen depth of the firn layer, (iii) the choice whether water is considered runoff only at the bottom of the firn pack or also somewhere halfway.

We have now changed the wording in the discussion to state clearly which parameters represent "design choices" in the bucket schemes. Furthermore, we have reworded the abstract and the conclusions to avoid the impression that the bucket scheme is the sole reason for (i) the disparity in runoff limits and (ii) the warmer firn.

3) Qualitative results: In sections 4.2.1 and 4.2.2, the results are entirely qualitative. While qualitative results are not necessarily bad, they should be accompanied by quantitative results. Phrases like "very similar" and "generally good" do not provide adequate description in a results section.

Reply: The two sections are deliberately written in a qualitative style because these two sections mainly describe the figures which quantify the differences. We have added mean values as well as standard deviations to the section for improved quantification.

4) Clarity issues. I have tried to note some of these below in line-by-line comments. There are numerous places that meanings are obscured by overly verbose explanations. This may sound contradictory, but there are also numerous instances of statements being made without explanation. But, in both cases the paper loses clarity. I suggest a thorough read-through to add clarity to descriptions and to remove statements that are not germane to the topic of a particular paragraph. This is not an issue that should prevent publication but would improve the paper greatly.

Reply: Thank you for these suggestions on which we comment below. In addition, we have once again streamlined the text and removed unnecessary phrases.

**Line by line comments:**

Line 72/Table 1: There seems to be a discrepancy between this list and what is analyzed in the paper. For example, the list includes fac_10m and lwc_1m, but as far as I can see those are not actually included anywhere in the analyses.

Reply: These parameters were part of the analysis, but their analysis has not yielded added insights. We have removed them from the list.

84: Why are you considering this "relatively coarse" when it is significantly finer than you RCM/FDM grid?

Reply: We have removed the words "relatively coarse". The next sentence states that the satellite sensor resolution is coarse compared to e.g. Landsat.

109: "(the maximum density of pure snow)" – this claim seems unfounded. Does snow need to have impurities to have higher density? Or are you suggesting that at that density it is no longer snow? In either case this seems to be a dubious claim; for example, snow at the base of an Alaskan snowpack will regularly exceed that value.

Reply: This statement was unclear and has been replaced by "Where surface density exceeds 450 kg m^-3, the minimum value of albedo declines between the minimum snow albedo (0.7) and clean ice albedo (0.55) as a linear function of increasing density." We have removed the word "pure" which was confusing.

114: I realize that this is not the appropriate venue for a complete description of the changs in MAR3.14, but if you are going to list them it would be appropriate to list how those changes are germane to the present work, e.g. how do the bug fixes in the cloud scheme and the rain/snow partitioning changes affect the runoff limit and/or meltwater production?

Reply: The recent improvements in MARv3.14 do not impact the present work. The problem in firn temperatures found in Vandecrux et al., (2024) using MARv3.12 is still present in MARv3.14.1 used here. To artificiality reduce the problem illustrated in Fig. 7 of Vandecrux et al., (2024), the new version of MAR in development (MARv3.14.3), applies a weak spatial smoothing between pixels of the temperature of the 1st snow layer at 20 m below the surface.

195: "search for parameters that show peculiar or unexpected values in the broader elevation range": related to clarity comment above – what constitutes peculiar or unexpected? How do you search for parameters?

Reply: This was mainly done visually, by first plotting the parameters as shown in Figs. 5, 6, 7, C3 and C4. We have reworded the sentence for improved clarity.

218: Specify which Appendix A (1-4)

Reply: done

Figure 2: I noted in my previous review as well – in Figure 2 the names of the transects that the panels are illustrating are buried in the caption. It would be much easier if the panels had the transect name printed on the panel itself – e.g. next to or in place of the coordinates listed. If you are keeping the coordinates, than I recommending adding the degree symbol and direction, i.e. 79.2oN, -26.8 oE. I realize the present formatting may be obvious to some readers but I suspect that is not the case for all.

Reply: We have implemented the suggested modifications.

238: Consider adding note in text stating the SD for RACMO 1k is not shown; I read this paragraph and went to figure 3 specifically to look at 1km SD.

Reply: 1 km RACMO has been removed from the manuscript.

309: "this implies that the difference in simulated runoff between MAR and IMAU-FDM increases in high-melt seasons" – can't you calculate this directly by summing that total runoff from MAR and IMAU-FDM for each year, and comparing the difference to the total runoff? That would make a much stronger argument here, especially for extending the argument beyond the K-transect.

Reply: We do not fully understand the comment. This is shown and quantified in Fig. 8 already? Or does the reviewer ask to do such an analysis Greenland-wide? We do not think it would add to clarity, rather to confusion as the manuscript explains the differences with a focus on a clearly defined study area (the K-transect).

Paragraph 315-320: I've read and reread this paragraph several times, and the reality is that the second half of it is not clearly written. What is the core takeaway meant to be? I think that it is that in high-melt years the additional area of the runoff zone in MAR creates a disproportionate increase in total runoff relative to RACMO (i.e., MAR usually has more runoff than RACMO, but in high-melt years it has a lot more runoff due to the increased size of the runoff area). I don't dispute the claim, but please work on the language to make your point clearer. I flagged this issue in my previous review, but a look at the changes document shows no meaningful change. Describing a fraction of a percentage as done presently obscures the meaning here. I disagree with the assertion in the response that "Adding volumes can lead to more confusion as the analysis focuses on a transect where runoff has, in a strict sense, the unit m2". Why is it confusing to add actual values that readers can reference, even if the units are a bit odd? Regarding the unit issue – you could simply assume a unit width of the transect. Or, you could change to something like, "In 2012, total MAR runoff along the K-transect exceeds IMAU-FDM by 29 %. However, the partitioning of that difference is disproportional: in the common runoff area, MAR runoff exceeds IMAU- FDM runoff by X%, meaning that most (75%) of the additional runoff in MAR is generated in the zone above maxY$%&'()*%.

Reply: We now assume unit width and state this in the text. The units are now $m^3$. Furthermore, we also use $m^3$ in the text and we hope that the previously confusing percentage values are now clearer. Furthermore, we have changed some of the wording in the paragraph.

Paragraph 315-320: Introducing 2019 as an example here is a bit confusing. I get that it serves as another example, but the results section mostly focuses on contrasting 2012 and 2017. I fear that a reader who is reading quickly will just assume this is another 2012/2017 comparison. If you want to keep the 2019 information, I suggest being more explicit that you are pivoting and discussing another high-melt year, e.g., first discuss 2012, then say something like, "to examine the robustness of this finding, we also examined the runoff in 2019, which was another above average melt year. Consistent with the 2012 results, for 2019 MAR predicts total runoff that is X%".

Reply: Agreed, done.

416: It may be appropriate to cite Van As, 2017 here:

van As, D., Bech Mikkelsen, A., Holtegaard Nielsen, M., Box, J. E., Claesson Liljedahl, L., Lindbäck, K., Pitcher, L., and Hasholt, B.: Hypsometric amplification and routing moderation of Greenland ice sheet meltwater release, The Cryosphere, 11, 1371–1386, https://doi.org/10.5194/tc-11-1371-2017, 2017.

Reply: Done.

482: "This means the situation where the two models diverge the most will become more frequent, simulated runoff will further diverge and uncertainty grow." I don't think this statement is entirely backed up by the analyses done in the paper – this is a claim about the transient response of the models to warming, but the manuscript does not include a rigorous analysis of the temporal trends in the modelled maximum runoff elevation (rather, it provides snapshots in time). While the statement is likely true, the paper does not provide evidence that the simulated runoff will further diverge. I suggest a simple change to acknowledge the speculative nature, such as, "This means the situation where we observe the two models diverge the most will become more frequent. We hypothesize that as a result, simulated runoff will further diverge and uncertainty will grow."

Reply: Done.

Data availability: I recognize that the RCM data are too big for a Zenodo repository, but there are other options for making data available. From the Copernicus Publications Data Policy: "The best way to provide access to data is by depositing them (as well as related metadata) in FAIR-aligned reliable public data repositories, assigning digital object identifiers, and properly citing data sets as individual contributions" and "In rare cases where the data cannot be deposited publicly (e.g., because of commercial constraints), a detailed explanation of why this is the case is required." In the modern era of open science, "data can be obtained directly from the authors" does not meet the standard of publicly accessible, nor does this statement comprise a detailed explanation.

Reply: For the final version of the manuscript, we will filter the RACMO data to only include the parameters we have used. This will reduce the amount of data sufficiently so that they can be uploaded to Zenodo.

Appendix B, Line 587: "in IMAU-FDM, the runoff limit is typically located where summer melt exceeds annual accumulation ($C_{RACMO} - M_{RACMO}$ = −0.19 ± 0.25 m w.e.); in MAR melt and accumulation at $maxY_{MAR}$ are similar (C − M = 0.03 ± 0.14 m w.e.)."

I don't get this – isn't this saying that the runoff limit is below the ELA in RACMO, and at the ELA in MAR? If accumulation is less than melt, you are in the ablation zone by definition?

Reply: Unfortunately, we do not fully understand the question. It is correct that the ELA must be located below the runoff limit. The ELA is defined as the elevation where the climatic mass balance is zero (Cogley et al., 2011). We now mention this definition much earlier in the manuscript. When there is substantial refreezing in the near surface, as it is the case in MAR, RACMO (and in reality near ELA and runoff limit of the Greenland ice sheet, see *Tedstone et al.,* 2025), then the ELA is located at altitudes where melt exceeds accumulation.

**References**

Cogley, J. G., Hock, R., Rasmussen, L. A., Arendt, A. A., Bauder, A., Braithwaite, R. J., et al. Glossary of glacier mass balance and related terms, Pub. L. No. 2 (2011). Paris: IHP-VII Technical Documents in Hydrology. Retrieved from http://unesdoc.unesco.org/images/0019/001925/192525e.pdf

Tedstone, A., Machguth, H., Clerx, N., Jullien, N., Picton, H., Ducrey, J., et al. (2025). Concurrent superimposed ice formation and meltwater runoff on Greenland's ice slabs. *Nature Communications*, *16*(4494). https://doi.org/10.1038/s41467-025-59237-9

---

## Author Response (AR3)

**Reply to editorial and reviewer comments**

We thank the editor and the reviewers for their very valuable feedback to our manuscript. Their critical comments have substantially improved our study.

We understand that both reviewers would have appreciated more explanations why the simulated firn temperatures of the two models vary so strongly. We acknowledge that this is an important aspect, which hopefully will be addressed in subsequent work.

Regarding the requested minor modifications, we have modified lines 344-345 as follows:

"The substantial differences between runoff limits simulated by MAR and IMAU-FDM (e.g. Figs. 3 and 4) could be caused by (i) differences in RCM simulated accumulation or melt, or (ii) differences in the parameterizations of firn and firn hydrology. A third possible reason are the differences between MAR and IMAU-FDM firn temperatures. We will discuss this aspect in the context of the differences between the firn parameterizations."

With this modification, the differences in firn temperatures are now mentioned as a potentially important factor much earlier and more prominently. We prefer to discuss the potential influence of the differences in firn temperature together with the influence of different firn parameterizations, mainly because differences in simulated firn temperatures are closely linked to differences in firn parameterizations.

Furthermore, we noticed that there was still a sentence in the manuscript that referred to Figs. 3 and 4 showing RACMO runoff limits. However, upon reviewer requests RACMO 1 km runoff limits had been removed from these figures in the previous round of revisions. Hence, we now removed the following sentence (lines 431 to 433):

"While RACMO has much fewer firn layers, the runoff limit is similarly immobile (Figs. 3 and 4) because RACMO uses a very similar bucket scheme with the same parameterization of irreducible liquid water saturation and a similarly deep firn column."

**Editorial comments:**

Dear authors,

Thank you for revising your manuscript thoroughly. The reviewers think your paper is much improved after revision. I'm happy to accept it for publishing in TC.

The reviewers raise concerns about the temperature discrepancy between the models in their previous review reports. The reviewer #1 suggested to further discuss the

discrepancy in the sentence in L 344-345 (see the review report). I agree with this suggestion. So please consider adding the discussion before finalizing your paper.

Best,

Dr. Kang Yang

Editor, The Cryosphere

**Reviewer 2:** General Comments:

I thank the authors for their consideration of my critiques and making associated edits. While I still find the lack of explanation of the large temperature discrepancies unsatisfying, I recognize that a full analysis of those differences is outside the scope of the present work (and likely a large enough task to warrant its own study). I appreciate that that language regarding this temperature difference has been edited to more clearly indicate that the reason is yet unknown.

I recommend that this paper be accepted as is.

**Reviewer 1**

It is maybe a bit unfortunate that it is not clear where the discrepancy between the models in simulated firn temperatures come from, but it is what it is. I think it is substantial improvement that this is now mentioned and discussed as a reason for why runoff limits vary between the models.

One suggestion is to also include it in the sentence in L 344-345: "The substantial differences between runoff limits simulated by MAR and IMAU-FDM (e.g. Figs. 3 and 4) could be caused by (i) differences in RCM simulated accumulation or melt, or (ii) differences in the parameterizations of firn and firn hydrology."

I would say that it could also be caused by the discrepancy in simulated firn temperature, or in any case, related to the cause of this discrepancy.